# Bypassing eigenstate thermalization with experimentally accessible quantum dynamics

**Amit Vikram[1]⋆**

**1** JILA and Center for Theory of Quantum Matter, Department of Physics, University of Colorado, Boulder CO 80309 USA

⋆ amitvikram.anand@colorado.edu

## Abstract

Eigenstate thermalization has played a prominent role as a determiner of the validity of quantum statistical mechanics since von Neumann's early works on quantum ergodicity. However, its connection to the dynamical process of quantum thermalization relies sensitively on nondegeneracy properties of the energy spectrum, as well as detailed features of individual eigenstates that are effective only over correspondingly large timescales, rendering it generically inaccessible given practical timescales and finite experimental resources. Here, we introduce the notion of energy-band thermalization to address these limitations, by coarse-graining over energy level spacings with a finite energy resolution. We show that energy-band thermalization implies the thermalization of an observable in almost all states (in any orthonormal basis) over accessible timescales without relying on microscopic properties of the energy eigenvalues or eigenstates, and conversely, can be efficiently accessed in experiments via the dynamics of a single mixed state (for a given observable) with only polynomially many resources in the system size. This allows us to directly determine thermalization, including in the presence of conserved charges, from this state: Most strikingly, if an observable thermalizes in this initial state over a finite range of times, then it must thermalize in almost all physical initial states over all longer timescales. As applications, we derive a finite-time Mazur-Suzuki inequality for quantum transport with approximately conserved charges, and establish the thermalization of local observables over finite timescales in almost all accessible states in (generally inhomogeneous) dual-unitary quantum circuits. We also propose measurement protocols for general many-qubit systems. This work initiates a rigorous treatment of quantum thermalization in terms of experimentally accessible quantities.

# 1 Introduction: Background and motivation

## 1.1 Classical statistical mechanics without ergodicity

Statistical mechanics aims to reduce the behavior of complex systems to a simple effective description in terms of statistical ensembles. The question of how to establish the validity of such a statistical description in any given system remains of foundational interest. In *classical* statistical mechanics, the conventional justification — widely invoked in introductory accounts [1–3], though often with some hesitation [1, 4] — is the ergodic hypothesis, originally due to Boltzmann. By supposing that a classical Hamiltonian system uniformly explores some region of phase space (often a surface of constant energy) over the course of its time evolution — in other words, shows ergodic dynamics — one attempts to justify describing the system by a uniform statistical distribution (the microcanonical distribution) over this region. If such a description is possible, the system is said to thermalize to this distribution.

    While the ergodic hypothesis has led to a rich mathematical theory of ergodic classical dynamics [5, 6] in sufficiently simple systems [7], it is an incredibly difficult problem to show ergodicity in systems with any realistic degree of complexity. Even if the ergodic hypothesis is assumed, the timescales required for the exploration of the phase space stretch far beyond the timescales at which statistical mechanics is known to be valid for typical observables of physical interest [8, 9]. An alternate approach is to focus on a class of "physical" observables (determined according to what is feasible to measure in a given system), and examine when such observables may admit a statistical description — without any regard for whether the system itself shows ergodic dynamics [8, 9]. For example, in a system of many particles, Khinchin [8, page 68] showed that if *single-particle* observables $a_i$ with respective mean values $\overline{a_i}$ (say) almost surely have vanishing "connected" autocorrelators at long times $t$:

$$\lim_{t \to \infty} \langle [a_i(t) - \overline{a_i}][a_i(0) - \overline{a_i}] \rangle \to 0, \tag{1}$$

where the average $\langle \cdot \rangle$ is over all trajectories in the phase space, then any observable constructed as $A = \sum_i c_i a_i$ may be described by the microcanonical ensemble over long times, in almost all *many-particle* initial states.

    This represents a considerable simplification of the problem of statistical mechanics from questions of the detailed dynamics of every conceivable many-particle observable in the phase space, as in the ergodic hypothesis, to a specific dynamical property of accessible single-particle observables. In practice, the average over *all* trajectories may be estimated by considering a few representative trajectories, though this averaging remains the only obstacle for complete accessibility. Crucially, Eq. (1) may hold even if the system has several conserved quantities (such as total momentum for a gas of pairwise-interacting particles) and fails to be formally ergodic on, e.g., an energy surface.

    The primary goal of this work is to develop a similar approach to address thermalization in quantum systems entirely in terms of the accessible properties of few-particle observables. In fact, we will show that interference effects and entanglement can be leveraged to achieve an even higher simplification than Eq. (1) in terms of the time and the number of initial states required (i.e., we can avoid a limit of infinite times and an explicit average over all possible trajectories): we only need the dynamics of a few-body observable over some *finite* time interval in a *single* easily-prepared (mixed) initial state corresponding to that observable.

## 1.2 Eigenstate thermalization in quantum statistical mechanics

The conventional approach for describing the thermalization of Hamiltonian quantum systems is based on the eigenstate thermalization hypothesis (ETH) [10–19]. Here, it is important to

refer to the eigenstates $|E_n\rangle$ corresponding to the energy levels $E_n$ of the Hamiltonian $\hat{H}$. In the simplest version of ETH, formulated by von Neumann [10], an observable $\hat{A}$ thermalizes to some thermal value $A_{\text{th}}$ in every state in an infinite time average, provided that its expectation value thermalizes in *every* energy eigenstate:

$$\langle E_n|\hat{A}|E_n\rangle \approx A_{\text{th}}, \tag{2}$$

*and* the energy levels $E_n$ are non-degenerate ($E_n \neq E_m$ if $n \neq m$). We will refer to this phenomenon specifically as eigenstate thermalization for the observable $\hat{A}$.

The more general statement of ETH [15–19] in wide present-day use, which is usually motivated [17] by comparison with random matrices — regarded as prototypes of complex quantum systems [20] — posits the following structure of the matrix elements of $\hat{A}$ in the energy eigenbasis:

$$\langle E_n|\hat{A}|E_m\rangle \approx A_{\text{th}}(E_n)\delta_{nm} + e^{-S(\overline{E})/2}f_A(E_n, E_m)R_{nm}. \tag{3}$$

Here, $A_{\text{th}}(E)$ and $f_A(E_1, E_2)$ are smooth functions of energy, $R_{nm}$ is an appropriate standard random matrix with $O(1)$ matrix elements (at least to an initial approximation that neglects correlations), and $\exp(S(\overline{E}))$ is the (smooth) density of states around energy $\overline{E} = (E_n + E_m)/2$, which is expected to scale with the Hilbert space dimension around this energy. It can be shown [15] that if the energy levels $E_n$ have non-degenerate spacings, then Eq. (3) implies the thermalization of $\hat{A}$ to the thermal value $A_{\text{th}}(E)$, at almost all times, for initial states supported near the energy $E$. Thus, in addition to eigenstate thermalization that is associated with infinite time averages, ETH also accounts for energy-dependence and thermal equilibrium. The *hypothesis* in ETH is that physically accessible observables in sufficiently complex systems satisfy Eq. (3).

But which systems are we to count as "sufficiently" complex? Again, motivated by the idea of random matrices as prototypical complex systems, one approach is to assume that the appropriate systems are those whose energy level statistics resembles the eigenvalue statistics of random matrices [17]. However, even in such systems, it is straightforward to construct observables (such as projectors onto the energy eigenstates $|E_n\rangle\langle E_n|$) that do not satisfy Eq. (3), though these are expected to be generally inaccessible in experiments. The statistics of the energy levels themselves, being observable-independent, may instead be shown to be directly related to a form of ergodic dynamics in the Hilbert space [21], reminiscent of the ergodic hypothesis. In particular, this notion is related to the ability to construct a Hermitian operator that *approximately* measures increments of time in the Hilbert space[1]. Much like the ergodic hypothesis, we do not expect this property of *dynamical* quantum ergodicity to be easily accessible in sufficiently complex quantum systems that are thermodynamically large, nor of immediate relevance to accessible few-particle observables. For example, measurements of energy level statistics have been proposed [22–24] and realized [25] only in systems with a handful ($\lesssim 10$) of qubits. We also note that some of the aforementioned measurements [23–25] as well as fluctuation-dissipation relations [26] may yield *indirect* signatures of ETH, but not a conclusive experimental determination. In contrast, macroscopic properties of the energy levels unrelated to the ergodic hypothesis, corresponding to a finite energy resolution in experiments, are accessible [22, 23] and set direct constraints on the dynamical emergence of statistical mechanics [27, 28]. The takeaway lesson that we wish to emphasize from these observations is as follows: for questions of experimental relevance to quantum statistical mechanics, it is crucial to work with a finite energy resolution rather than the microscopic properties of individual energy levels.

---

[1]This is related to the intuitive notion of ergodicity as follows: in a classical system satisfying the ergodic hypothesis, phase space coordinates may be loosely used to measure the time that would have elapsed since the system started in the intial state.

133 Separately, from a more theoretical perspective, we have argued in Ref. [21] that (observable-
134 independent) dynamical ergodicity and (observable-dependent) thermalization should be
135 understood as fundamentally different notions for quantum systems, and their connection
136 — if any — lies in additional, as yet unclear, physical restrictions on the class of interesting
137 observables. Recent numerical studies suggest that natural restrictions such as locality are
138 not sufficient to force a connection [29]. Moreover, one of the earliest numerical studies
139 of eigenstate thermalization [11] — which partly considers the question of how quantum
140 thermalization may occur even in a system with *accessible* conserved quantities (intuitively
141 "non-ergodic") — observes eigenstate thermalization even in systems without the appropriate
142 energy level distribution for dynamical ergodicity. For these reasons, although ergodicity
143 as determined by spectral statistics remains an interesting dynamical property of quantum
144 systems for *other* fundamental reasons, such as for understanding nonperturbative quantum
145 dynamics [30, 31] or for preparing highly entangled states for quantum teleportation in smaller
146 complex systems [32], we believe that an accessible approach to quantum statistical mechanics
147 should be independent of dynamical ergodicity (and therefore, the statistical properties of
148 energy levels), and closer in spirit to Eq. (1).

149 We will discuss questions of accessibility using the following rules of thumb, which are
150 conventional for quantum many-body measurements[2]. For an $N$-particle system, we consider
151 any quantity that requires a number of measurements, resolution or a time interval (relative
152 to some experimentally natural timescale) that scales at most as poly($N$) (representing some
153 polynomial in $N$) to be accessible. The lower the degree of the polynomial, the more accessible
154 the quantity. Somewhat more formally, if a quantity with value $Q \sim 1$ can be measured with a
155 finite number of resources (which fixes the normalization of $Q$ that is relevant for questions of
156 accessibility), then we expect that the following range of values of $Q$ are accessible:

$$\frac{1}{cN^{\alpha}} \leq Q \leq cN^{\alpha}, \text{ for some constants } c, \alpha > 0. \tag{4}$$

157 However, the timescales required to construct the energy eigenstates or dynamically explore
158 the Hilbert space grows at least linearly with the Hilbert space dimension $D$, which scales
159 exponentially $D \sim \exp(N)$ in the particle number $N$. The primary reason for their inaccessibility
160 is loosely the energy-time uncertainty principle: sensitivity to an energy eigenstate (zero energy
161 uncertainty) requires an infinite amount of time (infinite time uncertainty). This quantifies
162 why we do not expect ergodicity or the properties of the energy eigenstates to be accessible in
163 large many-particle systems.

164 In contrast to the ergodic hypothesis, ETH in its very formulation addresses accessible
165 observables. However, much like the ergodic hypothesis, it refers to the behavior of these
166 observables in a large family of (typically) inaccessible states: the energy eigenstates and
167 their simple superpositions (to determine the off-diagonal matrix elements). Correspondingly,
168 ETH is also considered very hard to prove in generic cases, except in sufficiently simple
169 models [18], especially in systems with certain special symmetry properties such as translation
170 invariance [34, 35]. Even in many of these simpler models, the requirements of nondegeneracy
171 of the energy levels and their spacings is a major obstacle for rigorously inferring thermalization
172 from ETH: while nondegeneracy is believed to be generic [15], it corresponds again to very fine
173 properties of the energy levels that may be difficult to show definitively in any given system.
174 Moreover, in an experiment, the resolution required to establish nondegeneracy is significantly
175 higher than that required for the statistics of energy levels [23]. Finally, even in the absence
176 of degeneracies, as is believed to be the case for "typical" systems, the time after which ETH

---

[2]We will also use the formal asymptotic notation [33] where relevant. In the $x \to \infty$ limit, $y = o(x)$ represents $y < cx$ for any constant $c$, $y = O(x)$ represents $y \leq cx$ for some constant $c$, $y = \Theta(x)$ amounts to $c_1 x \leq y \leq c_2 x$ for constants $c_1, c_2$, $y = \Omega(x)$ represents $y \geq cx$ for some $c$ and $y = \omega(x)$ represents $y > cx$ for any $c$.

can guarantee thermalization tends to be inaccessibly large. Similar conclusions apply to other eigenstate-based approaches to thermalization [36].

Let us now illustrate the magnitude of the problem with infinite time averages in ETH. Even assuming that somehow — despite their practical inaccessibility — we have a fairly good idea that an observable satisfies Eq. (3) and the system possesses nondegenerate energy level spacings, the standard constraint implied by ETH on the deviation of the expectation value $\langle \hat{A}(t) \rangle_\rho$ in an arbitrary state $\hat{\rho}$ from its thermal value $A_{\text{th}}$ in the appropriate range of energies is (e.g., [17]):

$$\lim_{T \to \infty} \frac{1}{2T} \int_{-T}^{T} dt \, \left( \langle \hat{A}(t) \rangle_\rho - A_{\text{th}} \right)^2 \leq \max_{n \neq m} \left| \langle E_n | \hat{A} | E_m \rangle \right|^2 \sim \exp[-S(\overline{E})]. \tag{5}$$

Crucially, the integral converges to values satisfying the above inequality for times larger than any level spacing in the system, typically $T \gg 2\pi/(\min_{n \neq m} |E_n - E_m|) \sim \exp(2S(\overline{E}))$ (see also the discussion around Eq. (22)). As we expect $e^{S(\overline{E})}$ to scale with the Hilbert space dimension around $\overline{E}$, we must have $S(\overline{E}) \sim N$. This means that the overall duration of time $t_{\text{ex}}$ during which $\hat{A}(t)$ fails to attain thermal equilibrium in $t \in [-T, T]$, i.e., deviates from $A_{\text{th}}$ by some $O(1)$ value, satisfies [up to $O(1)$ constants]:

$$\frac{t_{\text{ex}}}{2T} \lesssim \exp[-S(\overline{E})] \quad \Longrightarrow \quad t_{\text{ex}} \lesssim \exp[S(\overline{E})] \sim \exp(N). \tag{6}$$

In other words, ETH cannot (rigorously) constrain thermal equilibrium until the same timescale $e^{S(\overline{E})}$ associated with dynamical ergodicity. Even in practice, one can roughly estimate [15] the time scales of thermalization using the function $f(E_1, E_2)$, but only coupled with the properties of the initial state [17], which makes these estimates somewhat less "universal" than one may hope for. In terms of exact state-independent results, even for an observable that is known to satisfy the full statement of ETH, a guarantee of thermal equilibrium does not exist for accessible timescales due to Eq. (6). To our understanding, a similar shortcoming is present even in results on equilibration (without necessarily involving thermalization i.e. settling to the time averaged expectation value at almost all times, whether thermal or not) [37, 38] that rely on infinite time averages and nondegenerate spectra. For known equilibration results over finite times that account for degeneracies in the spectrum and a natural family of initial states [39], the timescale required to establish equilibration is generically $T \sim D \ln D \sim N \exp(N)$, and is therefore inaccessible in the thermodynamic limit.

From this viewpoint, ETH has successfully moved away from the ergodic hypothesis and towards accessible observables, but continues to be expressed in terms of inaccessible properties of these observables as well as the energy levels over inaccessible timescales, and in that sense remains somewhat similar to the ergodic hypothesis. Our primary goal in this context is to find more suitable observable-dependent formulations for quantum thermalization that can be accessed over *finite* timescales (but recovers eigenstate thermalization in the limit of infinite times) without requiring *any* refined knowledge of the energy levels or their properties.

## 1.3 Organization of this paper

The remaining Sections are organized as follows. Sec. 2 summarizes all of our main results at a physical level, with minimal formal details. This section primarily focuses on the introduction of a coarse-grained version of eigenstate thermalization that we call "energy-band thermalization", its implications for time-averaged thermalization as well as thermal equilibrium with accessible ranges of parameters, and accessibility through two point correlation functions such as autocorrelators.

Secs. 3 and 4 primarily review largely previously known, but in our estimation not widely known, material concerning classical and quantum (eigenstate) thermalization from a perspective that will motivate our quantum results. The subsequent sections 5-9 are concerned with our results on energy-band thermalization, its connection to the thermalization of observables in physical states, and accessibility in experiments.

Sec. 3 develops the connection between Eq. (1) and thermalization, through a slightly indirect route that considers a classical analogue of eigenstate thermalization. Our purpose in doing so is to show that one may attempt to characterize classical thermalization via this classical "eigenstate" thermalization, such that it becomes immaterial whether the system is ergodic or not, but this property is best accessed through autocorrelators. This allows one to bypass classical eigenstate thermalization, and directly address the thermalization of observables without reference to ergodicity or energy surfaces (Summary 3.3). We will use this line of reasoning to motivate our quantum developments, to move beyond quantum eigenstate thermalization.

Sec. 4 reviews the conventional quantum statement of eigenstate thermalization with infinite time averages, but also our version (Definition 4.2) of a stronger statement for thermalization in degenerate eigenspaces that seems not to be widely known at all, whose variants have appeared before in Refs. [40–42]. Despite its relevance only for infinite timescales, we will use this notion to motivate energy-band thermalization by regarding energy bands as "approximate" degenerate eigenspaces over small timescales that are not sensitive to their resolution.

In Sec. 5, we finally begin our treatment of thermalization over finite timescales, formally define energy-band thermalization (Definition 5.1), and show how it rigorously implies the thermalization of physical states over finite *and* longer timescales (Theorems 5.2 and 5.3). We will then show in Sec. 6 that energy-band thermalization can be accessed directly from autocorrelators (Proposition 6.1), reminiscent of Eq. (1), provided the thermal value of the observable is constant throughout the Hilbert space, as in Eq. (2). This allows us to directly connect autocorrelators to physical thermalization without reference to eigenstates (Summary 6.4). The generalization to energy-dependent thermal values, for a version of Eq. (3) that is nontrivial for finite timescales, requires quantum interference measures beyond autocorrelators and is developed in Sec. 7 (Theorems 7.1, 7.2 and Summary 7.3).

While Sections 3-7 mainly focus on time-averaged thermalization (which we refer to as "thermalization on average" and abbreviate as "thermalization o.a.") in the bulk of their technical statements for simplicity, we will show that our results generalize to thermal equilibrium in Sec. 8, through a trick of "cloning" the operator and taking time averages in a doubled Hilbert space (such as in Corollary 8.2). In Sec. 9 we sketch ways in which these notions can be efficiently measured in experiments, that in place of indirect signatures, allow the conclusive determination of quantum thermalization using an accessible number of resources. We conclude in Sec. 10 with some general statements and future directions.

For the Appendices, Appendix A derives a finite-time Mazur-Suzuki inequality for a finite dimensional quantum system, rigorously bounding completely positive time averages of autocorrelators in terms of approximately conserved charges. Appendix B illustrates an application of our results to show the thermalization of local observables in almost all initial states (in a physically accessible sense) of dual-unitary quantum circuits. Appendix C analyzes the connection between energy-band thermalization and eigenspace thermalization, and concludes that eigenspace thermalization is not always accessible even given energy-band thermalization; consequently, attempting to bypass the energy levels and eigenstates entirely may be in our best interest for an experimentally accessible formulation of quantum statistical mechanics. Finally, as all our proofs are relatively straightforward (and repeated) applications of the triangle and Cauchy-Schwarz inequalities [43], they are relegated to Appendix D, although what we perceive to be nontrivial physical statements are derived or explained in the course of the main

268  text or the relevant Appendices.

## 2  Summary of results

270  In this section, we will summarize our results at an intuitive level, with notation that slightly
271  differs at times from the rest of the text for easier readability. Each of the following three
272  subsections respectively summarizes Sec. 5, Sec. 6 and Sec. 7, while also incorporating the
273  contents of Secs. 8 and 9. For specific technical details, we refer to the technical Summaries 3.3,
274  6.4 and 7.3 for time-averaged thermalization, as well as Sec. 8 for thermal equilibrium and
275  Sec. 9 for measurement protocols.

### 2.1  Quantum thermalization with finite energy resolution

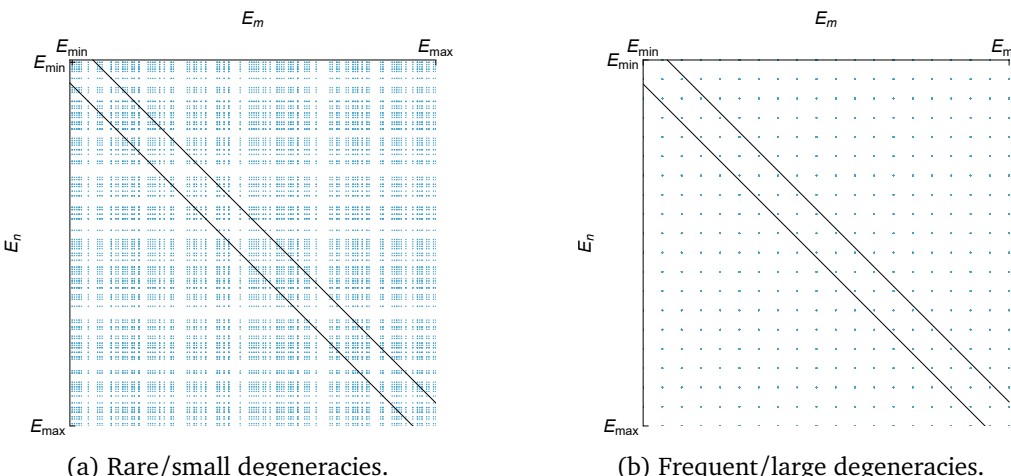

(a) Rare/small degeneracies.  (b) Frequent/large degeneracies.

Figure 1: Energy-band thermalization depicted in terms of the relevant pairs of energy
levels $(E_n, E_m)$ for two spectra with the same number of energy levels in the same range
$[E_{\min}, E_{\max}]$, but with (1a) having only occasional degeneracies, while (1b) is highly
degenerate. The solid diagonal lines represent the energy band of interest around
$E_n = E_m$ with a width $\Delta E$ [the orientation of the axes has been chosen to evoke matrix
elements, which however would be a plot of $(n, m)$ instead of $(E_n, E_m)$; see also Fig. 5].
In general, it may not be possible to accessibly distinguish the two spectra (1a) and (1b)
in an experiment, and the structure of energy level pairs inside the energy band can
be completely different in each case. However, it is sufficient to know that whichever
pairs of energy levels are in the band satisfy energy band thermalization, which is
an experimentally accessible question, to conclusively determine thermalization in
almost all initial states at all times larger than the timescale corresponding to the
inverse width of the energy band.

277  Rather than connecting ETH to observable-independent ergodic dynamics or the statistics
278  of energy levels as suggested by the traditional viewpoint [17], we will directly develop a
279  connection between the experimentally accessible dynamics of quantum observables, similar
280  to Eq. (1), and their thermalization in arbitrary physical basis states over experimentally
281  accessible timescales. It is nevertheless convenient to express this structure in terms of the
282  matrix elements of observables in the energy eigenstates (which form a preferred set of states
283  uniquely identified by the dynamics of a system). For this purpose, we will introduce the notion
284  of energy-band thermalization (whose structure is partly suggested by semiclassical proofs of

eigenstate thermalization [40, 44]) as a property of observables that does not directly depend on the energy levels of the system, but only on our (or an experimenter's) choice of an energy band resolution $\Delta E$. We will also show that this determines thermalization over finite times $T \gg 2\pi/\Delta E$, without requiring infinite time averages (summarizing Sec. 5).

Let us provide some context for why this should work. In the mathematical literature on semiclassical chaos [40, 44–50], it has been possible to prove eigenstate thermalization in the sense of Eq. (2) for classically accessible observables in certain quantum systems with a classical limit that satisfies the ergodic hypothesis, together with a (mild) suppression of off-diagonal elements in a shrinking energy band $\Delta E \to 0$ around the diagonal matrix elements. The collection of these two statements further implies classical ergodicity in the $T \to \infty$ limit. From our perspective, the fact that these statements successfully connect quantum matrix elements to purely classical dynamics strongly suggests[3] that these are appropriate notions for describing quantum thermalization even when the quantization of the energy levels is inaccessible, as in the $\hbar \to 0$ limit. As we aim to describe finite dimensional quantum systems, we do not have an explicit analogue of Planck's constant $\hbar$ available to us to safely take limits such as $T \to \infty$ without crossing the resolution of energy levels (we prefer not to take the thermodynamic limit of $N \to \infty$ particles in our results to retain applicability to finite but large systems as may be accessed in experiments); therefore, we must formulate our approach strictly for finite energy bands $\Delta E$ and finite time scales $T$.

For our initial summary of energy-band thermalization, we will specialize to situations where the thermal value $A_{\text{th}}$ does not have any dependence on energies (i.e., corresponding to Eq. (2) rather than Eq. (3)), but note that the generalization to energy-dependent thermalization is straightforward: we simply restrict our Hilbert space to energy shells with approximately constant $A_{\text{th}}(E_n) \approx A_{\text{th}}$. Our description here will largely be intuitive, and we refer to the details in subsequent sections for the full technical results.

### 2.1.1   Time-averaged thermalization

For some observable $\hat{A}$, instead of imposing Eq. (2) at the level of individual eigenstates, we consider its average behavior over the $D$ energy levels in the spectrum, when connecting pairs of energy levels whose spacing is within an energy band of width $\Delta E$:

$$[\hat{A}]_{\Delta E} \equiv \sum_{\substack{n,m: \\ |E_n - E_m| < \Delta E}} |E_n\rangle\langle E_n|\hat{A}|E_m\rangle\langle E_m|. \tag{7}$$

The relevant pairs of energy levels are depicted in Fig. 1. Now, we require that this restriction of $\hat{A}$ to the energy band thermalizes to $A_{\text{th}}$ (defined formally in Sec. 5, via Definition 5.1):

$$[\hat{A}]_{\Delta E} \approx A_{\text{th}}\hat{\mathbb{1}}. \tag{8}$$

Here, by $\hat{X} \approx \hat{Y}$, we mean that $D^{-1}\text{Tr}[(\hat{X} - \hat{Y})^2] < \epsilon^2$ for some chosen accuracy $\epsilon \ll 1$. Eq. (8) should intuitively be understood as a version of eigenstate thermalization, Eq. (2), that lacks sufficient energy resolution $\Delta E$ to necessarily identify individual eigenstates (which would be achieved if $\Delta E \to 0$) but, as we will shortly see, nevertheless implies Eq. (2) for almost all

---

[3]Why this is suggested by classical theorems is roughly as follows: in these theorems, the classical limit $\hbar \to 0$, which is very similar to the thermodynamic limit $N \to \infty$ in that both increase the Hilbert space dimension to infinity, is taken *before* the $t \to \infty$ limit in these proofs. The classical infinite time average does not correspond to timescales that diverge as $\hbar \to 0$ (which are already set to $\infty$ when taking the classical limit), and in fact necessarily converges over $O(\hbar^0)$ timescales, therefore being quite different from the quantum infinite time average. This means that the timescales referred to as "infinite" in the classical limit are actually well below the timescales needed to resolve the energy levels (which grow as a power of $1/\hbar$ by the uncertainty principle for quantized classical systems, and as $\exp(N)$ for fully quantum many-body systems).

320 eigenstates. Crucially, such a notion is fully insensitive to the question of how many collections
321 of the energy levels of the system populate the energy band $\Delta E$, and in this way ensures a
322 complete logical separation of energy-band thermalization from the energy levels.

323     More nontrivially, we can address thermalization dynamics in arbtirary sets of initial states.
324 Given that Eq. (8) holds, we will show that in any complete orthonormal basis of states $|\psi_k\rangle$
325 in the Hilbert space, $\hat{A}$ thermalizes in almost all basis states for any time average larger than
326 $2\pi/\Delta E$ (formally, Theorems 5.2 and 5.3 in Sec. 5):

$$\frac{1}{2T} \int_{t_0-T}^{t_0+T} dt \, \langle\psi_k(t)| \, \hat{A} \, |\psi_k(t)\rangle \approx A_{\text{th}}, \quad \text{for almost all } k, \text{ for any } T \gg \frac{2\pi}{\Delta E}. \tag{9}$$

327 Here, $x \gg y$ should be rigorously understood as $x > cy$ for some large constant $c > 1$, while $t_0$
328 is an arbitrarily chosen reference time (which could be any real number). For example, if $\hat{A}$ is a
329 projector, then this statement implies that projective measurements of $\hat{A}$ (which has outcomes 0
330 or 1 for a projector), collected over arbitrary times in a sufficiently long interval with the same
331 initial state for each run, acquire thermal statistics for almost all basis states. Moreover, if the
332 system is initialized to any given $|\psi_k\rangle$ and the observable $\hat{A}$ is weakly coupled to an external
333 system $M$ over the time interval $[t_0 - T, t_0 + T]$ (e.g. via an interaction of the form $g\hat{A} \otimes \hat{X}_M$),
334 then observables in the external system are directly sensitive to the time-average of $\hat{A}$ in Eq. (9)
335 according to first-order perturbation theory [51], making this a naturally observable form of
336 thermalization.

337     We emphasize that there is no mathematical restriction on the basis $|\psi_k\rangle$ to which this
338 result applies. For example, one could take the basis to be that of the energy eigenstates $|E_k\rangle$,
339 in which case Eq. (9) automatically implies eigenstate thermalization in the sense of Eq. (2)
340 for *almost all* energy eigenstates:

$$\langle E_n| \, \hat{A} \, |E_n\rangle \approx A_{\text{th}}, \text{ for almost all } n. \tag{10}$$

341 Unlike Eq. (3), we note that the deviations of the diagonal matrix elements from $A_{\text{th}}$ are
342 constrained only up to an $O(1)$ resolution $\epsilon$, which is a significant factor behind our statements
343 being restricted to "almost all" basis states.

344     By "almost all" basis states, we mean that the fraction of basis states (in any basis) to
345 which the above statement [Eq. (9)] applies approaches 1 as our experimental resolution and
346 timescale of observation improve (i.e., $\epsilon \to 0$, $T\Delta E \to \infty$). We describe this result as applying
347 to "almost all" basis states as this is the statement we expect to be accessible in experiments
348 for large systems; however, with sufficient resolution sensitive to the Hilbert space dimension
349 $D$, our results actually allow us to constrain thermalization in every single state in the Hilbert
350 space as well (which we expect may be possible for sufficiently small systems). Eq. (9) applies
351 independently of whether the energy levels are degenerate or show any kind of statistical
352 correlations. Most significantly, limits such as $T \to \infty$ or $\Delta E \to 0$ are not *required* for this result,
353 but may be taken if desired and allowed by the specific context. Our main physical emphasis
354 here is that knowing the properties of observables even at a low resolution corresponding to an
355 energy band $\Delta E$ is sufficient to establish thermalization over finite or infinite time averages
356 in an arbitrary orthonormal basis, without requiring any higher resolution knowledge of the
357 energy levels *a priori* such as in Eq. (2).

358     We will refer to such basis states in *any* orthonormal basis as "physical states" in the bulk of
359 the manuscript, mainly to emphasize that they can be experimentally prepared as outcomes of
360 a projective measurement [52] of any complete set of commuting observables that one may
361 have access to. For example, this could be the set of computational basis states in an $N$-qubit
362 system [as in Eq. (81)],

$$|0_1 0_2 \ldots 0_N\rangle, \, |0_1 0_2 \ldots 1_N\rangle, \, \ldots, \, |1_1 1_2 \ldots 0_N\rangle, \, |1_1 1_2 \ldots 1_N\rangle, \tag{11}$$

which may be efficiently accessed by measuring the computational state $\{0, 1\}$ of each qubit in parallel. We also use this terminology of "physical states" to emphasize the contrast with "typical states" (chosen according to the Haar measure [20]) in the Hilbert space, which have been shown [53, 54] to trivially thermalize all local observables with probability 1 in the thermodynamic limit. However, such "typicality" results have no implications for thermalization in any given orthonormal basis of states accessible in an experiment (such as computational basis states): any countable set of states forms a measure zero set according to the Haar measure, due to which no given basis state is necessarily a "typical" state. Eq. (9) is therefore significantly stronger than typicality results in applying to almost all states in any *finite* set of sufficient size, and to observables with specific properties.

We can additionally make a statement about *instantaneous* thermal equilibrium from Eq. (8) that, in a sense, extends Eq. (10) beyond individual energy eigenstates to arbitrary states of narrow energy width. Given any small energy shell $\mathcal{E}_{\Delta E}$ of width less than $\Delta E$ anywhere in the spectrum, and a basis of states $|\psi_q(\mathcal{E}_{\Delta E})\rangle$ supported only within this energy shell [which we expect to be $\Theta(D)$ in number for accessible $\Delta E$], then almost all states in the basis have thermal expectation values by default (Proposition 5.4):

$$\langle\psi_q(\mathcal{E}_{\Delta E})|\,\hat{A}\,|\psi_q(\mathcal{E}_{\Delta E})\rangle \approx A_{\text{th}}, \quad \text{for almost all } q. \tag{12}$$

It therefore follows that questions of thermalization dynamics for an observable satisfying energy-band thermalization generally arise only for states with energy support over a range larger than the corresponding energy-band width $\Delta E$.

### 2.1.2 Thermal equilibrium

In addition to time averages, one is also interested in showing that the observable attains thermal equilibrium at almost all times in a finite time range $t \in [t_0 - T, t_0 + T]$ centered at some time $t_0$, in a more general family of initial states than Eq. (12). Unlike the distinction between eigenstate thermalization in Eq. (2) and the full statement of ETH in Eq. (3), where ETH assumes a separate off-diagonal structure for thermal equilibrium, we will show that the same notion of energy-band thermalization is also sufficient to describe thermal equilibrium, though with a slightly weaker specification of initial states than for time-averaged thermalization in Eq. (9). Specifically, we will require (in Sec. 8) that a cloned version of $\hat{A}$ in a doubled Hilbert space (each with dynamics generated by $\hat{H}$, so that the doubled Hamiltonian is $\hat{H} \otimes \hat{\mathbb{1}} + \hat{\mathbb{1}} \otimes \hat{H}$) also satisfies energy-band thermalization for the doubled Hamiltonian:

$$[(\hat{A} - A_{\text{th}}\hat{\mathbb{1}}) \otimes (\hat{A} - A_{\text{th}}\hat{\mathbb{1}})]_{\Delta E} \approx 0. \tag{13}$$

We note that this cloning of the operator is a mere formal device, and such properties can be experimentally established entirely within a single copy of the system. This "cloning" strategy may be regarded as a quantum version of a theorem in classical ergodic theory [6], which states that a dynamical system is weakly mixing (which loosely corresponds to thermal equilibrium) if and only if its "cloned" version with two copies of the system is ergodic (which loosely corresponds to time-averaged thermalization). This is especially convenient for us, as it allows us to use the same conceptual notion of energy-band thermalization (but applied to the observable vs. its cloned version) to address both time-averaged thermalization and instantaneous thermal equilibrium over finite timescales.

First, let us consider constraints on matrix elements in the energy eigenbasis. For any observable $\hat{A}$ with finite eigenvalues (as $D \to \infty$), the squared diagonal fluctuations around $A_{\text{th}}$ as well as squared off-diagonal matrix elements (which we will collectively just call "squared matrix element fluctuations") must show the $O(D^{-1}) \sim \exp(-S(\overline{E}))$ scaling in ETH [Eq. (3)]

*on average*:

$$\frac{1}{D^2}\left[\sum_n(\langle E_n|\,\hat{A}\,|E_n\rangle - A_{\text{th}})^2 + \sum_{n\neq m}\left|\langle E_n|\,\hat{A}\,|E_m\rangle\right|^2\right] = D^{-1}\left(\frac{1}{D}\,\text{Tr}\left[(\hat{A}-A_{\text{th}}\hat{\mathbb{1}})^2\right]\right) = O(D^{-1}). \tag{14}$$

More nontrivial statements should constrain the *distribution* of these squared matrix element fluctuations given such an average, e.g., their variance can in principle be as large as a finite constant as $D \to \infty$. With this context, Eq. (13) implies that

$$\frac{1}{D^2}\left[\sum_n(\langle E_n|\,\hat{A}\,|E_n\rangle - A_{\text{th}})^4 + \sum_{n\neq m}\left|\langle E_n|\,\hat{A}\,|E_m\rangle\right|^4\right] \approx 0, \tag{15}$$

which is a nontrivial statement that the variance of the squared matrix element fluctuations is smaller than it could be, at least with $O(1)$ resolution (therefore, potentially still parametrically larger than the mean). That this is a statement without any restriction to an energy band is crucial for thermal equilibrium: it allows us to effectively constrain dynamics without time averaging (as instantaneous expectation values at arbitrary times involve all matrix elements of the observable). However, this statement also lacks the $O(D^{-1})$ resolution of (squared) off-diagonal ETH [Eq. (3)] (if one insists on accessible values of $\epsilon$), due to which it will turn out that we will need a more coarse-grained notion of "almost all" states to discuss the (accessible) dynamics of thermal equilibrium.

    If we restrict ourselves to statements that can be made with an experimentally accessible resolution, then Eq. (13) is sufficient for the observable to attain thermal equilibrium in two different ways, both involving sets of initial states. A direct variant of Eq. (9) is obtained by replacing $(\hat{A}-A_{\text{th}}\hat{\mathbb{1}})$ with its cloned version $(\hat{A}-A_{\text{th}}\hat{\mathbb{1}})\otimes(\hat{A}-A_{\text{th}}\hat{\mathbb{1}})$ and noting that the set of pairs of basis states $|\psi_k\rangle\otimes|\psi_\ell\rangle$ forms a complete orthonormal basis for the doubled Hilbert space. This is the statement that for almost all *pairs* of initial states $(|\psi_k\rangle, |\psi_\ell\rangle)$ in any basis, the fluctuations of the expectation value of $\hat{A}$ around $A_{\text{th}}$ have negligible correlations over a sufficiently large time interval (formally, Corollary 8.1):

$$\frac{1}{2T}\int_{t_0-T}^{t_0+T}dt\,\left(\langle\psi_k(t)|\,\hat{A}\,|\psi_k(t)\rangle - A_{\text{th}}\right)\left(\langle\psi_\ell(t)|\,\hat{A}\,|\psi_\ell(t)\rangle - A_{\text{th}}\right) \approx 0,$$

$$\text{for almost all }(k,\ell),\text{ for any }T\gg\frac{2\pi}{\Delta E}. \tag{16}$$

This lack of correlation between basis states can occur in two notably extreme ways (among a continuum of possibilities): (1) $\langle\psi_k(t)|\,\hat{A}\,|\psi_k(t)\rangle \approx A_{\text{th}}$ at almost all times for almost all $k$ (corresponding to thermal equilibrium in almost all basis states), or (2) $\langle\psi_k(t)|\,\hat{A}\,|\psi_k(t)\rangle$ and $\langle\psi_\ell(t)|\,\hat{A}\,|\psi_\ell(t)\rangle$ have sufficiently independent dynamics for $k\neq\ell$ (e.g., independent erratic fluctuations). However, we need an inaccessibly high resolution to differentiate these two possibilities, and therefore cannot sharpen this result to completely imply thermal equilibrium for a many-body system while insisting on accessibility (without, possibly, additional assumptions that we haven't found to be straightforward to rigorously formulate so far).

    Another statement directly concerning thermal equilibrium that follows from cloned energy-band thermalization pertains to *sufficiently large* statistical ensembles of basis states. For an ensemble $\hat{\rho}(t) = p_k|\psi_k(t)\rangle\langle\psi_k(t)|$, where $p_k \geq 0$ is the probability of the basis state $|\psi_k\rangle$ in the ensemble (normalized to $\sum_k p_k = 1$), we can measure the "size" $\mu(\rho)$ of the ensemble via

$$\mu(\rho) \equiv \frac{1}{D\sum_k p_k^2} \in [0,1], \tag{17}$$

439 which is an estimate of the fraction of basis states whose probability $p_k$ is comparable to
440 that of a "typical" state. Then, as long as $\mu(\rho) = \Theta(1) > 0$ [i.e. $\mu(\rho) > c_\mu$, for some small
441 constant $c_\mu$ determined by our experimental resolution that remains fixed even if $D \to \infty$],
442 which quantifies what we mean by a sufficiently large ensemble, Eq. (13) implies that thermal
443 equilibrium is attained at almost all times (formally, Corollary 8.2),

$$\sum_k p_k \langle \psi_k(t) | \hat{A} | \psi_k(t) \rangle \approx A_{\text{th}}, \text{ for almost all } t \in [t_0 - T, t_0 + T], \text{ for any } T \gg 2\pi/\Delta E. \quad (18)$$

444 Here, "almost all $t$" should be interpreted as "all $t$ except a subset of $[t_0 - T, t_0 + T]$ whose
445 length is smaller than $T/c_T$ for some large constant $c_T > 1$". Further, we note that the set of all
446 mixed states with $\mu(\rho) > c_\mu$ is, in an analogous sense, the set of "almost all" mixed states in the
447 Hilbert space (especially as we can take $c_\mu \to 0$ after $D \to \infty$ in a thermodynamic limit). In
448 this sense, Eq. (18) is a statement about thermal equilibrium in "almost all" states that, though
449 weaker in the admissible class of physical states than Eq. (9) that applies to pure states, is
450 considerably stronger than typicality results in applying to *all* mixed states of sufficient $\mu(\rho)$
451 (in any basis) without exception.

452    Once again, Eq. (18) appears to be the strongest statement on thermal equilibrium that we
453 can make with an *accessible* experimental resolution for thermodynamically large systems. For
454 comparison, it is directly analogous to weak-mixing in classical ergodic theory [5–7], which
455 only applies to ensembles of states; many natural observables in classical systems cannot
456 attain thermal equilibrium in any individual state even with the strongest forms of mixing, as
457 discussed in Sec. 8. We note the interesting tradeoff[4] in these "accessible" statements between
458 time-averaged thermalization in almost all states in Eq. (9), and state-averaged thermalization
459 at almost all times in Eq. (18). However, we also emphasize again that with sufficiently high
460 resolution (sensitive to the Hilbert space dimension $D$) as may be possible in small systems,
461 Eq. (18) can also be applied at the level of *individual* pure states $|\psi_k\rangle$ of the system.

462    In the context of many-body systems, Eq. (18) has particularly relevant implications for
463 the problem of thermalization of a subsystem in contact with a thermal bath [27, 28, 55, 56].
464 Specifically, consider splitting an $N$ particle system into an initially nonthermal "core" subsystem
465 $C$ with $N_C$ particles in an arbitrary pure state $|\psi\rangle_C$, with the remaining $N_B = N - N_C$ particles
466 functioning as a "statistical bath", being initialized to a (possibly, but not necessarily, thermal)
467 statistical ensemble of states via a mixed state density operator $\hat{\rho}_B$. The initial state of the
468 overall system is:

$$\hat{\rho}_\psi(0) = |\psi\rangle_C \langle \psi | \otimes \hat{\rho}_B. \quad (19)$$

469 Then, Eq. (18) implies that for *any* such partition into subsystems $(C, B)$ and *every* pure initial
470 state $|\psi\rangle_C$ in $C$, the observable $\hat{A}$ attains thermal equilibrium at almost all times in this state,

$$\text{Tr}[\hat{\rho}_\psi(t) \hat{A}] \approx A_{\text{th}}, \text{ for } all \ |\psi\rangle_C, \text{ for almost all } t \in [t_0 - T, t_0 + T], \text{ for any } T \gg 2\pi/\Delta E, \quad (20)$$

471 as long as $\hat{\rho}_B$ is sufficiently closed to a maximally mixed state, $\text{Tr}_B[\hat{\rho}_B^2] \leq c/D$ for some large
472 but accessible constant $c \gg 1$ (in the $D \to \infty$ limit), which also requires that $\exp(N_C)$ is
473 accessibly small, i.e., $N_C \leq c_3 (\log N)^\alpha$ for some suitable constants $c_3 \geq 0$, $\alpha \geq 0$ as in Eq. (4) [as
474 $\text{Tr}_B[\hat{\rho}_B^2] \geq 1/D_B$, where $D_B$ is the dimension of the $B$ subsystem]. We note that the observable
475 $\hat{A}$ may be an arbitrary observable in the Hilbert space; but if it happens to be an observable in
476 $C$, then Eq. (20) describes the thermalization of observables in the initially nonthermal core
477 $C$ in contact with a statistical bath $B$, due to the closed system dynamics generated by the
478 Hamiltonian $\hat{H}$.

---

[4]Precisely due to accounting for classical systems in which equilibration is not universally possible in individual states at specific times, it does not seem straightforward to do away with both the time and state averages altogether without additional assumptions (which must then be violated by suitable observables in classical systems).

479   Collectively, Eqs. (8), (9), (13), (16), and (18) establish a fully quantum connection between
480   properties of observables with a finite energy resolution and their thermalization over finite time
481   scales. Moreover, Eqs. (10) and (15) show that direct connections to eigenstate thermalization
482   [Eq. (2)] and (to a lesser extent) off-diagonal ETH fluctuations [Eq. (3)] may be established
483   even with such finite resolution.

## 2.2   Bypassing energy levels with finite-time autocorrelators

485   We will now describe how energy-band thermalization [Eqs. (8), (13)] is advantageous com-
486   pared to conventional statements of ETH [Eqs. (2), (3)] not only in the ability to access finite
487   time scales, but also because it can be directly accessed from an experimental measurement of
488   the dynamics of a single mixed initial state (corresponding to Sec. 6).

489   To motivate this relation, let us first consider a system with nondegenerate energy levels
490   $E_n$, and some observable of interest $A$, whose thermal value we again initially assume to be
491   constant throughout the spectrum (i.e., "global thermalization"). The *connected* autocorrelator
492   of $\hat{A}$, which is given by the autocorrelator of $\widehat{\delta A} = (\hat{A} - A_{\text{th}} \hat{\mathbb{1}})$, can be averaged over infinite time
493   to give

$$\lim_{T \to \infty} \frac{1}{2T} \int_{-T}^{T} dt \, \text{Tr}[\widehat{\delta A}(t) \widehat{\delta A}(0)] = \sum_n \left| \langle E_n | \hat{A} | E_n \rangle - A_{\text{th}} \right|^2. \tag{21}$$

494   This is a version of the Mazur-Suzuki equality [57, 58] for finite-dimensional quantum systems,
495   which has also been implicitly used for ETH in e.g. [41, 59]. In particular, the (approximate)
496   vanishing of the time-averaged autocorrelator (up to some normalization that we will presently
497   ignore) implies eigenstate thermalization in the sense of Eq. (2), and the reverse implication
498   also holds. From our point of view, however, the infinite time average is a major obstacle due
499   to its inaccessibility (for reasons similar to those connecting ETH to thermalization, see the
500   discussion around Eq. (6)). Quantitatively, for the $T \to \infty$ limit to begin approaching the right
501   hand side, we need

$$T \gg \frac{2\pi}{\min_{n \neq m} |E_n - E_m|}, \tag{22}$$

502   which is typically quadratic $T \gg D^2$ in the number of energy levels $D$, and therefore expo-
503   nentially long $T \gg \exp(N)$ in the number of particles $N$. This is much greater than the time
504   required to establish even, e.g., dynamical ergodicity ($T \sim D$). It would be desirable, and in
505   fact, essential for practical purposes, to constrain thermalization with finite time averages.

506   Once again, in the previously mentioned mathematical literature on semiclassical chaos [40,
507   49], the connection obtained between classical ergodicity and eigenstate thermalization in
508   quantized classical systems proceeds through *classical* autocorrelators as in Eq. (1). As these
509   are insensitive to the quantization of the spectrum, the existence of these results again strongly
510   suggests that accessible, finite-time autocorrelators should be able to constrain thermalization
511   in fully quantum systems.

512   To motivate our approach, let us consider how to constrain energy-band thermalization in
513   the sense of Eq. (8). One of the key challenges here is to account for a large number of phase
514   factors at finite times:

$$\text{Tr}[\widehat{\delta A}(t) \widehat{\delta A}(0)] = \sum_{n,m} e^{i(E_n - E_m)t} \left| \langle E_n | \widehat{\delta A} | E_m \rangle \right|^2. \tag{23}$$

515   In general, the vanishing of an autocorrelator of this form does not necessarily imply the
516   vanishing of any of the individual terms on the right hand side, because most of the terms add
517   up out of phase and could individually be quite large while their sum remains small. It would

518 seem that quantum interference effects pose a problem that must be tackled to achieve our
519 goal.

520      Our solution, which generalizes a more specific technique in the semiclassical literature [49],
521 is to consider time-averages of autocorrelators weighted by non-negative functions $w_+(t) \geq 0$
522 whose Fourier transforms $\widetilde{w}_+(\delta E) \geq 0$ are also non-negative[5]. For convenience, we call these
523 functions "completely positive" for want of better terminology (which is quite distinct from
524 completely positive maps on density operators [52]). As an aside, such functions also play an
525 important role in the fast scrambling problem, and more generally in the question of obtaining
526 many-body speed limits related to the energy-time uncertainty principle for different classes of
527 systems [27, 28].

528      We expect our more general formulation of these results (compared to semiclassical results)
529 in terms of completely positive functions to be crucial for utilizing these methods in experiments,
530 as described in Sec. 9. In particular, this allows us to better access specific energy bands and
531 energy shells in a continuous time system by *erratically* sampling a discrete set of times with
532 random spacings (we note that continuous time as in the semiclassical case is inaccessible in
533 experiments, while a regular sampling of discrete times would lead to rapid periodicity in the
534 energy domain, potentially including contributions from additional regions of the spectrum
535 where it may not be clear that the autocorrelator should decay).

536      If we consider completely positive time averages, we obtain, in place of the traditional
537 autocorrelators in Eq. (23), the weighted autocorrelator:

$$\int dt \, w_+(t) \operatorname{Tr}[\widehat{\delta A}(t)\widehat{\delta A}(0)] = \sum_{n,m} \widetilde{w}_+(E_n - E_m)\left|\langle E_n|\widehat{\delta A}|E_m\rangle\right|^2. \tag{24}$$

538 Here, all terms on the right hand side are non-negative, so a decay of the autocorrelator must
539 imply the smallness of nonvanishing terms. If the autocorrelator decays over a timescale $T_w$,
540 i.e., when $w_+(t)$ has support on a range of times[6] $t \in [-T_w, T_w]$, then we show that $\hat{A}$ satisfies
541 energy-band thermalization over all corresponding inverse energy scales (Proposition 6.1 in
542 Sec. 6):

$$[\hat{A}]_{\Delta E} \approx A_{\text{th}} \hat{\mathbb{1}}, \text{ for all } \Delta E \ll \frac{2\pi}{T_w}, \tag{25}$$

543 which, by Eq. (9), also implies that time-averaged thermalization is necessarily achieved for
544 almost all basis states over longer but comparable timescales to the decay of the autocorrelator
545 (Corollary 6.2),

$$\frac{1}{2T} \int_{t_0-T}^{t_0+T} dt \, \langle \psi_k(t)| \, \hat{A} \, |\psi_k(t)\rangle \approx A_{\text{th}}, \quad \text{for almost all } k, \text{ for any } T \gg T_w. \tag{26}$$

546      At this stage we no longer need to refer to the energy-band: it follows that the decay of the
547 connected autocorrelator Eq. (24) over a timescale $T_w$ implies the time-averaged thermalization
548 of the observable over almost all physical states, and vice versa (see Summary 6.4). A similar
549 result applies, via the cloning strategy (Sec. 8), to thermal equilibrium as in Eqs. (16) and (18).
550 This achieves our complete bypass of the finer properties of the energy levels, allowing us to
551 express quantum thermalization in terms of autocorrelators similar to Eq. (1), with a slight but
552 entirely tractable complication of completely positive time averages to account for quantum
553 interference effects in the traditional autocorrelator.

---

[5]The semiclassical approach, e.g. [49], is to integrate $\operatorname{Tr}[\widehat{\delta A}(t_1)\widehat{\delta A}(t_2)]$ over $t_1, t_2 \in [-T, T]$, which is contained as a special case of our approach, and then take the classical limit $\hbar \to 0$, and *then* $T \to \infty$.

[6]Here, we note that due to the $\widetilde{w}_+(\delta E) \geq 0$ requirement, it is not possible in general to shift the time average of the *autocorrelator* by a reference time $t_0$ to $[t_0 - T_w, t_0 + T_w]$, which would introduce additional phase factors $\widetilde{w}_+(\delta E) \to \widetilde{w}_+(\delta E)e^{-i\delta E t_0}$ unless $t_0 = 0$. In practice, this means that nonthermal behavior near $t = 0$ always contributes.

In most of the text, we restrict ourselves to autocorrelators of projectors $\hat{\Pi}_A$ as the "simplest" observables which can be most directly accessed in experiments via projective measurements [52], noting that other more complicated observables can be expanded in terms of projectors to their eigenbasis (e.g. $\hat{A} = \sum_a A_a \hat{\Pi}_a$) and are usually inferred from these projective measurements. For projectors, it suffices to prepare a single initial state $\hat{\rho}_A \propto \hat{\Pi}_A$ to determine their thermalization in (almost) all possible initial states. A key takeaway from this approach is then that the dynamics of a projector observable in a single quantum state fully determines whether it thermalizes in arbitrary "physical" initial states.

In general, these results imply that provided we have a way to exactly compute or measure autocorrelators in some system even over finite time scales, it is possible to definitively establish thermalization to some global, energy-independent value $A_{\text{th}}$ for any observable over all longer timescales. This requires that the autocorrelators thermalize within experimentally accessible timescales; in systems in which autocorrelator thermalization takes significantly longer, we can not conclude thermalization in physical states from this approach without waiting for the relevant timescales, however inaccessible. In the Appendices, we illustrate two applications of this approach.

First, the Mazur-Suzuki equality given by Eq. (21) with $\widehat{\delta A}$ replaced by $\hat{A}$ is of considerable interest in quantum transport, for connecting conserved quantities to autocorrelators (via inequalities). We show in App. A how Eq. (24) may be used to derive a finite-time inequality for approximately conserved charges for arbitrary finite-dimensional quantum systems, which has been a significant open problem [60, 61] to our understanding. Schematically, given an orthogonal (with respect to the trace inner product of operators) but not necessarily complete set of approximately conserved quantities $\hat{Q}_k$, with each dynamically fluctuating around its $t = 0$ value by $\langle \delta Q_k^2 \rangle_{T_w}$ on average, over the timescale $|t| \lesssim T_w$ corresponding to the window of averaging $w_+(t)$, our inequality is:

$$\int dt \, w_+(t) \text{Tr}[\hat{A}(t)\hat{A}(0)] \gtrsim \sum_k \frac{1}{\text{Tr}[\hat{Q}_k^2]} \left| \left| \text{Tr}[\hat{A}\hat{Q}_k] \right| - \sqrt{\langle \delta Q_k^2 \rangle_{T_w} \text{Tr}[\hat{A}^2]} \right|^2 \qquad (27)$$

[see Eq. (A.11) for a precise statement].

If the right hand side is sufficiently large, this shows that even a small set of known approximate and accessible conserved charges can prevent the thermalization of the autocorrelator to $A_{\text{th}}^2$ (for global thermalization). In contrast, the above bound is weaker if the dynamical fluctuations $\langle \delta Q_k^2 \rangle_{T_w}$ are large (i.e., the charge is not close to being conserved) and can potentially allow thermalization.

Second, we illustrate in App. B how our results may be used to show the thermalization of local observables in almost all basis states for any choice of basis in dual-unitary quantum circuits, in which autocorrelators can be computed exactly [62–64], with thermal values that are uniform across all states and do not depend on any conserved quantities such as energy.

## 2.3   Interference effects for energy shells and conserved charges

There is yet another challenge that requires a nontrivial modification to this approach with no classical or semiclassical counterpart. In the above intuitive discussion, we have assumed that $A_{\text{th}}(E_n) \approx A_{\text{th}}$ is constant throughout the spectrum (except for the Mazur-Suzuki inequality, which is formally independent of the thermal value). We have already stated that energy-band thermalization generalizes in a trivial way if $A_{\text{th}}(E_n)$ is some smooth non-constant function of energy — by restricting the Hilbert space to the part of the spectrum where $A_{\text{th}}(E_n)$ is approximately constant, which is fully accounted for in Sec. 5. However, connected autocorrelators such as Eq. (24) cannot be rigorously restricted to some region of the spectrum in general, because there is no single thermal value $A_{\text{th}}$ in terms of which we may define $\widehat{\delta A}$. Given that

we want $\hat{A}$ to be an experimental observable, it is not at all obvious that its restriction to an energy shell can be directly implemented experimentally.

Further, the conventional theoretical approach of regularizing, e.g., with a thermal density operator to focus on a narrow energy window is unsuitable for our purposes, because it would require showing properties of this external thermal state that are unlikely to be accessible (even for a rigorously justified regularization strategy in Ref. [28]). To illustrate the problem, let us say that $\hat{A} = \hat{a}_1 \otimes \hat{\mathbb{1}}_{(N-1)}$ is a single-particle observable $\hat{a}_1$ on particle 1. One strategy for restricting to some narrow range of energies, often useful in rigorous statements on thermalization [28,56], is to initialize the remaining $(N-1)$ particles in some state $\hat{\rho} \propto \hat{\mathbb{1}}_1 \otimes \hat{\rho}_{N-1}$ with narrow energy support and measure the autocorrelator of $\widehat{\delta a}_1 = \hat{a}_1 - a_{\text{th}}\hat{\mathbb{1}}_1$ in this state. In attempting to derive expressions analogous to Eq. (24), we get:

$$\int dt \, w_+(t) \text{Tr}\left[\hat{\rho}\left(\widehat{\delta a}_1(t)\widehat{\delta a}_1(0) \otimes \hat{\mathbb{1}}_{N-1}\right)\right] = \sum_{n,m,r} \widetilde{w}_+(E_n - E_m)\langle E_n|\widehat{\delta a}_1|E_m\rangle\langle E_m|\hat{\rho}|E_r\rangle\langle E_r|\widehat{\delta a}_1|E_n\rangle,$$
(28)

for example (one can get more symmetric expressions by factorizing $\hat{\rho} = (\hat{\rho}^{1/2})^2$, but this actually becomes more intractable here). Unless $\hat{\rho}$ is diagonal in the energy eigenbasis (which is typically hard to ensure or establish), it is not generally possible to show that the right hand side consists of non-negative terms (due to involving the product of at least three matrices, while Eq. (24) involved only two identical Hermitian matrices) without a detailed knowledge of the off-diagonal elements of $\hat{\rho}$ in relation to $\hat{a}_1$. Further, since $\hat{\rho}$ is generally supported on a *large* subsystem, even its diagonal elements may be highly fluctuating in the energy eigenbasis of the full system, and it may not be clear how to focus unambiguously on a specific energy shell of interest. We therefore lose the rigorous advantages of Eq. (24) in the conventional setting of thermalization in a narrow energy range.

The strategy we find most feasible, described in Sec. 7, is to take full advantage of quantum interference effects by moving beyond autocorrelators and the standard setting of the thermalization process, while remaining in an effectively "infinite temperature" problem where the initial state is allowed to be supported on the entire spectrum. First, let us define

$$\widehat{\delta A}_{\mathcal{E}} \equiv \hat{A} - A_{\text{th}}(\mathcal{E})\hat{\mathbb{1}},$$
(29)

where $A_{\text{th}}(\mathcal{E})$ is the thermal value in an energy shell of range $\mathcal{E}$ (assumed constant within the energy shell). Then we specifically show that a class of interference measures, which we generically call "quantum dynamical echoes", related to Loschmidt echoes [65–67],

$$\text{Tr}[e^{-i\hat{H}t_1}\widehat{\delta A}_{\mathcal{E}}e^{i\hat{H}t_2}\widehat{\delta A}_{\mathcal{E}}]$$
(30)

can be time-averaged with completely positive weight functions $w_+((t_1 + t_2)/2)$ and $v_+(t_1 - t_2)$ of linear combinations of $t_1$ and $t_2$ to constrain energy-band thermalization within the energy shell spanning $\mathcal{E}$, with any desired energy-band width $\Delta E$. While $w_+(t)$ plays a similar role to its counterpart in Eq. (24), the role of $v_+(t)$ is to allow (even retroactively) choosing a specific energy shell. This rigorously connects the properties of the energy eigenstates to certain Loschmidt echoes, which has so far remained one of the missing links between two key objects of interest that fall under the umbrella of "quantum chaos" [68], which we take to refer to a collection of loosely related notions with clear physical distinctions that are often used to study complex quantum systems (see, e.g., [69] for a summary of some links). We also show that the microscopic properties of energy levels may once again be bypassed completely, by directly relating the decay of these echoes to the thermalization of any physical basis states within the energy shell $\mathcal{E}$, as described for time-averaged thermalization in Summary 7.3, with its generalization to thermal equilibrium in Sec. 8.

639 Moreover, we show in Sec. 9 that these echoes can be efficiently measured in experiments
640 without any direct measurement of $A_{\mathrm{th}}(\mathcal{E})$ or a prior choice of the energy shell (which may be
641 obtained or enforced via classical post-processing), by measuring the following more straight-
642 forward "quantum dynamical echoes" (defined schematically here; see Secs. 7 and 9 for the
643 precise version):

$$L_{ab}(t_1, t_2) = \mathrm{Tr}[e^{-i\hat{H}t_1} \hat{\mathcal{A}}_a e^{i\hat{H}t_2} \hat{\mathcal{A}}_b], \tag{31}$$

644 where $a, b \in \{\varnothing, A\}$, and $\hat{\mathcal{A}}_\varnothing \equiv 1$, $\hat{\mathcal{A}}_A \equiv \hat{A}$. We propose an efficient measurement protocol
645 for these quantities involving projectors (as well as the autocorrelator used in Eq. (24) for
646 global thermalization, which is simpler to measure directly) by a modification of a quantum
647 interference measurement protocol for the spectral form factor [22], which uses an auxiliary
648 qubit to implement controlled dynamical evolution by the Hamiltonian $\hat{H}$ over different times.

649 This strategy can also account for macroscopic conserved charges[7]: in the presence of a
650 single conserved charge $\hat{Q}$ (for example) such that the thermal value $A_{\mathrm{th}}(\mathcal{E}, \mathcal{Q})$ depends [17,70]
651 on $Q$, being effectively constant over a range of eigenvalues $Q \in \mathcal{Q}$, we can update Eq. (29) to
652 the form:

$$\widehat{\delta A}_{\mathcal{E},\mathcal{Q}} \equiv \hat{A} - A_{\mathrm{th}}(\mathcal{E}, \mathcal{Q})\hat{\mathbb{1}}. \tag{32}$$

653 Provided that we have the ability to implement the symmetry transformation generated by $\hat{Q}$,
654 our results also imply that thermalization in the presence of accessible conserved quantities
655 can be addressed through echoes of the form:

$$\mathrm{Tr}[e^{-i\hat{H}t_1} e^{-i\hat{Q}s_1} \widehat{\delta A}_{\mathcal{E},\mathcal{Q}} e^{i\hat{Q}s_2} e^{i\hat{H}t_2} \widehat{\delta A}_{\mathcal{E},\mathcal{Q}}]. \tag{33}$$

656 Once again, these echoes can be accessed through measurements that do not require prior
657 knowledge of $A_{\mathrm{th}}(\mathcal{E}, \mathcal{Q})$:

$$L_{ab}(t_1, t_2; s_1, s_2) = \mathrm{Tr}[e^{-i\hat{H}t_1} e^{-i\hat{Q}s_1} \hat{\mathcal{A}}_a e^{i\hat{Q}s_2} e^{i\hat{H}t_2} \hat{\mathcal{A}}_b]. \tag{34}$$

658 Here, it is not essential for the charge $\hat{Q}_{\mathrm{exp}}$ that may be accessible in experiments to be the
659 exact conserved charge $\hat{Q}$, but just for the transformation $\exp(-i\hat{Q}_{\mathrm{exp}}s)$ generated by it for
660 accessible values of $s$ to be sufficiently close to the true symmetry transformation $\exp(-i\hat{Q}s)$ to
661 allow a determination of the above echo to within experimental errors (see also [71]).

662 In summary, we develop a conceptual framework for quantum statistical mechanics that
663 relies entirely on the measurable dynamical properties of a single state (per observable),
664 as illustrated through explicit measurement protocols. This parallels what we find to be
665 the simplest dynamical justification for classical statistical mechanics [8] (Eq. (1)), but with
666 additional interesting consequences of interference effects in the quantum case that allow us to
667 explicitly address finite time intervals, a finite energy resolution, and the presence of accessible
668 conserved charges in an exact manner.

## 669 3  Classical "eigenstate" thermalization without ergodicity

670 Now, with a primary view of motivating our quantum results, we will describe one possible
671 approach to the classical autocorrelator, Eq. (1), as the determiner of thermalization in classical

---

[7]As all Hamiltonians conserve linear combinations of the energy projectors $|E_n\rangle\langle E_n|$ (including all projectors
in degenerate eigenspaces), by a conserved charge we strictly just mean some observable $\hat{Q}$ formed by such a
linear combination — which commutes with the Hamiltonian $[\hat{H}, \hat{Q}] = 0$ — on which the thermal value of our
observable $\hat{A}$ of interest may strongly depend. By "macroscopic", we mean a conserved charge with an inaccessibly
large number of eigenvalues. For discrete conserved charges (with an accessible number of eigenvalues), it should
be straightforward to project onto a desired eigenspace of the charge.

dynamical systems. Rather than take a more direct route [8], we will follow a somewhat contrived path by first considering a classical notion of eigenstate thermalization as a possible explanation for thermalization. While classical eigenstate thermalization has been discussed before in, e.g., Ref. [41] *assuming* the ergodic hypothesis, it is crucial for our purposes to formulate this notion for more general non-ergodic systems to resemble the quantum situation and demonstrate independence from ergodicity.

After showing that classical eigenstate thermalization determines the thermalization of observables, we will arrive at the autocorrelator of Eq. (1) by asking how classical eigenstate thermalization may in turn be determined by the dynamics of the system. We will arrive at an equivalence between the thermalization of an observable in autocorrelators, eigenstates, and typical initial states — which allows us to directly connect the thermalization of autocorrelators to that in other states, without relying on eigenstate thermalization. This mirrors the route we will follow in quantum mechanics, but in a simpler context.

As our purpose in this section is to set up intuition rather than focus on the technical details, we will make a number of artificial simplifying assumptions and restrict ourselves to thermalization over (classically) infinite time averages.

## 3.1 Setup: classical ergodicity

Let us consider a classical system with phase space $\mathcal{P}$, typically having several degrees of freedom. It is impractical to precisely measure the state of the system with the accuracy of a single point $x \in \mathcal{P}$ (specifying both coordinates $q$ and momenta $p$); instead, one is more often concerned with whether the system is in a larger *collection* of states $A \subseteq \mathcal{P}$, say corresponding to a specific range of values of a few-body observable, or of collective thermodynamic quantities. To probe such questions, one defines the observable

$$\Pi_A(x) = \begin{cases} 1, & \text{if } x \in A, \\ 0, & \text{if } x \notin A. \end{cases} \tag{35}$$

When integrated against a distribution $\rho(x, t)$ (evolving with time $t$), $\Pi_A(x)$ measures the probability $\pi_A[\rho(x, t)]$ of the outcome $A$ in the distribution:

$$\pi_A[\rho(x, t)] = \int_{x \in \mathcal{P}} d\mu(x) \, \rho(x, t) \Pi_A(x). \tag{36}$$

Here, $\mu$ measures volumes in phase space, given e.g. by the Liouville measure $\int d^f q d^f p$ in Hamiltonian mechanics with $f$ degrees of freedom. It is convenient to focus on "bulky" initial distributions, in which any zero measure region has zero probability (i.e., for which the probability density is not singular):

$$\pi_B[\rho(x, t)] = 0 \quad \text{if} \quad \mu(B) = 0. \tag{37}$$

This anticipates quantum density operators by enforcing some uncertainty (nonzero phase space volume) in regions of nonzero probability.

In an ergodic system (or if $\mathcal{P}$ is chosen to be a region of phase space in which the system is ergodic), where almost all trajectories explore every region of $\mathcal{P}$, the time-average of the probability $\pi_A$ in any bulky distribution is proportional to the fraction of $\mathcal{P}$ occupied by $A$:

$$\int \overline{dt} \, \pi_A[\rho(x, t)] = \frac{\mu(A)}{\mu(\mathcal{P})} \pi_{\mathcal{P}}[\rho], \tag{38}$$

where $\int \overline{dt}$ is shorthand for time averaging, for which we may choose, e.g.:

$$\int \overline{dt} \, f(t) \equiv \lim_{T \to \infty} \frac{1}{T} \int_0^T dt \, f(t) \quad \text{or} \quad \lim_{T \to \infty} \frac{1}{2T} \int_{-T}^T dt \, f(t), \tag{39}$$

and $\pi_{\mathcal{P}}[\rho]$ represents the total probability in the initial distribution:

$$\pi_{\mathcal{P}}[\rho] \equiv \int_{x \in \mathcal{P}} d\mu(x)\, \rho(x, 0). \tag{40}$$

That the (time-averaged) probability becomes independent of the initial state is *formally* a powerful guarantee for statistical mechanics, that allows us to use the microcanonical ensemble on $\mathcal{P}$, $\rho_{mc}(x) = 1/\mu(\mathcal{P})$ (for time averages) in place of specific initial states. We will call this property *thermalization-on-average*, or thermalization *o.a.* for short. In an ergodic system, thermalization *o.a.* holds for every bulky initial distribution $\rho(x, 0)$ and every region $A$, no matter how refined or difficult to measure.

## 3.2 "Eigenstate" thermalization in non-ergodic systems

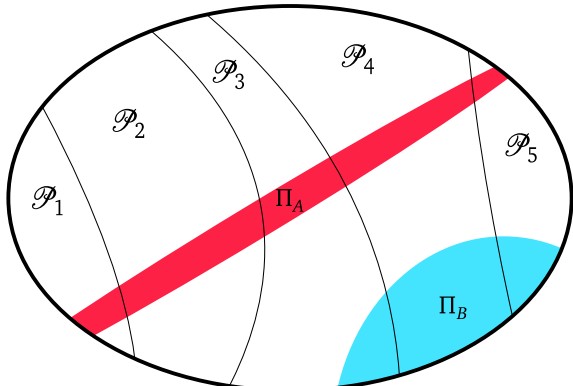

Figure 2: Schematic depiction of thermalization without ergodicity in a classical phase space $\mathcal{P}$ with 5 ergodic subsets $\mathcal{P}_k$. The observable $\Pi_A$ is considered to be equally distributed over these ergodic subsets (i.e., shows eigenstate thermalization), and must therefore thermalize in almost all initial states. The observable $\Pi_B$ is restricted to a few of these subsets, and cannot thermalize almost everywhere. Separately, this figure also provides schematic intuition for the Mazur-Suzuki inequality [Eq. (27) and App. A], where the conserved quantities are regarded to be projectors onto the ergodic subspaces $Q_k = \Pi_{\mathcal{P}_k}$, among which we have access to, say, $k = 3, 4, 5$. Here, $\Pi_B$ has a large overlap with these conserved quantities and its autocorrelator may fail to decay to its thermal value by the inequality, while $\Pi_A$ has sufficiently low overlap with all these conserved quantities to still allow autocorrelator thermalization (which in turn would imply eigenstate thermalization as well as thermalization).

Ergodicity is a difficult property to either prove or verify in any sufficiently large phase space, not least because of the sensitivity required to ensure that every infinitesimal region of $\mathcal{P}$ is visited by generic trajectories (in a 3D gas of $N \to \infty$ particles, this would require having precise knowledge of the $3N$ positions and momenta of every particle). Instead of relying on ergodicity, let us ask when Eq. (38) can be true for a given measurement $A$ in a non-ergodic system. This is of extreme physical relevance: all nontrivial Hamiltonian systems are non-ergodic due to the conservation of energy, as well as the frequent presence of various other conservation laws (such as total momentum or angular momentum, depending on the system under consideration and nature of interactions). Further, experimental uncertainties ensure that any initial state is statistically distributed over a range of energies $\Delta E$. Moreover, as ergodicity in the dynamical sense is difficult (in most cases, near-impossible) to establish for a sufficiently complex system, so a given system may as well be nonergodic for all practical

purposes. In this case, the phase space can be decomposed into an unknown number of subsets $\mathcal{P}_k$:

$$\mathcal{P} = \bigcup_k \mathcal{P}_k, \tag{41}$$

each of which is ergodic. These are typically surfaces of fixed energy, as well as fixed values of other conserved quantities (if present). Through a slight misuse of terminology that anticipates quantum mechanics, we will call the sets $\mathcal{P}_k$ — or more precisely, uniform distributions supported entirely on one of these sets — the "energy eigenstates" of the classical Hamiltonian (for example, because they correspond to the classical limit of the quantum energy eigenstates, and satisfy similar properties such as having a definite value of energy and being invariant under time evolution[8]). An identification between constant energy surfaces and eigenstates was proposed in Ref. [41] in a related context, *assuming* ergodicity on each energy surface; for our perspective, it is instead crucial that we identify eigenstates with the subsets $\mathcal{P}_k$ especially if the system is non-ergodic.

For simplicity, we will assume that there is a finite number $M$ of $\mathcal{P}_k$, each with nonzero measure[9] $\mu(\mathcal{P}_k) > 0$; formally, this simplifies the technical details of several arguments, and the case of a continuum of energy surfaces (typical in classical systems) can be recovered via a continuum limit $M \to \infty$ without altering any of our conclusions. This assumption allows us to focus on the physics of the problem, rather than the formal definition of induced measures on subsets and their integration. It also further anticipates quantum mechanics, in which there is a minimum phase space volume [72, 73] connected to the inverse purity of a pure state. With this simplification, due to the ergodicity of each subset, any observable $\Pi_A$ satisfies

$$\int \overline{dt} \; \pi_{A \cap \mathcal{P}_k}[\rho(x, t)] = \frac{\mu(A \cap \mathcal{P}_k)}{\mu(\mathcal{P}_k)} \pi_{\mathcal{P}_k}[\rho], \tag{42}$$

in any bulky distribution $\rho(x, 0)$.

While such non-ergodic systems are not conventionally associated with thermalization, let us suppose that $\Pi_A$ does somehow thermalize o.a., so that $\Pi_A$ is subject to (microcanonical) statistical mechanics despite the non-ergodicity of the system. The question we are interested in is the following[10]: "which observables $\Pi_A$ thermalize o.a. in a possibly non-ergodic system?"

This is answered by:

**Proposition 3.1** (Classical eigenstate thermalization implies thermalization o.a.). *An observable* $\Pi_A$ *thermalizes o.a. in every bulky initial distribution* $\rho(x, 0)$ *[as in Eq. (38)] if the region A is proportionally distributed in the ergodic subsets* $\mathcal{P}_k$ *according to their volume:*

$$\frac{\mu(A \cap \mathcal{P}_k)}{\mu(\mathcal{P}_k)} = \frac{\mu(A)}{\mu(\mathcal{P})}. \tag{43}$$

*Proof.* See App. D.1.1.                                                                 □

If Eq. (43) is satisfied, we say that the observable $\Pi_A$ shows *eigenstate thermalization*: its expectation value in every ergodic subset $\mathcal{P}_k$ is the thermal value $\mu(A)/\mu(\mathcal{P})$. Much like the quantum statement of eigenstate thermalization, classical eigenstate thermalization simplifies

---

[8]This parallel can be taken further in the Koopman-von Neumann framework [5] of classical Hilbert spaces comprised of functions on $\mathcal{P}$, in which classical dynamics has well-defined eigenstates.

[9]Any subsets of zero measure can be trivially absorbed into any nonzero measure subset without affecting its ergodicity.

[10]For simplicity, we consider exact thermalization. In more practical considerations, one must deal with approximate thermalization, i.e. being within some distance $\epsilon$ of the thermal value. It is straightforward to extend the classical statements in the section to this case, and we will unavoidably return to approximate thermalization in the quantum context.

the problem of thermalization in arbitrary (bulky) initial states to the problem of thermalization in $M$ specific initial states (the classical "energy eigenstates"), each given by uniform distribution supported entirely on one of the $\mathcal{P}_k$. But this does not make the problem accessible: we eventually want to take $M \to \infty$ for typical classical systems, with the number of initial states/measurements and the energy resolution required to access these states becoming insurmountable.

Fortunately, we can show that a further simplification is possible — we only need a single initial distribution to determine the behavior of all $M$ eigenstates [assuming that $\mu(A) > 0$]:

**Proposition 3.2** (A single initial distribution determines classical eigenstate thermalization). *For classical eigenstate thermalization [Eq. (43)] to hold for an observable $\Pi_A$, it is sufficient that $\Pi_A$ thermalizes o.a.:*

$$\int \overline{\mathrm{d}t}\ \pi_A[\rho_A(x,t)] = \frac{\mu(A)}{\mu(\mathcal{P})}, \tag{44}$$

*in a single initial distribution given by (with $\pi_{\mathcal{P}}(\rho_A) = 1$):*

$$\rho_A(x,0) \equiv \frac{1}{\mu(A)}\Pi_A(x). \tag{45}$$

*Proof.* See App. D.1.2. The intuition behind the proof (which is reminiscent of a proof of semiclassical wavefunction ergodicity theorems [40, 44, 49], which we will revisit for quantum thermalization) is as follows. The difference between the left and right hand sides of Eq. (44),

$$\left[\left[\int \overline{\mathrm{d}t}\ \pi_A[\rho_A(x,t)] - \frac{\mu(A)}{\mu(\mathcal{P})}\right]\right] \tag{46}$$

measures a weighted variance of the set of thermal values $\mu(A \cap \mathcal{P}_k)/\mu(\mathcal{P}_k)$ in the different eigenstates, while $\mu(A)/\mu(\mathcal{P})$ is their mean with the same weights. Thus, thermalization implies a vanishing of the weighted variance, and all eigenstates' thermal values must then be equal to their weighted mean, giving Eq. (43). $\square$

Taken together, Propositions 3.1 and 3.2 imply that the thermalization o.a. of the observable $\Pi_A$ in an arbitrary (bulky) initial state is completely determined by its thermalization in a single initial state $\rho_A$ — a situation in which we can hope to rigorously access thermalization problems with a manageable supply of (theoretical or experimental) resources, at least in principle. See Fig. 2 for a schematic depiction. Stated formally,

**Summary 3.3** (A single state determines if an observable thermalizes o.a. in all states). *The following statements are equivalent:*

    *1. The observable $\Pi_A$ thermalizes o.a. in the initial state $\rho_A$.*

    *2. The observable $\Pi_A$ shows eigenstate thermalization in all classical energy eigenstates $\mathcal{P}_k$.*

    *3. The observable $\Pi_A$ thermalizes o.a. in all bulky initial states $\rho(x,0)$.*

For all practical purposes, this allows us to directly connect thermalization in $\rho_A$ to thermalization in bulky initial states, without ever addressing the energy eigenstates, given that the autocorrelator is a much more accessible quantity than thermalization in eigenstates. We note that being a result about bulky initial states, there may always exist a measure zero subset of states (e.g., periodic orbits) that fail to thermalize. This "bypassing" of the eigenstates to directly connect some measurable dynamical feature of the observable to its thermalization in typical states will be the main theme underlying our quantum results.

### 3.3   Crossover remarks

We will now make a few remarks that will set up the crossover into quantum systems in the following sections. First, we emphasize that none of the results in this section had any dependence on the values of energy $E_k = H(x \in \mathcal{P}_k)$ on the ergodic subsets. This suggests that any quantum counterpart should be independent of the properties of energy levels, including degeneracies. This is complicated by interference effects in quantum mechanics, but we will show that such a formulation can be achieved in the course of this paper, including without the $T \to \infty$ limit.

Second, let us note that there is no difficulty whatsoever for, e.g. finite temperature thermalization, in which the thermal value of $\Pi_A$ may differ in different (dynamically closed) regions of the phase space (e.g., as a function of energy) instead of being uniformly $\mu(A)/\mu(\mathcal{P})$ everywhere. Here, we merely restrict our considerations to a subset of the phase space in which the thermal value is roughly constant. Such a restriction is more difficult in quantum mechanics, and we will have to adopt a strategy based on interference effects outlined in Sec. 7.

Lastly, let us consider how one might prepare the state in Eq. (45) for a measurement to directly probe the thermalization of $\Pi_A$. The simplest strategy is to take a representative set of points within the set $A$, and hope for a convergence of an average over this set of points to the actual average over all of $A$. This would usually entail showing the *typicality* of the behavior of $\Pi_A$ within $A$ for these points to represent the full ensemble. We will see that in the quantum counterpart of these measurements, Sec. 9, this typicality is automatically guaranteed across different measurement outcomes.

## 4   Quantum eigenspace thermalization with degeneracies

Now, we turn to the problem of quantum thermalization in a $D$-dimensional Hilbert space $\mathcal{H}$. As has long been recognized [10, 17, 19], it is impossible to have every conceivable observable thermalize (even o.a.) under Hamiltonian quantum dynamics with a complete orthonormal basis of energy levels $|E_n\rangle$ (which we assume are indexed in ascending order, $E_{n+1} \geq E_n$). This is partly because a Hamiltonian quantum system is intrinsically non-ergodic, in the classical sense, in the Hilbert space [10] due to conserving the overlaps $|\langle E_n|\psi(t)\rangle|^2$ of a state $|\psi(t)\rangle$ with the energy eigenstates $|E_n\rangle$.

Let us therefore ask a question analogous to what we used to motivate Proposition 3.1: "which observables $\hat{\Pi}_A$ thermalize o.a. in a quantum system?" Here, $\hat{\Pi}_A$ denotes a projector ($\hat{\Pi}_A^2 = \hat{\Pi}_A$) that may project onto a specific set of measurement outcomes of an observable $\hat{A}$, for example. As before, thermalization o.a. refers to thermalization "on average" over a range of times; we postpone a consideration of thermal equilibrium at individual times to Sec. 8. We will also generally consider mixed initial states described by density operators $\hat{\rho}$, whose evolution is given by:

$$\hat{\rho}(t) = e^{-i\hat{H}t}\hat{\rho}\,e^{i\hat{H}t}. \tag{47}$$

We will review two kinds of thermalization criteria for the observable for infinite time intervals, while setting up notation for the subsequent sections. One is the well known notion of eigenstate thermalization in the sense of Eq. (2), which implies thermalization o.a. given nondegenerate spectra (Sec. 4.1). Another criterion — which appears to be widely unknown (by our accounting, it has been discussed at least thrice before [40–42], but is not usually invoked in discussions of eigenstate thermalization in which nondegeneracy still plays a key role [17–19, 59]) — is that in systems with degenerate spectra, one can impose thermalization in degenerate *eigenspaces*, from which thermalization in other states follows (Sec. 4.2). Our version is somewhat stricter than Refs. [41, 42], which will prove useful later. We discuss this

phenomenon under the separate name of *eigenspace thermalization* to emphasize a conceptual distinction: in degenerate systems, an observable could satisfy eigenstate thermalization in a specific eigenbasis without eigenspace thermalization, and thereby fail to thermalize in most states. After discussing these notions, we will use eigenspace thermalization to qualitatively motivate energy-band thermalization in Sec. 4.3 (postponing a quantitative treatment to later sections), by arguing that energy levels within an energy band of resolution $\Delta E$ may be regarded as approximately degenerate.

## 4.1 Eigenstate thermalization + non-degeneracy $\implies$ idealized thermalization

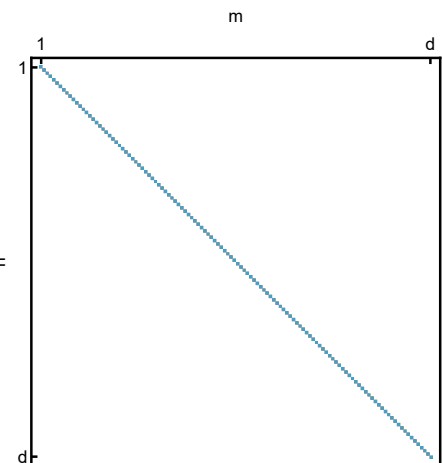

Figure 3: Schematic depiction of eigenstate thermalization in terms of the relevant matrix elements between pairs $(n, m)$ of the energy levels $(E_n, E_m)$; essentially, only the $n = m$ pairs are relevant. This structure is formally simple and convenient, and looks the same for any system in terms of matrix elements. However, in terms of energies, this is only a system-dependent subset of pairs having an energy gap of $\Delta E = 0$ if there are degeneracies, and cannot be completely specified in terms of an energy gap. Further, $\Delta E = 0$ is not accessible in many-body systems, nor is it sufficient to conclude thermalization in physically relevant situations.

To discuss eigenstate thermalization as a criterion for thermalization at a more technical level [10–19], we will find it convenient to restrict ourselves to a subspace $\Sigma_d \subseteq \mathcal{H}$ of $d$ energy levels to which initial states are assumed to belong, that we will formally call an energy shell (physically, $\Sigma_d$ may represent a narrow range of energies $E_n \in [E_{\min}, E_{\max}]$, and accounts for initial states with support in this range[11]; it may also exclude some energy levels if desired, for example by restricting to an eigenspace of a conserved charge $\hat{Q}$ within this range). For energy shells of experimentally accessible energy widths $\Delta E$, we expect that $d \sim D/c_\Sigma$ where $c_\Sigma$ is some "accessible" constant [i.e., in the range of $Q$ in Eq. (4); see also Eq. (188)]. Thus, $d$ is usually an inaccessibly large quantity, comparable to $D$.

The (microcanonical) thermal value of $\hat{\Pi}_A$ in $\Sigma_d$, i.e., its expectation value in the uniform distribution / maximally mixed state in $\Sigma_d$, is

$$\langle \hat{\Pi}_A \rangle_{\Sigma_d} = \frac{1}{d} \operatorname{Tr}[\hat{\Pi}_A \hat{\Pi}_{\Sigma_d}], \tag{48}$$

---

[11]For thermalization in a more general class of initial states with a narrow spread but with asymptotic tails outside a finite range, see [15]; for the behavior of states without a narrow spread, see also [74]. We do not consider these more realistic cases for simplicity, and expect that the extension to such cases is straightforward along the lines of Ref. [15], by expanding around energy shells.

where $\hat{\Pi}_{\Sigma_d}$ projects onto the energy shell. The formal statement connecting eigenstate thermalization to thermalization is then (depicted in Fig. 3):

**Proposition 4.1** (Quantum eigenstate thermalization implies thermalization o.a. given a nondegenerate spectrum [10, 15]). *If*

    *1.* $\hat{\Pi}_A$ *satisfies eigenstate thermalization to accuracy $\epsilon$ within $\Sigma_d$, i.e., for some small $\epsilon > 0$,*

$$\left| \langle E_n | \hat{\Pi}_A | E_n \rangle - \langle \hat{\Pi}_A \rangle_{\Sigma_d} \right| < \epsilon, \quad for\ all\ n : |E_n\rangle \in \Sigma_d, \tag{49}$$

    *and*

    *2. the energy levels $E_n$ within $\Sigma_d$ are nondegenerate ($E_n \neq E_m$ for $n \neq m$),*

*then $\hat{\Pi}_A$ thermalizes o.a. to $\langle \hat{\Pi}_A \rangle_{\Sigma_d}$, to accuracy $\epsilon$, for any initial density operator $\hat{\rho} : \Sigma_d \to \Sigma_d$ in $\Sigma_d$:*

$$\left| \int \overline{dt}\ \mathrm{Tr}[\hat{\rho}(t)\hat{\Pi}_A] - \langle \hat{\Pi}_A \rangle_{\Sigma_d} \right| < \epsilon. \tag{50}$$

*Proof.* Though well known [10, 15, 17–19], we briefly review the proof in App. D.2.1, as we will use similar arguments on other occasions in this paper. It is based on the following relation: If the energy levels $E_n$ within $\Sigma_d$ are nondegenerate, then for any observable $\hat{\Pi}_A : \mathcal{H} \to \mathcal{H}$ in the full system, and any density operator $\hat{\rho} : \Sigma_d \to \Sigma_d$ in the energy shell,

$$\int \overline{dt}\ \mathrm{Tr}[\hat{\rho}(t)\hat{\Pi}_A] = \sum_{n:|E_n\rangle \in \Sigma_d} \langle E_n | \hat{\rho} | E_n \rangle \langle E_n | \hat{\Pi}_A | E_n \rangle. \tag{51}$$

$\square$

While to all appearances, Proposition 4.1 is the direct quantization of Proposition 3.1 (classical eigenstate thermalization implies thermalization o.a.), there are two subtle difficulties with this statement:

    1. The nondegeneracy condition on the energy levels, while typical for most systems, is usually difficult to impose or verify with finite resources. Such fine details of the spectrum tend to be inconsequential for most dynamics at accessible time scales. No similar constraint on the energy levels (values of energy on the $\mathcal{P}_k$) occurs in the classical proposition 3.1.

    2. Correspondingly, the time average $\int \overline{dt} = \lim_{T\to\infty}$ in Eq. (50) (via Eq. (51)) explicitly requires taking the limit over timescales larger than the smallest nearest neighbor spacings, $T \gg 2\pi/|\min(E_{n+1} - E_n)|$. This approaches $T \sim D^2$ and becomes inaccessible in the thermodynamic limit — for instance, Eq. (51) would not hold if one takes $D \to \infty$ first and only then takes $T \to \infty$. On the other hand, the $T \to \infty$ limit in the classical statement is through accessible $O(1)$ timescales, even in the limit of an infinite number of degrees of freedom.

## 4.2 Eigenspace thermalization $\implies$ idealized thermalization

Let us now consider how to modify Proposition 4.1 to avoid these issues with accessibility. We initially focus on a single potentially degenerate energy level $E$, with eigenspace $\mathcal{H}(E)$ spanned

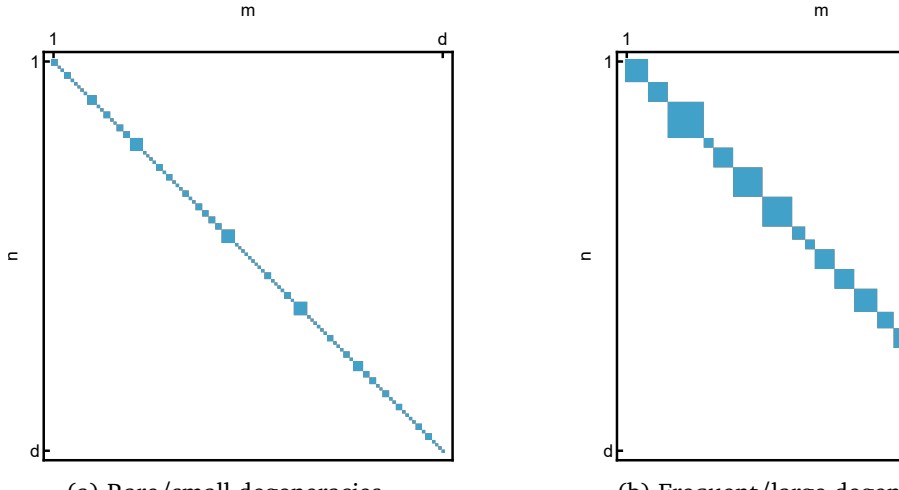

(a) Rare/small degeneracies.                    (b) Frequent/large degeneracies.

Figure 4: Schematic depiction of eigenspace thermalization in terms of the relevant matrix elements between pairs $(n, m)$ of the energy levels $(E_n, E_m)$; essentially, only the $E_n = E_m$ pairs are relevant. This matrix element structure can appear different between a spectrum with rare/small degeneracies (4a) and with frequent/large degeneracies (4b), and therefore more complicated than eigenstate thermalization. However, in terms of energy differences it corresponds to the extremely natural $\Delta E = 0$ independent of the spectral details, and can imply thermalization without any knowledge of the spectrum. Even in this case, the set of pairs with $\Delta E = 0$ is still not accessible, and as we will argue in Sec. 5.2, eigenspace thermalization is not necessary for accessible thermalization especially in highly degenerate systems.

by one or more orthonormal eigenstates[12] $|E_n\rangle$ with $E_n = E$; also let $\hat{\Pi}(E)$ be the projector onto $\mathcal{H}(E)$, which can be written as

$$\hat{\Pi}(E) = \sum_{n:E_n=E} |E_n\rangle\langle E_n|. \tag{52}$$

Every single state in $\mathcal{H}(E)$ is an eigenstate of the Hamiltonian with eigenvalue $E$. Correspondingly, there is no dynamics within $\mathcal{H}(E)$, up to a trivial overall phase that does not change the state:

$$|\psi(t)\rangle = e^{-iEt}|\psi\rangle, \quad \text{for all } |\psi\rangle \in \mathcal{H}(E). \tag{53}$$

It follows that every such state $|\psi\rangle$ is a separate ergodic subset of $\mathcal{H}(E)$.

Classically, we had required eigenstate thermalization to refer to thermalization in every ergodic subset of the phase space of interest. Appealing to the classical case for intuition, a natural way to impose eigenstate thermalization in a degenerate subspace is then to enforce thermalization for every ergodic subset (equivalently, eigenstate of the Hamiltonian), which in this case is every state $|\psi\rangle \in \mathcal{H}(E)$. To be concrete, for an observable $\hat{\Pi}_A$ with thermal value $\langle\hat{\Pi}_A\rangle_{\Sigma_d}$ (in some suitable energy shell $\Sigma_d$ which may be as small as just $\mathcal{H}(E)$), we require

$$\left|\langle\psi|\hat{\Pi}_A|\psi\rangle - \langle\hat{\Pi}_A\rangle_{\Sigma_d}\right| < \epsilon, \tag{54}$$

for some small $\epsilon$. Such a relation has been used to define thermalization in eigenspaces in, e.g., Ref. [41].

---

[12]When we use notation such as $|E_n\rangle$ for eigenstates, we will always assume that this labels an eigenstate within a complete orthonormal basis, as opposed to considering all available eigenstates in degenerate subspaces.

906  While the $|\psi\rangle$ are dynamically independent, they are not all linearly independent; it is
907 therefore desirable to re-express Eq. (54) in terms of a simpler, more tractable criterion in
908 $\mathcal{H}(E)$. For this purpose, we note that the Cauchy-Schwarz inequality (applied to the trace inner
909 product $\left|\text{Tr}[\hat{P}^\dagger\hat{Q}]\right| \leq \sqrt{\text{Tr}[\hat{P}^\dagger\hat{P}]\,\text{Tr}[\hat{Q}^\dagger\hat{Q}]}$ for $\hat{P} = |\psi\rangle\langle\psi|$ and $\hat{Q} = \hat{\Pi}(E)\left(\hat{\Pi}_A - \langle\hat{\Pi}_A\rangle_{\Sigma_d}\hat{\mathbb{1}}\right)\hat{\Pi}(E))$
910 implies:

$$\left|\langle\psi|\hat{\Pi}_A|\psi\rangle - \langle\hat{\Pi}_A\rangle_{\Sigma_d}\right| \leq \sqrt{\text{Tr}\left[\left\{\left(\hat{\Pi}_A - \langle\hat{\Pi}_A\rangle_{\Sigma_d}\hat{\mathbb{1}}\right)\hat{\Pi}(E)\right\}^2\right]}. \tag{55}$$

911 The right-hand side is a tractable, invariant measure of distance between $\hat{\Pi}_A$ and its thermal
912 value $\langle\hat{\Pi}_A\rangle_{\Sigma_d}\hat{\mathbb{1}}$ within $\mathcal{H}(E)$. It is therefore convenient to use this measure to *define* eigenspace
913 thermalization in degenerate subspaces, now focusing on energy shells with multiple energy
914 levels:

915 **Definition 4.2** (Eigenspace thermalization)**.** *Let E denote the different eigenvalues of a Hamil-*
916 *tonian, each with its eigenspace $\mathcal{H}(E)$ that may or may not be degenerate ($\dim\mathcal{H}(E) \geq 1$). We*
917 *say that an observable $\hat{\Pi}_A$ shows eigenspace thermalization to accuracy $\epsilon > 0$ in an energy shell*
918 $\Sigma_d = \bigcup_{E\in\mathcal{E}}\mathcal{H}(E)$ *over a set of energy eigenvalues $\mathcal{E}$ if:*

$$\text{Tr}\left[\left\{\left(\hat{\Pi}_A - \langle\hat{\Pi}_A\rangle_{\Sigma_d}\hat{\mathbb{1}}\right)\hat{\Pi}(E)\right\}^2\right] \equiv \sum_{\substack{n:\\E_n=E}}\sum_{\substack{m:\\E_m=E}}\left|\langle E_n|\hat{\Pi}_A|E_m\rangle - \langle\hat{\Pi}_A\rangle_{\Sigma_d}\delta_{nm}\right|^2 < \epsilon^2 \quad \text{for all } \mathcal{H}(E) \subseteq \Sigma_d.$$
$$\tag{56}$$

919  When there are no degeneracies, i.e., $\dim\mathcal{H}(E) = 1$, Eq. (56) reduces to the conventional
920 statement of eigenstate thermalization, Eq. (49); it also implies Eq. (54) in the presence of
921 degeneracies. As depicted in Fig. 4, the mild difference between Eq. (56) and conventional
922 eigenstate thermalization is the implied constraint on *off-diagonal* matrix elements within
923 degenerate subspaces[13]:

$$\sum_{\substack{n:\\E_n=E}}\sum_{\substack{m:m\neq n\\E_m=E}}\left|\langle E_n|\hat{\Pi}_A|E_m\rangle - \langle\hat{\Pi}_A\rangle_{\Sigma_d}\delta_{nm}\right|^2 < \epsilon^2. \tag{57}$$

924  Now, let us see how this more specific form of eigenstate thermalization implies thermaliza-
925 tion, by generalizing Proposition 4.1:

926 **Proposition 4.3** (Quantum eigenspace thermalization implies thermalization o.a.)**.** *If $\hat{\Pi}_A$ satis-*
927 *fies eigenspace thermalization to accuracy $\epsilon$, i.e., Eq. (56), within an energy shell $\Sigma_d = \bigcup_{E\in\mathcal{E}}\mathcal{H}(E)$,*
928 *then $\hat{\Pi}_A$ thermalizes o.a. to $\langle\hat{\Pi}_A\rangle_{\Sigma_d}$, to accuracy $\epsilon$, for any initial density operator $\hat{\rho} : \Sigma_d \to \Sigma_d$ in*
929 $\Sigma_d$:

$$\left|\int\overline{\mathrm{d}t}\,\text{Tr}[\hat{\rho}(t)\hat{\Pi}_A] - \langle\hat{\Pi}_A\rangle_{\Sigma_d}\right| < \epsilon. \tag{58}$$

930 *Proof.* In the presence of degeneracies, given a complete orthonormal basis $\{|E_n\rangle\}_{n=0}^{D-1}$ of energy
931 eigenstates, the analogue of Eq. (51) for the infinite time average is:

$$\int\overline{\mathrm{d}t}\,\text{Tr}[\hat{\rho}(t)\hat{\Pi}_A] = \sum_{E\in\mathcal{E}}\left[\sum_{|E_n\rangle,|E_m\rangle\in\mathcal{H}(E)}\langle E_n|\hat{\rho}|E_m\rangle\langle E_m|\hat{\Pi}_A|E_n\rangle\right]. \tag{59}$$

---

[13]Even with $\epsilon = \Theta(1)$, such a constraint can at times be more stringent than the full statement of the eigenstate
thermalization hypothesis (ETH), in which the off-diagonal matrix elements of operators are suppressed by factor
$e^{-S(E)/2} \sim 1/\sqrt{d}$ in suitable energy shells. In conventional ETH, the left hand side of Eq. (56) may be as large
as $(\dim\mathcal{H}(E))^2 e^{-S(E)} = \Theta(d)$ (when $\dim\mathcal{H}(E) = \Theta(d)$), rather than being bounded by some small $\epsilon^2$. For our
purposes, it is also useful to make a distinction between the conventional statement of ETH and Definition 4.2
to carefully keep track of which variants of eigenstate thermalization contribute to which dynamical processes of
thermalization.

932  Applying the Cauchy-Schwarz inequality to Eq. (59) in different ways (together with the triangle
933  inequality $|x + y| \leq |x| + |y|$) gives Eq. (58), as described in App. D.2.2.                             $\square$

934      A key consequence of Proposition 4.3 is that we are able to describe the phenomenon of
935  thermalization without direct reference to the energy eigenvalues themselves, in particular
936  without even knowing the degree of degeneracy of each eigenvalue. The only input we need is
937  that the observable $\hat{\Pi}_A$ satisfies eigenspace thermalization — Eq. (56) — in every *eigenspace*
938  $\mathcal{H}(E)$, given only that they exist in some energy shell of interest, irrespective of the number or
939  nature of these eigenspaces.

## 4.3  Preview: Eigenspace thermalization and "approximate" degeneracies

941  Now, let us briefly consider thermalization o.a. over finite times, as a qualitative preview of
942  the more rigorous results to follow. At an intuitive level, we can tackle this case by appealing
943  to the notion of eigenspace thermalization — not just for degenerate subspaces, but even for
944  distinct energy levels in a small range of energies— as follows. At any time $T$, an initial density
945  operator $\hat{\rho}$ evolves into

$$\hat{\rho}(T) = \sum_{n,m} e^{-i(E_n - E_m)T} |E_n\rangle\langle E_n| \hat{\rho} |E_m\rangle\langle E_m|. \tag{60}$$

946  For the purposes of dynamics at times $|t| < T$, one can regard all sets of energy levels satisfying

$$|E_n - E_m| \ll \frac{2\pi}{T} \tag{61}$$

947  as approximately degenerate, as the corresponding phase factors in Eq. (60) are $e^{-i(E_n - E_m)t} \approx 1$.
948  By intuitively extending Proposition 4.3, we should expect that an observable $\hat{\Pi}_A$ thermalizes
949  o.a. to $\langle\hat{\Pi}_A\rangle$ in any state $\hat{\rho}$ (note that we have now assumed a single thermal value $\langle\hat{\Pi}_A\rangle$ over
950  the entire Hilbert space) provided it satisfies eigenspace thermalization in such approximately
951  degenerate blocks:

$$\sum_{\substack{n,m: \\ (E_n - E_m) \ll 2\pi/T}} \left| \langle E_n| \left( \hat{\Pi}_A - \langle\hat{\Pi}_A\rangle\hat{\mathbb{1}} \right) |E_m\rangle \right|^2 \approx 0. \tag{62}$$

952  We will develop this argument rigorously in the following section, but we note for now that this
953  intuition is most directly captured by Eq. (12) in Sec. 2 or Proposition 5.4 at a more technical
954  level.

# 5  Energy-band thermalization for accessible timescales

956  We have noted that eigenspace thermalization guarantees thermalization o.a. over infinitely
957  long timescales even in a degenerate spectrum. In this section, we are concerned with adapting
958  this notion to practically accessible time scales, in the form of energy-band thermalization. We
959  will approach this by first defining a weighted time average, which will underlie most of our
960  subsequent results. We will show that such weighted time averages thermalize given that an
961  observable satisfies energy-band thermalization. While we will motivate our definition of this
962  property to some extent in this section (in addition to the qualitative arguments in Sec. 4.3),
963  its ultimate justification (and utility) lies in the fact that it is the property that can be most
964  directly accessed via the dynamics of a single state (Secs. 6 and 7), and therefore directly lends
965  itself to experimental measurements (Sec. 9).

966      Much of our discussion in this section will rely on an energy bandwidth $\Delta E > 0$, such
967  that we are only interested in pairs of energy levels $(E_n, E_m)$ satisfying $|E_n - E_m| < \Delta E$. We

emphasize that the bandwidth $\Delta E$ itself is not a property of the energy spectrum, but may be chosen by hand (such as by an experimenter) without any knowledge of the spectrum (e.g., in a fully degenerate spectrum within the energy shell $\Sigma_d$, any choice of $\Delta E > 0$ would allow all pairs of energy levels, while a non-degenerate spectrum will only allow a restricted set of pairs, and we do not need to know which of these is the case for a given system). Our results, which do not rely on whether any given pair of energy levels belong to this bandwidth, therefore retain their independence from the energy eigenvalues $E_n$.

## 5.1 Weighted time averages

At a technical level, it will be convenient to work with weighted time averages of some observable $\hat{\Pi}_A$ in some state $\hat{\rho}$:

$$\int \mathrm{d}t\, w(t)\,\mathrm{Tr}[\hat{\rho}(t)\hat{\Pi}_A] = \sum_{n,m} \widetilde{w}(E_n - E_m)\langle E_n|\hat{\rho}|E_m\rangle\langle E_m|\hat{\Pi}_A|E_n\rangle. \tag{63}$$

Here,

$$\widetilde{w}(\delta E) \equiv \int \mathrm{d}t\, w(t)e^{-i\delta E t}, \tag{64}$$

is the Fourier transform of $w(t)$. With different choices of the weight $w(t)$, we can get different time-domain quantities of interest: for example, $w(t) = 1/T$ in $t \in [0, T]$ or $w(t) = 1/(2T)$ in $t \in [-T, T]$ gives the conventional time average (which can be shifted to, e.g., $[t_0 - T, t_0 + T]$ for a time average centered around $t_0$), while $w(t) = \delta(t - t_0)$ gives the instantaneous expectation value of $A$ at $t = t_0$. In all these cases, we will assume that the time integral of $w(t)$ is normalized to

$$\widetilde{w}(0) = \int \mathrm{d}t\, w(t) = 1. \tag{65}$$

The reason behind considering more general $w(t)$ than the conventional time average is that, as we will see in Sec. 6, the conventional time average does not allow rigorous results of the kind we are seeking due to interference effects. While these effects are not an obstacle in the present section, we introduce $w(t)$ here for greater generality.

We will also take $w(t) \geq 0$. Among other things, through a relation between non-negativity conditions and Fourier transforms [75,76], this implies that

$$|\widetilde{w}(\delta E)| \leq |\widetilde{w}(0)| = 1. \tag{66}$$

Finally, we need to formalize the notion of when $w(t)$ represents a "long-time average", relative to the timescale set by the bandwidth $\Delta E$ For this, we want $w(t)$ to have appreciable magnitude at least over some long time interval $T \sim 2\pi/\Delta E$, such that $\widetilde{w}(\delta E)$ is appreciable only within $|\delta E| < \Delta E$. Specifically, for some small cutoff $0 < w_0 < 1$, we require that $\widetilde{w}(\delta E)$ satisfies:

$$|\widetilde{w}(\delta E)| \leq w_0, \quad \text{for all } \delta E \geq \Delta E. \tag{67}$$

We emphasize that such a condition may be guaranteed merely by our choice of $\Delta E$ and function $w(t)$, without any knowledge of the energy spectrum of the system (we do not require, for example, that any energy level spacings with $\delta E \geq \Delta E$ even exist in the system).

## 5.2 Thermalization in energy bands

For $w(t)$ satisfying Eq. (67), we should expect the thermal behavior of weighted time averages to be determined by the full structure of $\hat{\Pi}_A$ within the bandwidth $\Delta E$. Using the intuition

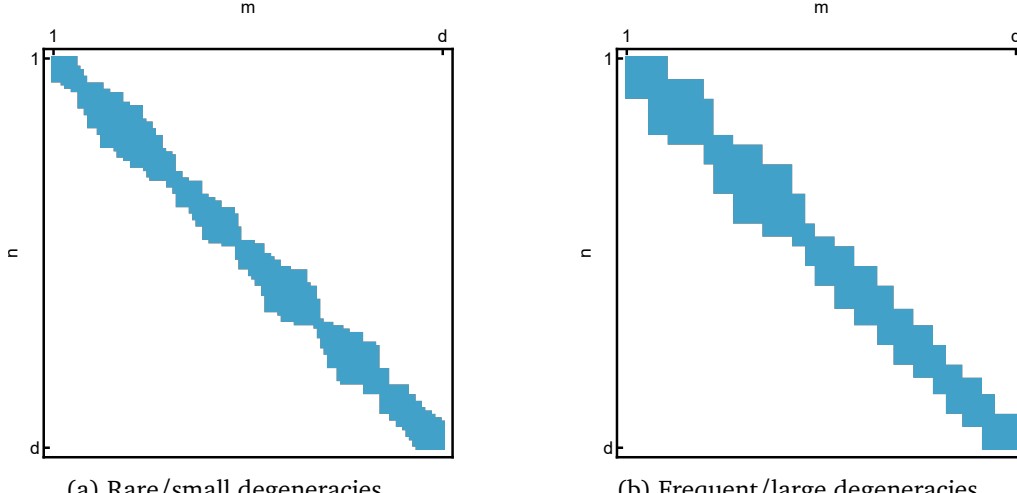

(a) Rare/small degeneracies.                         (b) Frequent/large degeneracies.

Figure 5: Schematic depiction of energy-band thermalization in terms of the relevant matrix elements between pairs $(n, m)$ of the energy levels $(E_n, E_m)$. While the matrix element structure can appear to be very erratic and strongly dependent on spectral fluctuations in the differences $E_n - E_m$, such as between spectra with rare (5a) as opposed to very frequent (5b) degeneracies, that the energy band is defined naturally in terms of the physically relevant energies $|E_n - E_m| < \Delta E$ makes this the most accessible criterion for thermalization compared to eigenstate or eigenspace thermalization. These matrix elements are to be contrasted with Fig. 1 [respectively, (5a with 1a) and (5b with 1b)] which depicts how energy-band thermalization is independent of the details of the spectrum.

of "approximate eigenspace thermalization" developed in Sec. 4.3, we should expect that the appropriate condition on $\hat{\Pi}_A$ is that

$$\text{Tr}\left[ \left\{ \left( \hat{\Pi}_A - \langle \hat{\Pi}_A \rangle_{\Sigma_d} \hat{\mathbb{1}} \right) \hat{\Pi}(E_0, \Delta E) \right\}^2 \right] \tag{68}$$

be small, where $\hat{\Pi}(E_0, \Delta E)$ projects onto an "approximate eigenspace" i.e. energy window of width $\Delta E$ centered at $E_0$ within an energy shell $\Sigma_d$.

However, merely dividing the spectrum into such separate energy windows neglects interference between the different energy windows, and is not technically sound for the following reason: we also want to constrain the matrix elements of $\hat{\Pi}_A$ that connect energies within a separation $\Delta E$ that may belong to different (consecutive) energy windows. Any specific choice of "approximate eigenspaces" would necessarily assign some sets of levels closer than $\Delta E$ to different approximate eigenspaces. One solution is to take an "averaged" variant over all possible choices of these energy windows in the energy shell $\Sigma_d$, in other words averaging over $E_0$, effectively obtaining an "energy-band selector" of bandwidth $\Delta E$, over a basis of eigenstates $|E_n\rangle$:

$$\sum_{n:E_n \in \Sigma_d} |E_n\rangle\langle E_n| \otimes \left[ \sum_{\substack{m:E_m \in \Sigma_d \\ |E_n - E_m| < \Delta E}} |E_m\rangle\langle E_m| \right], \tag{69}$$

into which two operators may be inserted by respectively connecting $|E_m\rangle$ to $\langle E_n|$ and $\langle E_m|$ to $|E_n\rangle$ in a trace. This allows us to define operators restricted to an energy band as in Eq. (7), in

which we now formally include the energy shell:

$$[\hat{A}]_{\Sigma_d, \Delta E} \equiv \sum_{\substack{n,m: \\ |E_n\rangle, |E_m\rangle \in \Sigma_d \\ |E_n - E_m| < \Delta E}} |E_n\rangle\langle E_n|\hat{A}|E_m\rangle\langle E_m|. \tag{70}$$

We emphasize again that these sums are to be taken over any complete orthonormal basis of eigenstates $|E_n\rangle$ within $\Sigma_d$, and not all the eigenstates that may constitute degenerate subspaces. The relevant matrix elements are pictorially depicted in Fig. 5.

We can now define "energy-band thermalization"[14] by analogy with eigenspace thermalization [Definition 4.2], with the pair of eigenspace projectors $\hat{\Pi}(E)$ replaced by the energy-band selector in Eq. (69):

**Definition 5.1** (Energy-band thermalization). *Given $\Delta E > 0$, we say that an observable $\hat{\Pi}_A$ shows energy-band thermalization with bandwidth $\Delta E$ in an energy shell $\Sigma_d$, to accuracy $\epsilon$, if*

$$\frac{1}{d}\operatorname{Tr}\left\{\left[\hat{\Pi}_A - \langle\hat{\Pi}_A\rangle_{\Sigma_d}\hat{\mathbb{1}}\right]^2_{(\Sigma_d, \Delta E)}\right\} \equiv \frac{1}{d}\sum_{n: E_n \in \Sigma_d}\sum_{\substack{m: E_m \in \Sigma_d \\ |E_n - E_m| < \Delta E}}\left|\langle E_n|\hat{\Pi}_A|E_m\rangle - \langle\hat{\Pi}_A\rangle_{\Sigma_d}\delta_{nm}\right|^2 < \epsilon^2. \tag{71}$$

The normalization by $d = \dim\Sigma_d$ on the left hand side of Eq. (71) (which is essentially a rescaling of the accuracy parameter $\epsilon$ relative to eigenspace thermalization) will prove convenient later; intuitively, one can think of the normalization as representing the aforementioned average over all choices of energy windows in the $d$-dimensional energy shell, weighted proportional to their dimension. More importantly, at a physical level, we have chosen this normalization because we typically expect $\epsilon = O(1)$ (meaning $\epsilon$ is at most a finite number in the $d \to \infty$ limit) with this choice (whose justification will become apparent in Sec. 6.2, but as a quick justification, we note that $\operatorname{Tr}[\hat{A}]/d = O(1)$ when $\hat{A}$ has eigenvalues of $O(1)$ magnitude, and therefore should be "accessible").

Before examining its implications for accessible time scales, let us consider the connection between energy-band thermalization and eigenspace thermalization. Eigenspaces are characterized by an energy difference of $\delta E = 0$, and are therefore completely contained in the energy band: Eq. (71) includes all the terms relevant for eigenspace thermalization in $\Sigma_d$, with the only complication being the $1/d$ normalization. This normalization suggests that given energy-band thermalization to accuracy $\epsilon$:

1. Every eigenspace $\mathcal{H}(E) \subseteq \Sigma_d$ thermalizes to accuracy $\epsilon\sqrt{d}$:

$$\operatorname{Tr}\left[\left\{\left(\hat{\Pi}_A - \langle\hat{\Pi}_A\rangle_{\Sigma_d}\hat{\mathbb{1}}\right)\hat{\Pi}(E)\right\}^2\right] < \epsilon^2 d, \tag{72}$$

   as all the other terms (i.e., contributions not solely from $\mathcal{H}(E)$) in Eq. (71) are non-negative.

2. *On average* (with each eigenspace weighted according to its dimension), the eigenspaces within $\Sigma_d$ thermalize to the same accuracy $\epsilon$:

$$\frac{1}{d}\sum_{\mathcal{H}(E) \subseteq \Sigma_d}\operatorname{Tr}\left[\left\{\left(\hat{\Pi}_A - \langle\hat{\Pi}_A\rangle_{\Sigma_d}\hat{\mathbb{1}}\right)\hat{\Pi}(E)\right\}^2\right] < \epsilon^2, \tag{73}$$

   again by dropping the nonnegative contributions that connect distinct energies $(E_n - E_m) \neq 0$.

---

[14]We note that this structure also occurs if one combines semiclassical results [40,44] on diagonal and off-diagonal matrix elements connected to classically ergodic systems, and formally removes the $\Delta E \to 0$ limit.

Eq. (72) suffices to conclude eigenspace thermalization if $\epsilon \ll 1/\sqrt{d}$, but for larger $\epsilon$, we will need to work with Eq. (73). As we discuss in App. C, Eq. (73) can constrain the "size" of the subspace in $\Sigma_d$ in which all eigenspaces violate eigenspace thermalization to accuracy $\lambda$, as measured by the total number $n_\lambda$ of eigenspaces $\mathcal{H}(E)$ that intersect this subspace:

$$n_\lambda < \frac{\epsilon^2}{\lambda^2} d. \tag{74}$$

However, for this to be nontrivial with "accessible" values of $\epsilon$, the number of distinct energy levels in the energy shell should be close to its dimension, $n \gg \epsilon^2 d/\lambda^2$, so that $n_\lambda \ll n$ by the above constraint. In that case, we can claim "weak eigenspace thermalization" ($n_\lambda/n \ll 1$), in analogy with the notion of "weak ETH" [18, 34, 35] where almost all (but not necessarily all) eigenstates thermalize; more discussion on the physical relevance of such "weak" notions will follow in the Conclusion, Sec. 10.

However, there is a complication: the number of eigenspaces is guaranteed to be accessible in an experiment only if $n$ itself is accessibly small[15], such as $n < cN^\alpha$ for an $N$-particle system, which requires an unusually high level of degeneracy (close to $\Theta(d)$) in most energy levels. It is therefore difficult to establish that a system has the requisite high number of eigenspaces in most cases.

In connecting energy-band thermalization to eigenspace thermalization, we seem to have arrived at a tradeoff:

1. If one wants to keep to "accessible" values of $\epsilon$, one can only show weak eigenspace thermalization, that too in a (formally) restricted class of systems where the number of levels still scales as nearly $d$ (i.e. where a high degree of degeneracy is not typical), though this may cover several cases of interest (e.g., such as systems with a time reversal symmetry showing Kramers' twofold degeneracy [68], but otherwise assumed to be nondegenerate).

2. To fully constrain eigenspace thermalization, one must work with "inaccessibly" small values of $\epsilon$, but this works for any Hamiltonian system.

To remedy this situation, we must bypass eigenspace thermalization (and therefore, eigenstate thermalization) completely; this is what we turn to next. We will see that energy-band thermalization can directly constrain thermalization dynamics in any given basis of physical states, with accessible values of $\epsilon$ and without any dependence on the energy spectrum. More significantly, even in systems known to be nondegenerate, it allows us to establish thermalization over finite timescales.

## 5.3    Energy-band thermalization $\implies$ almost all physical states thermalize

To see the impact of energy-band thermalization on thermalization dynamics in an initial state $\hat{\rho}$ for a long-time average $w(t)$ satisfying Eq. (67), i.e., with $\widetilde{w}(\delta E)$ being appreciable only inside the bandwidth $\Delta E$, we turn to the expression in Eq. (63) that expresses this time average in the energy eigenbasis. From this expression, we obtain:

**Theorem 5.2** (Energy-band thermalization implies thermalization o.a. over accessible timescales)**.**
*If $\hat{\Pi}_A$ satisfies energy-band thermalization with bandwidth $\Delta E > 0$ and accuracy $\epsilon > 0$ within an energy shell $\Sigma_d$ [Eq. (71)], and one considers time-averaging with a weighting function $w(t)$ with $w_0$-bandwidth smaller than $\Delta E$, i.e. that satisfies [Eq. (67)]:*

$$|\widetilde{w}(\delta E)| \leq w_0, \quad \text{for all } \delta E \geq \Delta E,$$

---

[15]This is because in a system with $n$ eigenspaces, the spectral form factor [68] (see also Sec. 9), used to measure energy level statistics [22, 23], may oscillate at late times around values as small as $1/n$.

then $\hat{\Pi}_A$ *thermalizes o.a. to* $\langle \hat{\Pi}_A \rangle_{\Sigma_d}$ *in any initial state* $\hat{\rho} : \Sigma_d \to \Sigma_d$ *in the energy shell, with accuracy:*

$$\left| \int dt \, w(t) \, \text{Tr}[\hat{\rho}(t)\hat{\Pi}_A] - \langle \hat{\Pi}_A \rangle_{\Sigma_d} \right| < \left( \epsilon + w_0 \sqrt{\langle \hat{\Pi}_A \rangle_{\Sigma_d}} \right) \sqrt{d \, \text{Tr}[\hat{\rho}^2]}. \tag{75}$$

*Proof.* The proof is similar to that of Proposition 4.3, except that one now separates the contributions within the bandwidth $\Delta E$ from those outside, and applies the triangle and Cauchy-Schwarz inequalities separately to each set of contributions. Each contribution now has an explicit $\hat{\rho}$ dependence, necessitated by our normalization of energy band thermalization (this would have also been relevant for eigenspace thermalization in Proposition 4.3 if we had chosen a different normalization); the outside-$\Delta E$ contribution has a dependence on $w_0$ in addition to $\hat{\rho}$. See App. D.3.2 for details.                                                    $\square$

In Eq. (75), we finally observe the quantitative effect of choosing accessible timescales via the $w_0$ term. But that is not all: The $\sqrt{d \, \text{Tr}[\hat{\rho}^2]}$ factor is quite nontrivial, and will ensure that we cannot in general show the thermalization of *every* physical state, but only for almost all physical states.

To see this, first let us consider a pure initial state $\hat{\rho}$, with $\text{Tr}[\hat{\rho}^2] = 1$. For energy-band thermalization to imply thermalization o.a. to accuracy $\lambda > 0$,

$$\left| \int dt \, w(t) \, \text{Tr}[\hat{\rho}(t)\hat{\Pi}_A] - \langle \hat{\Pi}_A \rangle_{\Sigma_d} \right| < \lambda, \tag{76}$$

in such a pure state, one must satisfy both

$$\epsilon < \frac{\lambda}{\sqrt{d}} \quad \text{AND} \quad w_0 < \frac{\lambda}{\sqrt{d \langle \hat{\Pi}_A \rangle_{\Sigma_d}}}. \tag{77}$$

In general, $\langle \hat{\Pi}_A \rangle_{\Sigma_d} < 1$ (though still some accessible value by assumption, i.e. we assume that thermalization to this value can be detected), so the condition on $w_0$ is somewhat less stringent than the one on $\epsilon$. But for thermalization o.a. in an individual pure state, we seem to require an inaccessibly small accuracy $\epsilon$ as well as an inaccessibly long time average in $w(t)$. We will see, however, that we can still constrain pure state thermalization with accessible resources if we allow a small fraction of pure states to fail to thermalize.

Now, consider mixed states, for which

$$\text{Tr}[\hat{\rho}^2] = \frac{1}{\mu(\rho)d}, \tag{78}$$

for some parameter $\mu(\rho) \leq 1$. For such states, it is sufficient that

$$\epsilon < \lambda \sqrt{\mu(\rho)} \quad \text{AND} \quad w_0 < \lambda \sqrt{\frac{\mu(\rho)}{\langle \hat{\Pi}_A \rangle_{\Sigma_d}}} \tag{79}$$

for energy-band thermalization to guarantee thermalization with accuracy $\lambda$, provided that $\mu(\rho)$ is "accessible". This is in fact anticipated by classical mechanics: such states are precisely the quantum analogue of the "bulky" states in Sec. 3. This analogy is quite strong[16]: if a classical limit exists, the parameter $\mu(\rho)$ can be directly identified with the phase space volume

---

[16]We have sometimes been tempted to refer to this (old and well established) semiclassical relation as "Entanglement = Phase space volume", with the qualifier that entanglement here means the exponential of the second Rényi entropy, following a set of conjectures in quantum gravity [77] relating quantum information measures to more geometric (semi-)classical ones.

of a classical region in which (say) $\rho$ has uniform support (if it is associated with such a classical state) [72, 73].

We also recall that classically, ergodicity in initial states of nonzero measure $\mu(\rho) > 0$ is interpreted as ergodicity "almost everywhere" in phase space. To formulate an analogous statement in quantum mechanics, let us consider any complete orthonormal basis for the energy shell $\Sigma_d$:

$$\mathcal{B} = \{|k\rangle\}_{k=0}^{d-1} \quad \subset \Sigma_d. \tag{80}$$

This basis will be taken to represent a collection of "physical states" in the system. For example, in the full Hilbert space ($\Sigma_d \to \mathcal{H}$) of an $N$ qubit system, $\mathcal{B}$ could be the set of computational basis states:

$$\mathcal{B} = \left\{ |s_1, \dots, s_N\rangle : s_j \in \{0, 1\} \right\}, \tag{81}$$

or bases generated by any accessible set of simple unitary gates applied to these states.

For the dynamics of each of these basis states, with respective initial density operators given by:

$$\hat{\rho}_k(0) = |k\rangle\langle k|, \tag{82}$$

we consider the deviation of the weighted time average of $\hat{\Pi}_A$ from its thermal value in the energy shell:

$$\lambda_k[w] \equiv \int \mathrm{d}t \, w(t) \, \mathrm{Tr}[\hat{\rho}_k(t)\hat{\Pi}_A] - \langle \hat{\Pi}_A \rangle_{\Sigma_d}. \tag{83}$$

Energy-band thermalization constrains these deviations as follows:

**Theorem 5.3** (Energy-band thermalization implies almost all physical basis states thermalize o.a.)**.** *Let $\hat{\Pi}_A$ satisfy energy-band thermalization with bandwidth $\Delta E > 0$ and accuracy $\epsilon > 0$ within an energy shell $\Sigma_d$ [Eq. (71)], and consider time-averaging with a weighting function $w(t)$ with $w_0$-bandwidth smaller than $\Delta E$, i.e. that satisfies [Eq. (67)]:*

$$|\widetilde{w}(\delta E)| \leq w_0, \quad \textit{for all } \delta E \geq \Delta E.$$

*Then, for any complete orthonormal basis of pure states $\mathcal{B}$ within the energy shell $\Sigma_d$, the fraction $f_\lambda[\mathcal{B}]$ of states that fail to thermalize o.a. to accuracy $\lambda$, i.e. [with $\overline{\Theta}(x) = 1$ if $x$ is true and $0$ otherwise]*

$$f_\lambda[\mathcal{B}] \equiv \frac{1}{d} \sum_{|k\rangle \in \mathcal{B}} \overline{\Theta}\big(|\lambda_k[w]| \geq \lambda\big) \tag{84}$$

*is constrained by:*

$$f_\lambda[\mathcal{B}] < \frac{\sqrt{2}}{\lambda}\left(\epsilon + w_0\sqrt{\langle \hat{\Pi}_A \rangle_{\Sigma_d}}\right). \tag{85}$$

*Proof.* This can be proved by a quantitative version, described in App. D.3.3, of the following argument that the fraction is small, $f_\lambda[\mathcal{B}] \ll 1$. The ensemble of basis states with $\lambda_k[w] \geq 0$ and the ensemble of basis states with $\lambda_k[w] < 0$ must together contain all $d$ basis states, so each of them contains either $\Theta(d)$ states or a vanishing fraction of states in $\mathcal{B}$. If the former is true for one ensemble, the corresponding ensemble is a "bulky" state and almost all constituent pure states must thermalize o.a., as the deviation of $\hat{\Pi}_A$ from its thermal value in the bulky state averages its deviations in the pure states with the same sign. If the latter is true, then the ensemble is a vanishing fraction of states in any case, and may be included in $f_\lambda[\mathcal{B}]$ in a negligible overestimation. $\qquad\square$

As long as $\epsilon, w_0 \ll \lambda$, the fraction of *basis* states with non-thermal time averages to accuracy $\lambda$ (weighted by a given $w(t)$) is vanishingly small. But we cannot force $f_\lambda[\mathcal{B}] < 1/d$ (which would ensure that all states thermalize) with accessible values of $\epsilon$ and $w_0$: there appears to be

no accessible way to rule out a small fraction of non-thermal *physical* states in any accessible time average. These are a quantum analogue of measure-zero exceptions, such as periodic orbits in a classical system showing measure-theoretic ergodicity. As noted in Eq. (77), it is possible to rule out these features with "inaccessible" values of $\epsilon$ and $w_0$ by directly considering the thermalization of pure states given energy-band thermalization, provided that such small values of $\epsilon$ are allowed for $\hat{\Pi}_A$ by the system.

Another interesting question is whether the same set of states that thermalize o.a. for a given $w(t)$ will continue to do so for a different weight function. We do not attempt a nontrivial answer to this question here, but note that given any finite set of $M_w$ weight functions $\{w_k(t)\}_{k=1}^{M_w}$ each with a fraction of at most $f_\lambda$ states that fail to thermalize o.a., a fraction $1 - M_w f_\lambda$ of basis states must thermalize over all these choices of weight functions, which may be regarded as "almost all" if $f_\lambda \ll 1/M_w$. For all practical purposes then, if one envisions making measurements with a finite number $M_w$ of weight functions, it is sufficient to constrain the fraction of nonthermal states to an accuracy determined by $M_w$ for "almost all" physical states to thermalize.

In a very similar way, we can show additional results concerning instantaneous thermal equilibrium in small energy shells. For this purpose, let us consider a shell $\Sigma_{\Delta E} \subset \Sigma_d$ of $d_{\Delta E}$ energy levels whose level spacings are entirely within the energy band:

$$|E_q - E_r| < \Delta E, \quad \text{for all } E_q, E_r \in \Sigma_{\Delta E}. \tag{86}$$

Then, the following statements hold:

**Proposition 5.4** (Instantaneous thermal equilibrium in small energy shells)**.** *Let $\hat{\Pi}_A$ satisfy energy-band thermalization with bandwidth $\Delta E > 0$ and accuracy $\epsilon > 0$ within an energy shell $\Sigma_d$. Then,*

1. *$\hat{\Pi}_A$ thermalizes to $\langle \hat{\Pi}_A \rangle_{\Sigma_d}$ in any state $\hat{\rho} : \Sigma_d \to \Sigma_d$ in the energy shell, with accuracy:*

$$\left| \text{Tr}[\hat{\rho}\hat{\Pi}_A] - \langle \hat{\Pi}_A \rangle_{\Sigma_d} \right| < \epsilon \sqrt{d \, \text{Tr}[\hat{\rho}^2]} \tag{87}$$

2. *If $\mathcal{B}_{\Delta E}$ is an orthonormal basis for $\Sigma_{\Delta E}$ with $d_{\Delta E}$ states, the fraction $f_\lambda[\mathcal{B}_{\Delta E}]$ of basis states that fail to thermalize to accuracy $\lambda$, i.e.*

$$f_\lambda[\mathcal{B}_{\Delta E}] \equiv \frac{1}{d_{\Delta E}} \sum_{|k\rangle \in \mathcal{B}_{\Delta E}} \overline{\Theta}\left( \left| \langle k|\hat{\Pi}_A|k\rangle - \langle \hat{\Pi}_A \rangle_{\Sigma_d} \right| \geq \lambda \right) \tag{88}$$

   *is constrained by:*

$$f_\lambda[\mathcal{B}_{\Delta E}] < \frac{\epsilon}{\lambda}\sqrt{\frac{2d}{d_{\Delta E}}}. \tag{89}$$

*Proof.* The proof is a straightforward adaptation of those of Theorems 5.2 and 5.3, but here $w_0$ does not occur due to a lack of time averaging and the fact that there are no contributions from outside the bandwidth $\Delta E$. $\qquad\square$

We expect that the right hand side of Eq. (89) can be made small with sufficiently small $\epsilon$ if $d_{\Delta E}$ is not too much smaller than $d$ (as should be the case for accessible values of $\Delta E$). This implies, for example, that the dynamics of thermalization tends to be relevant (or at least nontrivial) only for initial states whose support is larger than $\Delta E$, if an observable is known to satisfy energy-band thermalization with this bandwidth.

In summary, Theorems 5.2 and 5.3 are some of our key conceptual results, and complete one half of our present effort to describe the thermalization of states without any specific reference

to eigenstates. In particular, we have shown that energy-band thermalization directly implies the thermalization (o.a.) of (almost all) physical states in any basis that one might be able to access in an experiment, over accessible timescales. The remaining half entails establishing energy-band thermalization through accessible quantum dynamical processes, which is the goal of the next section.

# 6 Global thermalization from a single initial state

For the purpose of conclusively establishing energy-band thermalization, having already implicated the autocorrelator

$$\text{Tr}[e^{-i\hat{H}t}\hat{\Pi}_A e^{i\hat{H}t}\hat{\Pi}_A]$$

in Sec. 2.2, all that remains is to rigorously connect the behavior of autocorrelators over *accessible* time scales to energy-band thermalization. Here, it becomes necessary to separate two different possibilities for the operator $\hat{\Pi}_A$:

1. Global thermalization: The thermal value $\langle\hat{\Pi}_A\rangle_{\Sigma_d} = \langle\hat{\Pi}_A\rangle$ is uniform for any choice of energy shell $\Sigma_d$ of interest, matching its thermal value $\langle\hat{\Pi}_A\rangle \equiv \text{Tr}[\hat{\Pi}_A]/D$ in the full Hilbert space $\mathcal{H}$.

2. Energy shell thermalization: The thermal value $\langle\hat{\Pi}_A\rangle_{\Sigma_d}$ differs between energy shells, varying (usually) smoothly as a function of energy.

The reason for separating these cases is as follows: if $\hat{\Pi}_A$ could be restricted to any energy shell $\Sigma_d$ of interest, then one could simply shrink the global Hilbert space to the energy shell $\mathcal{H} \to \Sigma_d$ and treat the case of energy-dependent thermalization as if it were global thermalization. However, we do not expect that observables of relevance in experiments can be rigorously restricted to such energy shells while maintaining certain properties that we require (e.g., coupling to a finite temperature state in an external bath would require accounting for some inaccessible properties of the finite temperature state), necessitating an alternate strategy (involving interference effects) to access energy-band thermalization in this case.

To connect autocorrelators to global energy-band thermalization, we only need a straightforward adaptation of the proofs of wavefunction ergodicity [40, 44, 49] from global eigenstate thermalization to energy-band thermalization with a careful accounting of weighted time averages. This is carried out in Sec. 6.2. More significantly, in Sec. 6.3, we combine this result with those of Sec. 5 to completely bypass properties of the energy levels, and directly connect the decay of autocorrelators to global thermalization, realizing a fully quantum analogue of the classical statement containing Eq. (1).

The extension to energy-shell thermalization when projections to the energy shell are not directly possible is more nontrivial, and discussed in Sec. 7. For both of these developments, however, we need to consider a special class of weighting functions $w(t)$ for time averages, which we now turn to as the subject of Sec. 6.1.

## 6.1 Completely positive time averages

In view of the classical statement Summary 3.3 relating thermalization to classical dynamics in a single initial distribution, let us consider the mixed state

$$\hat{\rho}_A \equiv \frac{1}{\text{Tr}[\hat{\Pi}_A]}\hat{\Pi}_A \tag{90}$$

associated with a uniform distribution over the eigenstates of the observable $\hat{\Pi}_A$ with eigenvalue 1. The $w(t)$-weighted time average of $\hat{\Pi}_A$ in this state, corresponding to the autocorrelator, differs from the thermal value $\langle \hat{\Pi}_A \rangle$ by [from Eq. (63)]:

$$\int dt\, w(t)\, \mathrm{Tr}[\hat{\rho}_A(t)\hat{\Pi}_A] - \langle \hat{\Pi}_A \rangle = \frac{1}{\mathrm{Tr}[\hat{\Pi}_A]} \sum_{n,m} \widetilde{w}(E_n - E_m) \left| \langle E_n | \left( \hat{\Pi}_A - \langle \hat{\Pi}_A \rangle \hat{\mathbb{1}} \right) | E_m \rangle \right|^2, \quad (91)$$

where we recall that $\langle \hat{\Pi}_A \rangle = \mathrm{Tr}[\hat{\Pi}_A]/D$.

Conveniently, if we could set

$$\widetilde{w}(E_n - E_m) = \overline{\Theta}(|E_n - E_m| < \Delta E), \quad (92)$$

associated with

$$w(t) = \frac{\Delta E}{\pi} \mathrm{sinc}(\Delta E t), \quad (93)$$

then the right hand side of Eq. (91) is proportional to the quantity of interest in energy-band thermalization [Eq. (71)], and we would have already achieved our goal of connecting the latter to autocorrelators. But this is a problem from an accessibility standpoint: $w(t) = (\Delta E/\pi)\mathrm{sinc}(\Delta E t)$ is nonzero in an infinite range of times $t \in (-\infty, \infty)$, and we would be no better off than the infinite time average in Eq. (21). On the other hand, if we could ensure that $\widetilde{w}(\delta E) \geq 0$ even if $w(t)$ has support on some finite interval of time, then all terms on the right hand side would be non-negative, and we can derive rigorous inequalities by dropping the subset of terms with $|E_n - E_m| \geq \Delta E$.

We will therefore impose special positivity conditions on $w(t)$. We will denote $w(t)$ by $w_+(t)$ if the following two conditions are satisfied:

1. Non-negative values in the time-domain (as in Sec. 5.1):

$$w_+(t) \geq 0, \text{ for all } t. \quad (94)$$

Through a relation between non-negativity conditions and Fourier transforms [75, 76], this implies:

$$|\widetilde{w}_+(E)| < |\widetilde{w}_+(0)| = 1, \quad (95)$$

as in Eq. (66).

2. Non-negative values of the Fourier transform:

$$\widetilde{w}_+(\delta E) \geq 0, \text{ for all } \delta E \in \mathbb{R}. \quad (96)$$

We will call a weight function $w_+(t)$ satisfying Eqs. (65), (94), and (96) a "completely positive" weight function. One example is:

$$w_+(t) = \frac{1}{T}\left(1 - \frac{|t|}{T}\right)\overline{\Theta}(|t| \leq T), \quad (97)$$

where $\overline{\Theta}(x) = 1$ if $x$ is true and 0 otherwise. However, the conventional time average $w(t) = (2T)^{-1}\overline{\Theta}(|t| \leq T)$ is not completely positive. Another important class of functions that are completely positive, which we will not discuss until Sec. 9, are discrete-time versions of Eq. (97) which are more accessible in an experimental setting.

Additionally, for some $0 < W < 1$, let $\Delta E_W > 0$ be such that:

$$\widetilde{w}_+(\delta E) > W, \quad \text{for all } |\delta E| < \Delta E_W. \quad (98)$$

We note again that the range of allowed values of $\Delta E_W$ for any given $W$ is purely a property of our choice of function $w_+(t)$, and does not depend on any properties (such as energy levels) of the system.

Where we used the general (non-negative) weight functions $w(t)$ to consider thermalization o.a. in arbitrary initial states, we will use completely positive weight functions $w_+(t)$ to probe autocorrelators and derive rigorous bounds on energy-band thermalization. With this restriction, each term in Eq. (91) is necessarily non-negative.

## 6.2 Energy-band thermalization from autocorrelators

Now, we will explicitly show that energy-band thermalization can be inferred from autocorrelators with a completely positive weighting function $w_+(t)$ which could, in principle, have support only on a finite range (or even finite set) of times. The intuition behind this relation has been described following Eq. (91), and we begin here by filling in the technical details. Formally, we will adapt a proof strategy that appeared in Refs. [40, 44, 49], generalize it to time averages with weights $w_+(t)$ and energy-band thermalization in place of (weak) eigenstate thermalization, and excise from it all references to the classical limit or limits such as $T \to \infty$ (or $\Delta E \to 0$). The following proposition results:

**Proposition 6.1** (Autocorrelators that thermalize o.a. imply global energy-band thermalization). *For a given choice of $w_+(t)$ and $W > 0$, with $\Delta E_W > 0$ satisfying Eq. (98), if $\hat{\Pi}_A$ thermalizes o.a. to $\langle\hat{\Pi}_A\rangle$ in $\hat{\rho}_A$ to accuracy $\epsilon_A$,*

$$\left| \int dt \, w_+(t) \operatorname{Tr}[\hat{\rho}_A(t)\hat{\Pi}_A] - \langle\hat{\Pi}_A\rangle \right| < \epsilon_A, \tag{99}$$

*then for any $\Delta E < \Delta E_W$ the observable $\hat{\Pi}_A$ shows energy-band thermalization in the full Hilbert space $\mathcal{H}$ with bandwidth $\Delta E$ and accuracy $\epsilon$ given by (where $D = \dim \mathcal{H}$):*

$$\frac{1}{D} \operatorname{Tr}\left\{ \left[ \hat{\Pi}_A - \langle\hat{\Pi}_A\rangle \hat{\mathbb{1}} \right]^2_{(\mathcal{H},\Delta E)} \right\} < \epsilon^2 = \frac{\operatorname{Tr}[\hat{\Pi}_A]}{WD} \epsilon_A. \tag{100}$$

*Proof.* This follows by separating the $|E_n - E_m| < \Delta E$ terms in Eq. (91) from the $|E_n - E_m| \geq \Delta E$ terms, and noting that the latter are all non-negative; see App. D.4.1. $\qquad\square$

The connection between this proposition and the standard wavefunction ergodicity theorems is as follows: If we take the classical limit of the autocorrelator as well as the quantum $D \to \infty$ limit, then take $W \to 1$ so that $\Delta E_W \to 0$ [for, say, the choice of $w_+(t)$ in Eq. (97)], and combine the above proposition with Prop. C.1, then we recover (our equivalent of) the standard wavefunction ergodicity results by which classical ergodicity implies (weak) eigenstate (or eigenspace) thermalization for almost all states as well as vanishing near-diagonal matrix elements of $\hat{\Pi}_A$. Proposition 6.1 is therefore essentially a quantitative version of this direction of the semiclassical theorem. The reverse direction, where these quantum properties imply classical ergodicity, is conventionally proved only in the classical limit for the appropriate classes of models in the limit of $T \to \infty$. In our case, Theorem 5.3 for fully quantum systems replaces the classical reverse direction with a purely quantum result about thermalization in an arbitrary basis, and in its formulation in terms of energy shells, also anticipates the reverse direction for energy shell thermalization (to be discussed in Sec. 7).

## 6.3 Bypassing energy levels: Global thermalization from autocorrelators

Let us see how the combination of Proposition 6.1 and the quantum thermalization results of Sec. 5 allows us to discuss global thermalization without any specific reference to the

1291  energy levels. First, we can directly relate the decay of autocorrelators to the thermalization of
1292  physical states without relying on energy-band thermalization as an intermediary. Specifically,
1293  combining Proposition 6.1 and Theorem 5.3 gives:

**Corollary 6.2** (Autocorrelator thermalization o.a. implies almost all physical states thermalize
1295  o.a.). *Let Eq. (99) hold for $\hat{\Pi}_A$ with $\langle \hat{\Pi}_A \rangle = \mathrm{Tr}[\hat{\Pi}_A]/D$ and let any weight function $w(t)$, satisfying*

$$|\widetilde{w}(\delta E)| \leq w_0, \quad \text{for all} \ \ |\delta E| \geq \Delta E_W, \tag{101}$$

1296  *be given. Then for any complete orthonormal basis $\mathcal{B}$ of the full Hilbert space $\mathcal{H}$, the fraction $f_\lambda[\mathcal{B}]$*
1297  *of basis states $|k\rangle \in \mathcal{B}$, for which thermalization o.a. to accuracy $\lambda$*

$$\left| \int \mathrm{d}t \, w(t) \, \mathrm{Tr}[\hat{\rho}_k(t)\hat{\Pi}_A] - \langle \hat{\Pi}_A \rangle \right| < \lambda \tag{102}$$

1298  *fails to occur, is constrained by:*

$$f_\lambda[\mathcal{B}] < \frac{1}{\lambda} \sqrt{\frac{2\,\mathrm{Tr}[\hat{\Pi}_A]}{D}} \left( w_0 + \frac{\epsilon_A}{W} \sqrt{\frac{\mathrm{Tr}[\hat{\Pi}_A]}{D}} \right). \tag{103}$$

1299  Here, $\mathrm{Tr}[\hat{\Pi}_A] \leq D$, and we should expect the right hand side of Eq. (103) to be small
1300  provided $\epsilon_A$ and $w_0$ are small and $W$ is comparable to 1.

1301  Furthermore, on account of Theorem 5.2, energy-band thermalization implies the ther-
1302  malization of autocorrelators with weight functions $w(t)$ of "longer duration" than $w_+(t)$.
1303  Once again, using Proposition 6.1 to directly connect autocorrelators to these longer duration
1304  autocorrelators, we have a "feed-forward" result:

**Corollary 6.3** (Autocorrelator thermalization o.a. feeds forward to longer timescales). *For the
1306  same setting as Proposition 6.1, if the autocorrelator thermalizes o.a. with weight $w_+(t)$:*

$$\left| \int \mathrm{d}t \, w_+(t) \, \mathrm{Tr}[\hat{\rho}_A(t)\hat{\Pi}_A] - \langle \hat{\Pi}_A \rangle \right| < \epsilon_A, \tag{104}$$

1307  *then for any weight function $w(t)$ satisfying*

$$|\widetilde{w}(\delta E)| \leq w_0, \quad \text{for all} \ \ |\delta E| \geq \Delta E_W, \tag{105}$$

1308  *the autocorrelator thermalizes o.a. to accuracy:*

$$\left| \int \mathrm{d}t \, w(t) \, \mathrm{Tr}[\hat{\rho}_A(t)\hat{\Pi}_A] - \langle \hat{\Pi}_A \rangle \right| < w_0 + \frac{\epsilon_A}{W} \sqrt{\frac{\mathrm{Tr}[\hat{\Pi}_A]}{D}}. \tag{106}$$

1309  Finally, let $\mathcal{B}_A$ be any complete orthonormal eigenbasis of $\hat{\Pi}_A$ (which has degenerate
1310  eigenvalues in $\{0,1\}$). Then we can write the initial state $\hat{\rho}_A$ as

$$\hat{\rho}_A = \frac{1}{\mathrm{Tr}[\hat{\Pi}_A]} \sum_{\substack{|k_A\rangle \in \mathcal{B}_A: \\ \hat{\Pi}_A|k_A\rangle = |k_A\rangle}} |k_A\rangle \langle k_A|. \tag{107}$$

1311  If we know for certain that a fraction $\left( 1 - f_\lambda[\mathcal{B}_A] \right)$ of this eigenbasis thermalizes o.a. with
1312  accuracy $\lambda$ and weight function $w(t)$ (and the remaining $f_\lambda[\mathcal{B}_A]D$ basis states each has a $\hat{\Pi}_A$
1313  expectation value of at most 1), then the fraction of basis states with $\hat{\Pi}_A$-eigenvalue 1 that fail

to thermalize to this accuracy cannot exceed $f_\lambda[\mathcal{B}_A]D/\text{Tr}[\hat{\Pi}_A]$. Then, applying the triangle inequality to the thermalization of Eq. (107) implies that

$$\left| \int dt\, w(t)\text{Tr}[\hat{\rho}_A(t)\hat{\Pi}_A] - \langle \hat{\Pi}_A \rangle \right| \leq \frac{D}{\text{Tr}[\hat{\Pi}_A]} f_\lambda[\mathcal{B}_A](1-\lambda) + \lambda. \tag{108}$$

For this to provide a nontrivial bound on the autocorrelator, it is essential that:

$$\frac{\text{Tr}[\hat{\Pi}_A]}{D} \gg f_\lambda[\mathcal{B}_A](1-\lambda); \tag{109}$$

in other words, the projector $\hat{\Pi}_A$ should be sufficiently "bulky" (i.e., by coarse-graining over many pure states such as for a few-body observable) for this to be possible with accessible $\lambda$ and $f_\lambda[\mathcal{B}_A]$.

We emphasize, however, that it may still be possible for a complete basis of states that is *not* an eigenbasis of $\hat{\Pi}_A$ to thermalize this observable without implying the thermalization of the autocorrelator, nor therefore the thermalization of all other states. This strongly implicates $\hat{\rho}_A$ or any of its eigenbases as the most direct determiners of thermalization — which is a more accessible set of states than the complete set of all product states implicated in a slightly different setting [56] — while the observable may continue to thermalize in certain other bases of states (at least over inaccessibly long timescales) without any such implications for global thermalization [16, 78].

Collectively, these results establish a complete framework for the global thermalization of physical states in a quantum system that do not require any mention of the energy levels or eigenstates of a system (though the properties of observables in eigenstates also follow from Proposition 6.1, and e.g. the connection to eigenspaces discussed in Sec. 5.2). We summarize these results at an intuitive level as follows (where "timescale" refers to the choice of $w(t)$ or $w_+(t)$):

**Summary 6.4** (Bypassing eigenstates for global thermalization o.a.)**.** *The following implications describe the (global) quantum thermalization o.a. of a given observable without reference to eigenstates:*

1. *The thermalization o.a. of the autocorrelator over a fixed timescale implies the thermalization o.a. of almost all physical states at all longer timescales [Corollary 6.2].*

2. *The thermalization o.a. of the autocorrelator over a fixed timescale implies its thermalization o.a. at all longer timescales [Corollary 6.3].*

3. *The thermalization o.a. of almost all physical states over a fixed timescale implies the thermalization o.a. of the autocorrelator over the same timescale, if the observable is sufficiently bulky [Eq. (108)].*

*In particular, the thermalization of the autocorrelator over a fixed timescale is expected to be an accessible property.*

# 7 Thermalization in energy shells with conserved charges

Here, we will tackle the problem of accessing thermalization in energy shells in terms of the dynamics of a single initial state. As we have indicated previously, if it is possible to prepare/measure variants of $\hat{\Pi}_A$ that are projected to a single energy shell $\Sigma_d$, i.e.:

$$\hat{\Pi}_A(\Sigma_d) = \hat{\Pi}_{\Sigma_d}\hat{\Pi}_A\hat{\Pi}_{\Sigma_d}, \tag{110}$$

then the considerations of Sec. 6 can be generalized in a straightforward manner to energy shells by taking our full Hilbert space to be $\Sigma_d$ and directly applying our previous considerations to the observable $\hat{\Pi}_A(\Sigma_d)$ in place of $\hat{\Pi}_A$. In most systems, however, we do not expect this to be experimentally feasible: $\hat{\Pi}_{\Sigma_d}$ is often not easy to implement, and $\hat{\Pi}_A(\Sigma_d)$ is often not a projector, being correspondingly difficult to prepare or measure. An alternate strategy convenient for experiments, involving a finite temperature bath as in Eq. (28), does not appear to allow a conclusive determination of thermalization with accessible measurements due to a strong dependence on the (seemingly inaccessible) precise energy distribution and quantum coherence properties of the finite temperature state. Due to these complications, we will develop a somewhat different method of specializing to energy shells, beyond the trivial case of taking the shell to be our full Hilbert space. We will also briefly describe at the end of this section how the same strategy generalizes in the presence of conserved quantities other than energy, on which the thermal value may show a macroscopic dependence.

### 7.1   Motivation: general structure of accessible thermalization

To motivate our strategy, let us consider how to constrain the matrix elements of $(\hat{\Pi}_A - \langle \hat{\Pi}_A \rangle_{\Sigma_d} \hat{\mathbb{1}})$ within an energy-band in $\Sigma_d$ given its expectation value in an arbitrary "operator" $\hat{\gamma}$:

$$\mathrm{Tr}\left[ \hat{\gamma} \left( \hat{\Pi}_A - \langle \hat{\Pi}_A \rangle_{\Sigma_d} \hat{\mathbb{1}} \right) \right] = \sum_{n,m} \langle E_n | \hat{\gamma} | E_m \rangle \langle E_m | \left( \hat{\Pi}_A - \langle \hat{\Pi}_A \rangle_{\Sigma_d} \hat{\mathbb{1}} \right) | E_n \rangle. \tag{111}$$

Generically, $\hat{\Pi}_A$ has nonvanishing matrix elements between any two energy levels in $\mathcal{H}$. Further, individual off-diagonal matrix elements are typically inaccessible for both $\hat{\gamma}$ and $\hat{\Pi}_A$. To obtain a rigorous inequality for the subset of matrix elements within the energy band without having detailed knowledge of the other off-diagonal elements, the easiest strategy is to ensure that each term on the right hand side is non-negative, so that the subset of energy-band terms is always constrained to be no greater than the expectation value on the left hand side. This requires precise phase cancellations between the matrix elements of $\hat{\gamma}$ and $(\hat{\Pi}_A - \langle \hat{\Pi}_A \rangle_{\Sigma_d} \hat{\mathbb{1}})$:

$$\arg \langle E_n | \hat{\gamma} | E_m \rangle = -\arg \left[ \langle E_m | \left( \hat{\Pi}_A - \langle \hat{\Pi}_A \rangle_{\Sigma_d} \hat{\mathbb{1}} \right) | E_n \rangle \right], \tag{112}$$

Crucially, Eq. (112) appears to require $\hat{\gamma}$ to be constructed out of $(\hat{\Pi}_A - \langle \hat{\Pi}_A \rangle_{\Sigma_d} \hat{\mathbb{1}})$ (to ensure that $\hat{\gamma}$ knows about the phase of the right hand side) sandwiched between suitably chosen "phase-cancelling" functions of $\hat{H}$ (so that one can implement dynamics or restrict to an energy shell without altering the phase). This appears to be a natural way to ensure this phase cancellation without any detailed analytical knowledge of $\hat{\Pi}_A$, beyond the ability to prepare/measure linear functionals of this observable. Given some (possibly multidimensional) parameter $s$, the preceding argument suggests the general structure

$$\hat{\gamma} = \frac{1}{\mathrm{Tr}[\hat{\Pi}_A]} \int \mathrm{d}s \, \xi(s) \, f_s(\hat{H}) \left( \hat{\Pi}_A - \langle \hat{\Pi}_A \rangle_{\Sigma_d} \hat{\mathbb{1}} \right) g_s(\hat{H}) \tag{113}$$

of accessible operators $\hat{\gamma}$ constructed as linear functionals of the observable (where $1/\mathrm{Tr}[\hat{\Pi}_A]$ just specifies a convention for normalization). This is because the matrix elements of this operator are then

$$\langle E_n | \hat{\gamma} | E_m \rangle = \frac{1}{\mathrm{Tr}[\hat{\Pi}_A]} \int \mathrm{d}s \, \xi(s) \, f_s(E_n) g_s(E_m) \, \langle E_n | \left( \hat{\Pi}_A - \langle \hat{\Pi}_A \rangle_{\Sigma_d} \hat{\mathbb{1}} \right) | E_m \rangle, \tag{114}$$

which satisfies Eq. (112) due to the Hermiticity of $\hat{\Pi}_A$, provided that the functions $f_s$ and $g_s$ together contribute a factor with zero phase (i.e. their contribution is non-negative):

$$\int \mathrm{d}s \, \xi(s) \, f_s(E_n) g_s(E_m) \geq 0, \quad \text{for all } E_n, E_m. \tag{115}$$

This structure is precisely what was implicitly obtained in Secs. 6.1 and 6.2: these sections correspond to the following choice of $\hat{\gamma}$, with $s = t$, $\xi(s) = w_+(t)$ and $f_s(\hat{H}) = g_s(\hat{H})^\dagger = e^{-i\hat{H}t}$ [for which Eq. (111) is equivalent to Eq. (91) with weight $w_+(t)$, provided $\langle\hat{\Pi}_A\rangle = \text{Tr}[\hat{\Pi}_A]/D$]:

$$\hat{\gamma}_A^{\text{global}} = \frac{1}{\text{Tr}[\hat{\Pi}_A]} \int dt \; w_+(t) e^{-i\hat{H}t} \left(\hat{\Pi}_A - \langle\hat{\Pi}_A\rangle_{\Sigma_d} \hat{\mathbb{1}}\right) e^{i\hat{H}t}, \tag{116}$$

with the complete positivity of $w_+(t)$ ensuring that the matrix elements of $\hat{\gamma}_A^{\text{global}}$,

$$\langle E_n|\hat{\gamma}_A^{\text{global}}|E_m\rangle = \frac{1}{\text{Tr}[\hat{\Pi}_A]} \widetilde{w}_+(E_n - E_m)\langle E_n|\left(\hat{\Pi}_A - \langle\hat{\Pi}_A\rangle_{\Sigma_d} \hat{\mathbb{1}}\right)|E_m\rangle, \tag{117}$$

perfectly cancel out any complex phases as in Eq. (112).

## 7.2   Echo dynamics for accessing energy shells

For energy shell thermalization, in addition to controlling the energy band via $E_n - E_m$, we will need to control the overall range of energies, say via $(E_n + E_m)/2$, and require it to be centered around some $E_c$ that may correspond to the "center" (or some other point) in the energy shell. This suggests taking a 2-dimensional parameter $s = (t_1, t_2)$, with

$$f_s(\hat{H}) = e^{-i\hat{H}t_1} e^{iE_c t_1}, \quad g_s(\hat{H}) = e^{i\hat{H}t_2} e^{-iE_c t_2}, \quad \text{and } \xi(s) = w_+\left(\frac{t_1 + t_2}{2}\right) v_+(t_1 - t_2), \tag{118}$$

for completely positive weight functions $w_+(t)$ and $v_+(t)$, so that

$$\int ds \; \xi(s) f_s(E_n)g_s(E_m) = \int dt_1 dt_2 \; w_+\left(\frac{t_1 + t_2}{2}\right) v_+(t_1 - t_2) \; e^{-i(E_n - E_c)t_1} e^{i(E_m - E_c)t_2}$$

$$= \widetilde{w}_+(E_n - E_m)\widetilde{v}_+\left(\frac{E_n + E_m}{2} - E_c\right) \geq 0, \tag{119}$$

satisfying Eq. (117). As $v_+(t)$ is completely positive, $\widetilde{v}_+(0) = 1$ is the maximum value of its Fourier transform, ensuring that the weights in the energy domain peak (or at least attain a maximum) at $(E_n + E_m)/2 = E_c$. Such a choice yields the following operator:

$$\hat{\gamma}_A^{\text{shell}} = \frac{1}{\text{Tr}[\hat{\Pi}_A]} \int dt_1 dt_2 \; w_+\left(\frac{t_1 + t_2}{2}\right) v_+(t_1 - t_2) e^{-iE_c(t_1 - t_2)} \; e^{-i\hat{H}t_1}\left(\hat{\Pi}_A - \langle\hat{\Pi}_A\rangle_{\Sigma_d} \hat{\mathbb{1}}\right) e^{i\hat{H}t_2} \tag{120}$$

in which we must seek to constrain the expectation values of $\hat{\Pi}_A$. For $v_+(t_1 - t_2) \to \delta(t_1 - t_2)$ [e.g., interpreted as a $T \to 0$ limit of Eq. (97) if one wants to ensure complete positivity and finite time ranges], $\gamma_{\text{shell}} \to \gamma_{\text{global}}$, so our choice also recovers the case of global thermalization as a special case.

It is also convenient to express the expectation value of our observable (more precisely, its deviation from its thermal value) in $\hat{\gamma}_A^{\text{shell}}$ in terms of the "accessible" initial state $\hat{\rho}_A = \hat{\Pi}_A/\text{Tr}[\hat{\Pi}_A]$. We have:

$$\text{Tr}\left[\hat{\gamma}_A^{\text{shell}}\left(\hat{\Pi}_A - \langle\hat{\Pi}_A\rangle_{\Sigma_d} \hat{\mathbb{1}}\right)\right] = \int dt_1 dt_2 \; w_+\left(\frac{t_1 + t_2}{2}\right) v_+(t_1 - t_2) e^{-iE_c(t_1 - t_2)} \left\{ \text{Tr}[e^{-i\hat{H}t_1}\hat{\rho}_A e^{i\hat{H}t_2}\hat{\Pi}_A]\right.$$

$$\left. -2\langle\hat{\Pi}_A\rangle_{\Sigma_d} \text{Tr}[\hat{\rho}_A e^{-i\hat{H}(t_1 - t_2)}] + \frac{\langle\hat{\Pi}_A\rangle_{\Sigma_d}^2}{\langle\hat{\Pi}_A\rangle} \frac{1}{D} \text{Tr}[e^{-i\hat{H}(t_1 - t_2)}]\right\}, \tag{121}$$

where $\langle \hat{\Pi}_A \rangle = \mathrm{Tr}[\hat{\Pi}_A]/D$ is the global average of $\hat{\Pi}_A$ as before. This is a weighted time average of the following combination of quantities we will refer to as "quantum dynamical echoes" (corresponding to the terms above enclosed by braces; we will use the term collectively for all the below quantities as well as Eq. (121)):

$$L_{AA}(t_1, t_2) - 2\langle \hat{\Pi}_A \rangle_{\Sigma_d} L_A(t_1 - t_2) + \frac{\langle \hat{\Pi}_A \rangle_{\Sigma_d}^2}{\langle \hat{\Pi}_A \rangle} L_{\mathcal{H}}(t_1 - t_2), \tag{122}$$

where

$$L_{AA}(t_1, t_2) \equiv \mathrm{Tr}[e^{-i\hat{H}t_1} \hat{\rho}_A e^{i\hat{H}t_2} \hat{\Pi}_A], \tag{123}$$

$$L_A(t) \equiv \mathrm{Tr}[e^{-i\hat{H}t} \hat{\rho}_A], \tag{124}$$

$$L_{\mathcal{H}}(t) \equiv \frac{1}{D} \mathrm{Tr}[e^{-i\hat{H}t}]. \tag{125}$$

As noted in Sec. 2.3, $L_{AA}(t_1, t_2)$ is a coarse-grained variant of the Loschmidt echo, which measures quantum interference effects between two possible evolution times of the state $\hat{\rho}_A$ as witnessed by the observable $\hat{\Pi}_A$. $L_A(t)$ is a stranger quantity that acts on the initial state $\hat{\rho}_A$ but has no observable, and we would like to refer to it as a "one-sided" echo; alternatively, it can be interpreted as an echo of the maximally mixed state with the observable $\hat{\Pi}_A$. Finally, $L_{\mathcal{H}}(t)$ is the spectral function of the Hamiltonian $\mathcal{H}$, measuring the Fourier transform of its *probability* density of energy levels (normalized to 1); its squared magnitude is the spectral form factor [68, 79]:

$$K(t) = |L_{\mathcal{H}}(t)|^2, \tag{126}$$

which has been of considerable interest as an observable-independent probe of energy level statistics, such as for the measurements discussed in Sec. 1. We will show in Sec. 9 that all three quantities $L_{AA}(t_1, t_2)$, $L_A(t)$ and $L_{\mathcal{H}}(t)$ are experimentally accessible quantities: the first two from the dynamics of a single initial state, and the third from state-independent dynamics. These are the only quantities requiring quantum measurements in an experiment; the remaining operations in Eq. (121), in particular the selection of the energy shell via $E_c$ and precise mathematical form of the weight function (other than the choice of the set of times $t_1, t_2$), may be implemented classically at the post-processing stage.

Finally[17], given some $V$ such that $0 < V < 1$ let us introduce $\mathcal{E}_V$ as a set of energy values containing $E_c$ in which $\widetilde{v}_+(E - E_c)$ is as large as $V$:

$$\widetilde{v}_+(E - E_c) \geq V, \quad \text{for all } E \in \mathcal{E}_V. \tag{127}$$

This statement is the energy shell analogue of Eq. (98) for the energy band. The set $\mathcal{E}_V$ will define our energy shell of interest as Hilbert space spanned by *all* the energy levels in $\mathcal{H}$ within this set:

$$\Sigma_{\mathcal{E}_V}(V) \equiv \bigcup_{E \in \mathcal{E}_V} \mathcal{H}(E). \tag{128}$$

## 7.3 Energy shell thermalization from quantum dynamical echoes

Given the setup above, we are now in a position to derive rigorous results on energy shell thermalization, analogous to Sec. 6. Our first task is to constrain energy-band thermalization within an energy shell based on the expectation value of $\hat{\Pi}_A$ in $\hat{\gamma}_A^{\text{shell}}$, which is the energy-shell counterpart of Proposition 6.1. This can be done by using the expression for $\hat{\gamma}_A^{\text{shell}}$ in Eq. (120)

---

[17]Here, we assume that $\mathcal{E}_V$ will typically be a closed set, while we have taken the energy-band to be an open set/interval; this is purely a choice of convention.

to write the expectation value on the left hand side of Eq. (121) in the energy eigenbasis, given by the general form in Eq. (111):

$$\mathrm{Tr}\left[\hat{\gamma}_A^{\text{shell}}\left(\hat{\Pi}_A - \langle\hat{\Pi}_A\rangle_{\Sigma_d}\hat{\mathbb{1}}\right)\right] = \frac{1}{\mathrm{Tr}[\hat{\Pi}_A]}\sum_{n,m}\widetilde{w}_+(E_n - E_m)\widetilde{v}_+\left(\frac{E_n + E_m}{2} - E_c\right)\left|\langle E_m|\left(\hat{\Pi}_A - \langle\hat{\Pi}_A\rangle_{\Sigma_d}\hat{\mathbb{1}}\right)|E_n\rangle\right|^2. \tag{129}$$

As we have ensured that each term on the right hand side is non-negative, we can restrict the expression to pairs $(E_n, E_m)$ satisfying $|E_n - E_m| < \Delta E_W$ and $(E_n + E_m)/2 \in \mathcal{E}_V$ (in fact, a subset of these corresponding to energy-band thermalization), and obtain an inequality bounding this expression from above by the left hand side. This addresses energy-band thermalization in $\Sigma(\mathcal{E}_V)$ with bandwidth $\Delta E$:

**Theorem 7.1** (Quantum dynamical echoes constrain energy-band thermalization in energy shells). *For a given choice of weighting functions $w_+(t)$, $v_+(t)$ and constants $W, V > 0$, with a bandwidth $\Delta E_W > 0$ satisfying Eq. (98) and a set of energies $\mathcal{E}_V$ satisfying Eq. (127) containing $E = E_c$ for a given "central" energy $E_c$, if the expected deviation of $\hat{\Pi}_A$ from $\langle\hat{\Pi}_A\rangle_{\Sigma_d}$ in the corresponding operator $\hat{\gamma}_A^{\text{shell}}$ is less than $\epsilon_A$,*

$$\left|\mathrm{Tr}\left[\hat{\gamma}_A^{\text{shell}}\left(\hat{\Pi}_A - \langle\hat{\Pi}_A\rangle_{\Sigma_d}\hat{\mathbb{1}}\right)\right]\right| < \epsilon_A, \tag{130}$$

*then for any energy shell $\Sigma_d \subset \Sigma(\mathcal{E}_V)$ of dimension $d$, the observable $\hat{\Pi}_A$ satisfies energy-band thermalization with bandwidth $\Delta E$ to accuracy $\epsilon$, given by:*

$$\frac{1}{d}\mathrm{Tr}\left\{\left[\hat{\Pi}_A - \langle\hat{\Pi}_A\rangle_{\Sigma_d}\hat{\mathbb{1}}\right]^2_{(\Sigma_d, \Delta E)}\right\} < \epsilon^2 = \frac{\mathrm{Tr}[\hat{\Pi}_A]}{WVd}\epsilon_A. \tag{131}$$

*Proof.* The proof follows in a similar manner to that of Proposition 6.1, except that now we only consider pairs of energy levels satisfying the conditions $|E_n - E_m| < \Delta E_W$ and $E_n, E_m \in \Sigma_d$; see App. D.5.1. $\square$

As $W, V$ may be chosen by hand to have accessible values (usually comparable to 1), for the bound in Eq. (131) to be nontrivial, we require that

$$\epsilon_A \ll \frac{d}{\mathrm{Tr}[\hat{\Pi}_A]}. \tag{132}$$

For the accessibility of this bound, it is necessary that the right hand side be accessible (i.e., not too small). For few body observables in a many-body system, we expect that $\mathrm{Tr}[\hat{\Pi}_A] \sim D$, which means that $\dim \Sigma_d / \dim \mathcal{H} = d/D$ cannot be too small if Theorem 7.1 is to remain useful; e.g., we may want to impose

$$\frac{d}{D} \gtrsim \frac{1}{N^\alpha}, \tag{133}$$

in a system of $N$ particles, for some $\alpha > 0$, for accessibility purposes. Intuitively, this is in line with what we saw for global thermalization in Sec. 6: a vanishingly small fraction of states may fail to thermalize with accessible resources (Sec. 5) even with autocorrelator thermalization, and to have higher resolution and restrict this fraction (perhaps all the way to $< 1/D$) one must allow "inaccessible" values of $\epsilon_A$. Here, we see that the energy shell $\Sigma_d$ must be larger than an "inaccessibly small" fraction of the Hilbert space for us to be able to guarantee that it is not composed entirely or predominantly of states that do not thermalize. In this case, a rather convenient fact is that the ratio $d/D$ corresponding to a given energy shell is also measurable — e.g., it can be obtained by a suitable integral of $L_\mathcal{H}(t)$ [Eq. (125)], which is the Fourier transform of the probability density of energy levels.

Another important question is the converse direction: does energy-band thermalization in some energy shell $\Sigma_d$ imply the vanishing of a quantum dynamical echo of the form given by Eq. (129)? Only if this were so can we expect a measurement of these echos to reliably identify energy-band thermalization (though, in practice, likely in a somewhat smaller shell than $\Sigma_d$). Intuitively, it is straightforward to see that this is the case due to the equality in Eq. (129): if we make $\widetilde{w}_+$ and $\widetilde{v}_+$ vanishingly small outside the energy band and shell of interest, then the vanishing of the matrix elements of the observable implies a vanishing of the echo. Formally, this is established by a generalization of Theorem 5.2 (but where we specialize to the specific echo in Eq. (129)):

**Theorem 7.2** (Energy-band thermalization in energy shells constrains quantum dynamical echoes). *If $\hat{\Pi}_A$ satisfies energy-band thermalization to $\langle \hat{\Pi}_A \rangle_{\Sigma_d}$ with bandwidth $\Delta E > 0$ and accuracy $\epsilon > 0$ within an energy shell $\Sigma_d$ [Eq. (71)] spanning energy levels in the set $\mathcal{E}$, and one considers weighting functions $w(t)$ with bandwidth smaller than $\Delta E$, i.e. which satisfies [Eq. (67)]:*

$$|\widetilde{w}(\delta E)| \leq w_0, \quad \text{for all } \delta E \geq \Delta E,$$

*and $v(t)$ whose Fourier transform is vanishingly small outside the set of energies $\mathcal{E}_{\Delta E}$ consisting of energy levels in the original energy set $\mathcal{E}$ containing $E_c$ that are at least as far as a bandwidth $\Delta E$ from any boundaries of this set:*

$$|\widetilde{v}(E - E_c)| < v_0, \quad \text{for all } E \notin \mathcal{E}_{\Delta E} \equiv \{E : [E - \Delta E, E + \Delta E] \subseteq \mathcal{E}\}, \tag{134}$$

*then the quantum dynamical echo given by Eq. (121) (without requiring the complete positivity restriction on $v(t)$ and $w(t)$) vanishes to accuracy:*

$$\text{Tr}\left[ \hat{\gamma}_A^{shell} \left( \hat{\Pi}_A - \langle \hat{\Pi}_A \rangle_{\Sigma_d} \hat{\mathbb{1}} \right) \right] < \epsilon_A = \frac{d}{\text{Tr}[\hat{\Pi}_A]} \epsilon^2 + v_0 + v_0 w_0 + w_0. \tag{135}$$

*Proof.* The proof follows in a similar manner to that of Proposition 6.1, except that now we have to split energy level pairs into those satisfying and not satisfying the set of conditions $|E_n - E_m| < \Delta E$ and $E_n, E_m \in \Sigma_d$. This is worked out in detail in App. D.5.2. $\qquad\square$

Most noteworthy in this bound is the fact that the relationship between $\epsilon_A$ and $\epsilon$ is consistent between Theorems 7.1 and 7.2 (taking $W, V \sim 1$ and $w_0, v_0 \ll 1$):

$$\epsilon_A \sim \frac{d}{\text{Tr}[\hat{\Pi}_A]} \epsilon^2, \tag{136}$$

which shows that energy-band thermalization implies a decay of echoes to precisely the range of magnitudes necessary to establish energy-band thermalization in turn, indicating a strong interdependence[18].

Now, we can use the interdependence between the echoes defined above and energy-band thermalization on the one hand, and the results connecting energy-band thermalization to the thermalization of (almost all) physical states, to bypass energy levels and directly connect

---

[18]For formal reasons having to do with the fact that in practice, $W, V \to 1$ but $w_0, v_0 \to 0$, we cannot choose the same averaging duration and the corresponding energy window for both Theorems, which would show the exact equivalence of energy-band thermalization and decaying echoes. Instead, their interdependence is to be interpreted as follows: energy-band thermalization over a certain energy scale implies decaying echoes over a longer timescale (compared to the inverse energy scale), which in turn implies energy-band thermalization with a narrower energy scale and so on. Intuitively speaking, this is "almost" an equivalence, which we expect can be made rigorous in some limit of infinite time averages *after* first taking a thermodynamic or classical limit, analogous to equivalence statements between autocorrelators and global eigenstate thermalization in semiclassical wavefunction ergodicity theorems [40, 49].

quantum dynamical echoes to the thermalization of physical states (similar to Sec. 6). To simplify our presentation, we will not explicitly state rigorous versions of these results (to derive which is a fairly straightforward if complicated exercise), but just indicate how they may be obtained. That the decay of an echo implies the thermalization of physical states in energy shells follows by combining Theorem 7.1 with either of Theorems 5.2 and 5.3.

For the reverse direction, i.e., the thermalization of almost all physical states in an energy shell $\Sigma_d$ implying the decay of echoes, it is technically simpler to follow a somewhat roundabout route that takes advantage of our previously established results. Noting that both the thermalization and the energy band thermalization of $\hat{\Pi}_A$ on the one hand, and the corresponding properties of its projection $\hat{\Pi}_A(\Sigma_d) \equiv \hat{\Pi}_{\Sigma_d} \hat{\Pi}_A \hat{\Pi}_{\Sigma_d}$ are respectively equivalent for states restricted to the energy shell $\Sigma_d$, let us momentarily focus on the projected observable. Then, the thermalization of $\hat{\Pi}_A$ (and therefore, $\hat{\Pi}_A(\Sigma_d)$) in almost all physical (basis) states in the energy shell implies the decay of autocorrelators of $\hat{\Pi}_A(\Sigma_d)$ by Eq. (108). The decay of the autocorrelator implies the energy-band thermalization of $\hat{\Pi}_A(\Sigma_d)$, and therefore $\hat{\Pi}_A$ in $\Sigma_d$ by Proposition 6.1. By Theorem 7.2, energy-band thermalization implies the decay of echoes.

It is also evident that the combination of Theorems 7.1 and 7.2 implies that decaying echoes over some timescale implies decaying echoes over larger timescales. We then obtain the following summary for energy shell thermalization:

**Summary 7.3** (Bypassing eigenstates for energy shell thermalization o.a.). *The following implications describe the quantum thermalization o.a. of a given observable in some energy shell without reference to eigenstates:*

1. *The decay of a suitable quantum dynamical echo over a fixed timescale implies the thermalization o.a. of almost all physical states in the energy shell at all longer timescales [Theorems 7.1 and 5.3].*

2. *The decay of suitable echoes over a fixed timescale implies the decay of corresponding echoes at all longer timescales slightly exceeding the fixed timescale [Theorems 7.1 and 7.2].*

3. *The thermalization o.a. of almost all physical states in an energy shell over a fixed timescale implies the decay of a suitable quantum dynamical echo over a slightly longer timescale, if the observable is sufficiently bulky [Eq. (108), Proposition 6.1 and Theorem 7.2].*

*In particular, the decay of quantum dynamical echoes over a fixed timescale is expected to be an accessible property.*

Another aspect of thermalization such a strategy can account for is the presence of conserved quantities. In addition to energy, the thermal value may also depend on some conserved charge(s) $\hat{Q}$ that commutes with the Hamiltonian, $[\hat{H}, \hat{Q}] = 0$ (for simplicity, we will illustrate our arguments with a single conserved charge). In that case, we will have to restrict our subspace not just to e.g. an energy interval, but to an energy shell within this interval with the conserved charge in a range $\mathcal{Q}$ centered at $Q_c$ with width $\Delta Q$. Here, in addition to evolving the state with the Hamiltonian $\hat{H}$, one must also apply the symmetry transformation generated by the conserved charge, through echoes such as

$$L_{AA}(t_1, t_2; s_1, s_2) \equiv \mathrm{Tr}[e^{-i\hat{H}t_1} e^{-i\hat{Q}s_1} \hat{\rho}_A e^{i\hat{Q}s_2} e^{i\hat{H}t_2} \hat{\Pi}_A]. \tag{137}$$

Then, with a suitable weight function $\nu_{Q+}(t)$ that restricts $Q$ to the desired range of values, Theorems 7.1 and 7.2 generalize in a straightforward manner to thermalization with accessible conserved charges, as does Summary 7.3.

Here, in theory, it is crucial that $\hat{Q}$ commutes with $\hat{H}$ so that they have a shared eigenbasis. However, in an experiment, one may only be able to implement Eq. (137) with some other

operator $\hat{Q}_{\text{exp}}$ that may not strictly commute with the Hamiltonian, but has similar dynamics to $\hat{Q}$ at accessible times (see also App. A for another discussion of how approximate conserved quantities are not an obstacle in our approach). In this case, experimentally measuring $L_{AA}^{\text{exp}}$ [Eq. (137) with $\hat{Q}$ replaced by $\hat{Q}_{\text{exp}}$] should still allow a close estimation of the theoretically precise $L_{AA}$ echo, up to experimental errors, provided that the scales of $t$ and $s$ are accessible. As accessible values of $L_{AA}$ rigorously constrain energy shell thermalization (even with small errors), an experimental inability to access the precise theoretical conserved charge $\hat{Q}$ is not an obstacle for accessibility. This allows us to tackle thermal values $A_{\text{th}}(\mathcal{E}, \mathcal{Q})$ that depend on conserved charges, without requiring any sensitive knowledge of the finer properties of the energy spectrum.

# 8 Thermal equilibrium: time averages in a cloned Hilbert space

## 8.1 Cloned Hilbert spaces

So far, we have only considered the time-averaged dynamics of thermalization (with the exception of Proposition 5.4), with the promise of generalizing these results to thermal equilibrium. In this section, we will develop this generalization by showing that an observable attaining thermal equilibrium for almost all times in a given state is equivalent to the thermalization o.a. of a "cloned" observable in a "cloned state" in two copies of the Hilbert space. This is a quantum analogue of a classical result on weak mixing of product systems [6] — where a dynamical system is weak-mixing if and only if its product with itself (a doubled system) is ergodic.

This is in fact quite straightforward to show: consider a state $\hat{\rho}$ and an observable $\hat{A}$ in some Hilbert space $\mathcal{H}$. The condition for thermal equilibrium, where the state attains a thermal value $A_{\text{th}}$ for almost all times [weighted by $w(t)$], is

$$\int \mathrm{d}t \; w(t) \big( \mathrm{Tr}[\hat{\rho}(t)\hat{A}] - A_{\text{th}} \big)^2 < \epsilon, \tag{138}$$

for some small $\epsilon$. In the $D^2$-dimensional doubled Hilbert space $\mathcal{H} \otimes \mathcal{H}$, we define the following "cloned" operators[19]:

$$\hat{\rho}_2(t) \equiv \hat{\rho}(t) \otimes \hat{\rho}(t) \tag{139}$$

$$\widehat{\delta A}_2 \equiv \big( \hat{A} - A_{\text{th}}\hat{\mathbb{1}} \big) \otimes \big( \hat{A} - A_{\text{th}}\hat{\mathbb{1}} \big), \tag{140}$$

together with the trace operation, which factorizes for product operators:

$$\mathrm{Tr}_{\mathcal{H} \otimes \mathcal{H}}[A_L \otimes A_R] = \mathrm{Tr}[A_L]\,\mathrm{Tr}[A_R]. \tag{141}$$

Implicitly in the definition of $\hat{\rho}_2(t)$, we have assumed that the dynamics of the cloned Hilbert space is generated by the Hamiltonian

$$\hat{H}_2 \equiv \hat{H} \otimes \hat{\mathbb{1}} + \hat{\mathbb{1}} \otimes \hat{H}, \tag{142}$$

whose energy eigenvalues are given by $E_{\ell n} = E_\ell + E_n$ (which is at least doubly degenerate for $n \neq \ell$, as $E_{\ell n} = E_{n\ell}$), which means that the traditional thermalization results associated with

---

[19]We use the terminology "cloned" instead of mere doubling, because we specifically want to restrict these considerations to operators or states of the form $\hat{A} \otimes \hat{A}$ or $\hat{\rho} \otimes \hat{\rho}$, the latter corresponding to *cloning* the quantum state [52]. In contrast, a mere doubled Hilbert space also admits e.g. operators $\hat{A} \otimes \hat{B}$ that are not of the "cloned" form.

ETH automatically do not apply in the cloned Hilbert space; we are instead forced to work with eigenspace or energy-band thermalization. Given these cloned operators, we have the following equality:

$$\int dt \, w(t) \big( \text{Tr}[\hat{\rho}(t)\hat{A}] - A_{\text{th}} \big)^2 = \int dt \, w(t) \, \text{Tr}_{\mathcal{H}\otimes\mathcal{H}}[\hat{\rho}_2(t)\widehat{\delta A}_2]. \tag{143}$$

Therefore, the condition for the thermal equilibrium of $\hat{A}$ in the state $\hat{\rho}$ is that $\widehat{\delta A}_2$ thermalizes o.a. to 0 in the state $\hat{\rho}_2$:

$$\left| \int dt \, w(t) \, \text{Tr}_{\mathcal{H}\otimes\mathcal{H}}[\hat{\rho}_2(t)\widehat{\delta A}_2] \right| < \epsilon. \tag{144}$$

As the simplest example, let us consider cloned eigenspace thermalization for a spectrum $E_n$ that is nondegenerate for a single system. Then, a level $E_{\ell n}$ in the cloned system is doubly degenerate for $\ell \neq n$, and eigenspace thermalization [Eq. (56)] for the operator $\widehat{\delta A}_2$ gives:

$$\text{Tr}\left[ \left\{ \widehat{\delta A}_2 \hat{\Pi}(E_{\ell n}) \right\}^2 \right] < \epsilon, \tag{145}$$

for *any* pair of levels $E_n, E_\ell$ in the original system. Substituting Eq. (140), we get

$$2\left| \langle E_n|\hat{A}|E_n \rangle - A_{\text{th}} \right|^2 \left| \langle E_\ell|\hat{A}|E_\ell \rangle - A_{\text{th}} \right|^2 + 2\left| \langle E_n|\hat{A}|E_\ell \rangle \right|^4 < \epsilon^2 \tag{146}$$

The constraint on the first term gives diagonal eigenstate thermalization for $\hat{A}$, while the second term incorporates a flavor of *off-diagonal* eigenstate thermalization (even when considering only *diagonal* cloned matrix elements), conventionally associated with thermal equilibrium:

$$\left| \langle E_n|\hat{A}|E_\ell \rangle \right|^4 < \epsilon^2/2. \tag{147}$$

However, where conventional off-diagonal eigenstate thermalization [see Eq. (3)] requires these off-diagonal matrix elements to be suppressed by $e^{-S(\overline{E})/2} \sim 1/\sqrt{d}$, here if $\epsilon$ is "accessible", then we do not expect such a large suppression. Nevertheless, we will show below that this is sufficient resolution to access two *statistical* forms of thermal equilibrium over accessible timescales (as opposed to the infinitely long times required in off-diagonal eigenstate thermalization) in the setting of energy-band thermalization.

## 8.2  Cloned energy-band thermalization $\implies$ thermal equilibrium

All of the conclusions of the previous sections (qualitatively) follow for thermal equilibrium if we consider thermalization o.a. in the cloned Hilbert space. This includes updating our considerations to the energy bands and energy shells of the cloned Hilbert space. There is, however, a caveat: a narrow energy shell in $E_{mn}$ does not usually correspond to a narrow energy shell for the original system, but instead allows a full range of (pairs of) single-system energies such that their average is in the shell. In the case of global thermalization, this is not at all an obstacle: there is no restriction to energy shells. We therefore expect all the considerations of Sec. 6 to go through for global thermal equilibrium. For energy shell thermalization, however, we should make sure that we are accessing the energy shells of a single system and not the cloned system. In our view, the cleanest way to do so is to not work with an energy shell of $\mathcal{H} \otimes \mathcal{H}$, but to explicitly restrict the relevant quantities to a subspace corresponding to the cloned energy shell of a single system, $\Sigma_d \otimes \Sigma_d$.

For example, using Definition 5.1 for energy-band thermalization in the subspace $\Sigma_d \otimes \Sigma_d$ of $\mathcal{H} \otimes \mathcal{H}$ gives the criterion[20]:

$$\frac{1}{d^2} \text{Tr} \left\{ \left[ \left( \hat{\Pi}_A - \langle \hat{\Pi}_A \rangle_{\Sigma_d} \hat{\mathbb{1}} \right) \otimes \left( \hat{\Pi}_A - \langle \hat{\Pi}_A \rangle_{\Sigma_d} \hat{\mathbb{1}} \right) \right]^2_{(\Sigma_d \otimes \Sigma_d, \Delta E)} \right\}$$

$$\equiv \frac{1}{d^2} \sum_{\substack{n,m,k,\ell : E_n, E_m, E_k, E_\ell \in \Sigma_d \\ |E_n + E_\ell - E_m - E_k| < \Delta E}} \left| \langle E_n | \hat{\Pi}_A | E_m \rangle - \langle \hat{\Pi}_A \rangle_{\Sigma_d} \delta_{nm} \right|^2 \left| \langle E_\ell | \hat{\Pi}_A | E_k \rangle - \langle \hat{\Pi}_A \rangle_{\Sigma_d} \delta_{\ell k} \right|^2 < \epsilon^2, \quad (148)$$

for the form of energy-band thermalization relevant to thermal equilibrium (which we will refer to as "cloned energy-band thermalization" where the need occurs). If we consider the corresponding constraint on the terms in Eq. (146), we get (with $\delta \hat{\Pi}_A = \hat{\Pi}_A - \langle \hat{\Pi}_A \rangle_{\Sigma_d} \hat{\mathbb{1}}$)

$$\left( \frac{1}{d} \sum_{n \in \Sigma_d} \left| \langle E_n | \delta \hat{\Pi}_A | E_n \rangle \right|^2 \right)^2 + \frac{1}{d^2} \sum_{n,m \in \Sigma_d} \left| \langle E_n | \delta \hat{\Pi}_A | E_m \rangle \right|^4 < \epsilon^2. \quad (149)$$

While the constraint on the first term merely restricts eigenstate thermalization [as in Eq. (2)], the constraint on the second term is more nontrivial. First, we have the identity

$$\frac{1}{d^2} \sum_{n,m \in \Sigma_d} \left| \langle E_n | \delta \hat{\Pi}_A | E_m \rangle \right|^4 \leq \left( \frac{1}{d} \text{Tr} \left[ (\hat{\Pi}_{\Sigma_d} \delta \hat{\Pi}_A)^2 \right] \right)^2, \quad (150)$$

with the right hand side expected to be $O(1)$ for a projector. As long as $\epsilon \leq \text{Tr}[(\hat{\Pi}_{\Sigma_d} \delta \hat{\Pi}_A)^2]/d$, Eq. (149) provides a nontrivial constraint on the variance of the squared matrix elements $|\langle E_n | \delta \hat{\Pi}_A | E_m \rangle|^2$, in other words implying a degree of *uniformity* of the "distribution" of the trace $\text{Tr}[(\hat{\Pi}_{\Sigma_d} \delta \hat{\Pi}_A)^2] = \sum_{n,m \in \Sigma_d} |\langle E_n | \delta \hat{\Pi}_A | E_m \rangle|^2$ among these matrix elements (as opposed to, e.g., having only a single nonzero matrix element).

Now let us turn to thermalization dynamics. Using Theorem 5.2 with a cloned state of the form in Eq. (139) gives in place of Eq. (144) [noting that $\text{Tr}_{\mathcal{H} \otimes \mathcal{H}}[\hat{\rho}_2(t)] = \text{Tr}[\hat{\rho}]^2$ and $\langle \hat{\Pi}_A \otimes \hat{\Pi}_A \rangle = \langle \hat{\Pi}_A \rangle^2$]:

$$\left| \int dt \, w(t) \left( \text{Tr}[\hat{\rho}(t) \hat{\Pi}_A] - \langle \hat{\Pi}_A \rangle_{\Sigma_d} \right)^2 \right| < \left( \epsilon + w_0 \langle \hat{\Pi}_A \rangle_{\Sigma_d} \right) d \, \text{Tr}[\hat{\rho}^2]. \quad (151)$$

If $\hat{\rho}$ is an individual pure state, i.e. $\text{Tr}[\hat{\rho}^2] = 1$, then we need a resolution of $\epsilon \ll 1/d$ and sufficiently long times that $w_0 \ll 1/d$ to conclude that the state attains thermal equilibrium over this time scale. However, unlike Theorem 5.3, we cannot directly conclude thermal equilibrium in "almost all" basis states from here with accessible values of $\epsilon$ and $w_0$. This is because a physical basis in the cloned energy shell is dominated by states $|k\rangle \otimes |\ell\rangle$ where $k \neq \ell$, while the cloned states $|k\rangle \otimes |k\rangle$, which are of primary interest for thermal equilibrium in the single system, form a vanishingly small fraction $d/d^2$ of the basis states of the doubled Hilbert space as $d \to \infty$. Given this limitation, there are two kinds of physically relevant statements we can make from Eq. (151) with accessible values of the associated parameters.

The first such statement concerns whether expectation value fluctuations (around the thermal value) of $\hat{\Pi}_A$ in pairs of basis states are correlated over long times[21]. In analogy

---

[20]This differs from what is called "quantum mixing" in the semiclassical chaos literature [48, 49], associated with a weakly mixing classical system [5–7], which corresponds to shifting a shrinking energy band with $\Delta E \to 0$ to be centered around some $\delta E = \delta E_c$ as opposed to $\delta E = 0$. We prefer to work with cloned energy-band thermalization for general quantum systems due to its more direct implications for thermal equilibrium over a discrete set of times, while it is not clear to us if the aforementioned semiclassical notion will generalize to be similarly accessible except with a continuum of times.

[21]This is essentially the physical content of *weak-mixing* in classical ergodic theory [5–7].

1631 with the lead-up to Eq. (5.3), let us introduce a measure of time-averaged correlations of the
1632 fluctuations of $\hat{\Pi}_A$ around $\langle\hat{\Pi}_A\rangle_{\Sigma_d}$ between the $k$-th and $\ell$-th basis states $\hat{\rho}_k(0) = |k\rangle\langle k|$ and
1633 $\hat{\rho}_\ell(0) = |\ell\rangle\langle\ell|$ in the basis $\mathcal{B}$:

$$\Lambda_{k\ell}[w] \equiv \int \mathrm{d}t \; w(t)\Big(\mathrm{Tr}[\hat{\rho}_k(t)\hat{\Pi}_A] - \langle\hat{\Pi}_A\rangle_{\Sigma_d}\Big)\Big(\mathrm{Tr}[\hat{\rho}_\ell(t)\hat{\Pi}_A] - \langle\hat{\Pi}_A\rangle_{\Sigma_d}\Big) \tag{152}$$

1634 If $\Lambda_{k\ell}[w]$ is "large", we consider the fluctuations of $\hat{\Pi}_A$ to be correlated between the states $|k\rangle$
1635 and $|\ell\rangle$ over the time interval represented by $w(t)$; otherwise, they are effectively uncorrelated
1636 (for which it is sufficient that the fluctuations themselves are small, even if identical, e.g., for
1637 $k = \ell$). Now, we obtain from Eq. (151) (here, substituting for Theorem 5.2 if expressed in
1638 terms of $\hat{\rho}_{\mathcal{H}\otimes\mathcal{H}}$) and Theorem 5.3 applied to the cloned Hilbert space (noting that if $|k\rangle, |\ell\rangle \in \mathcal{B}$,
1639 then the set of product states $|k\rangle \otimes |\ell\rangle$ forms a $d^2$-element orthonormal basis for $\mathcal{H} \otimes \mathcal{H}$):

1640 **Corollary 8.1** (Cloned energy-band thermalization implies almost all pairs of physical basis
1641 states have uncorrelated expectation value fluctuations o.a.). *Let $\hat{\Pi}_A$ satisfy cloned energy-band*
1642 *thermalization with bandwidth $\Delta E > 0$ and accuracy $\epsilon > 0$ within an energy shell $\Sigma_d$ [Eq. (71)],*
1643 *and consider time-averaging with a weighting function $w(t)$ with $w_0$-bandwidth smaller than $\Delta E$,*
1644 *i.e. that satisfies [Eq. (67)]:*

$$|\widetilde{w}(\delta E)| \leq w_0, \quad \text{for all } \delta E \geq \Delta E.$$

1645 *Then, for any complete orthonormal basis of pure states $\mathcal{B}$ within the energy shell $\Sigma_d$, the fraction*
1646 *$F_\Lambda[\mathcal{B}]$ of pairs $(|k\rangle, |\ell\rangle)$ of states in which fluctuations of the expectation value of $\hat{\Pi}_A$ around its*
1647 *thermal value are correlated o.a. by at least an amount $\Lambda$, i.e. [with $\overline{\Theta}(x) = 1$ if $x$ is true and $0$*
1648 *otherwise]*

$$F_\Lambda[\mathcal{B}] \equiv \frac{1}{d^2} \sum_{|k\rangle,|\ell\rangle\in\mathcal{B}} \overline{\Theta}\big(|\Lambda_{k\ell}[w]| \geq \Lambda\big) \tag{153}$$

1649 *is constrained by:*

$$F_\Lambda[\mathcal{B}] < \frac{\sqrt{2}}{\Lambda}\Big(\epsilon + w_0\langle\hat{\Pi}_A\rangle_{\Sigma_d}\Big). \tag{154}$$

1650 The other physically relevant statement is to constrain thermal equilibrium at "almost all
1651 times" for a sufficiently mixed density operator $\hat{\rho}$. For this purpose, let us take advantage of
1652 the fact that $w(t) \geq 0$ (by assumption), and the integrand on the left hand side of Eq. (151) is
1653 nonnegative as well. Then, for some $0 < \kappa < 1$ (with the expectation that $\kappa < 1$ is small, but
1654 not vanishingly small), the set of times $W_{\mathrm{ex}}[\kappa] \subseteq \mathbb{R}$, in which

$$\Big(\mathrm{Tr}\big[\hat{\rho}(t \in W_{\mathrm{ex}}[\kappa])\,\hat{\Pi}_A\big] - \langle\hat{\Pi}_A\rangle_{\Sigma_d}\Big)^2 \geq \frac{\epsilon + w_0\langle\hat{\Pi}_A\rangle_{\Sigma_d}}{\kappa} d\,\mathrm{Tr}[\hat{\rho}^2], \tag{155}$$

1655 i.e., the state $\hat{\rho}$ does not attain thermal equilibrium to a weaker accuracy i.e. larger by a
1656 constant $1/\kappa$, is constrained by

$$\int_{t\in W_{\mathrm{ex}}[\kappa]} \mathrm{d}t \; w(t) < \kappa, \tag{156}$$

1657 due to Eq. (151), as the integrand at these times is at least given by the right hand side of
1658 (155), and at the remaining times cannot be less than zero. At all other times, the mixed state
1659 $\hat{\rho}$ attains thermal equilibrium to accuracy:

$$\Big|\mathrm{Tr}\big[\hat{\rho}(t \notin W_{\mathrm{ex}}[\kappa])\,\hat{\Pi}_A\big] - \langle\hat{\Pi}_A\rangle_{\Sigma_d}\Big| < \sqrt{\frac{\epsilon + w_0\langle\hat{\Pi}_A\rangle_{\Sigma_d}}{\kappa}}\sqrt{d\,\mathrm{Tr}[\hat{\rho}^2]}. \tag{157}$$

1660  We therefore see that to ensure that $\hat{\rho}$ attains thermal equilibrium to some small but finite
1661  accuracy for a large weighted fraction $(1 - \kappa)$ of times, we must establish cloned energy-band
1662  thermalization to accuracy $\epsilon \ll \kappa$, and wait for sufficiently large times that $w_0 \ll \kappa / \langle \hat{\Pi}_A \rangle_{\Sigma_d}$, as
1663  long as $\mathrm{Tr}[\hat{\rho}^2] = O(1/d)$.

1664      Now, let us apply Eq. (157) to an orthonormal basis $\mathcal{B}$, by considering an initial ensemble
1665  of basis states with some probability distribution $(p_k \in [0,1])_{k \in \mathbb{Z}_d}$:

$$\hat{\rho}(0) = \sum_k p_k |k\rangle\langle k|. \tag{158}$$

1666  Normalizing this distribution to $\sum_k p_k = 1$, the "size" of the ensemble may be measured by
1667  its participation fraction $\mu(\rho) = 1/(d \sum_k p_k^2) \in [1/d, 1]$, which estimates what fraction of the
1668  $d$ basis states are significantly represented in this distribution. As per Eq. (78), $\mu(\rho)$ can be
1669  thought of as the size of a hypothetical "phase space volume" associated with this ensemble, if
1670  one wishes to draw an analogy with classical systems. Then we get thermal equilibration for
1671  sufficiently large ensembles:

1672  **Corollary 8.2** (Cloned energy-band thermalization implies the thermal equilibration of suffi-
1673  *ciently large ensembles of physical states at almost all times). Let $\hat{\Pi}_A$ satisfy cloned energy-band*
1674  *thermalization with bandwidth $\Delta E > 0$ and accuracy $\epsilon > 0$ within an energy shell $\Sigma_d$ [Eq. (148)],*
1675  *and consider time-averaging with a weighting function $w(t)$ with $w_0$-bandwidth smaller than $\Delta E$,*
1676  *i.e. that satisfies [Eq. (67)]:*

$$|\widetilde{w}(\delta E)| \le w_0, \quad \text{for all } \delta E \ge \Delta E.$$

1677  *Then, for any constant $0 < \kappa < 1$, there exists a (possibly empty) set of times $W_{ex}[\kappa]$ spanning at*
1678  *most a small weighted fraction of available times [Eq. (156)]:*

$$\int_{t \in W_{ex}[\kappa]} \mathrm{d}t \, w(t) < \kappa,$$

1679  *such that at any time $t \notin W_{ex}[\kappa]$ outside this set, for any complete orthonormal basis of pure states*
1680  *$\mathcal{B}$ within the energy shell $\Sigma_d$, any ensemble $(p_k, |k\rangle)$ of basis states $|k\rangle$ with respective probabilities*
1681  *$p_k \in [0,1]$ such that $\sum_k p_k = 1$ attains thermal equilibrium to accuracy $\lambda$:*

$$\left| \sum_k p_k \langle k(t)| \hat{\Pi}_A |k(t)\rangle - \langle \hat{\Pi}_A \rangle_{\Sigma_d} \right| < \lambda, \text{ for all } t \notin W_{ex}[\kappa], \tag{159}$$

1682  *provided that the distribution has a sufficiently large participation fraction:*

$$\mu(\rho) \equiv \frac{1}{d \sum_k p_k^2} \ge \frac{\epsilon + w_0 \langle \hat{\Pi}_A \rangle_{\Sigma_d}}{\kappa \lambda^2}. \tag{160}$$

1683      We see once again that the only way to make this statement (equilibration at almost all
1684  times) apply to individual basis states (i.e., for $\mu(\rho) = 1/d$) is to have $\epsilon, w_0 \ll 1/d$. It appears
1685  unlikely that we can make stronger statements regarding thermal equilibrium for an individual
1686  state with accessible resources. For example, while it has been shown (even accounting for
1687  degeneracies, though in apparently inaccessible ways) that the equilibration of *all* observables
1688  can occur at almost all times in special classes of initial states (with a large energy spread) [39],
1689  this typically requires a time scale of $T \gtrsim d \ln d$ to establish equilibration, which corresponds to
1690  inaccessible values of $w_0$ (e.g., for a uniform time average over an interval $T$, this corresponds to
1691  $w_0 \sim 1/(d \ln d)$, even smaller than what is required to constrain individual states in Eq. (160)).

We are not aware of stronger results on equilibration elsewhere in the literature, but it would be interesting to explore if additional assumptions in specific classes of systems can sharpen these results on equilibration.

It also appears that this limitation to ensembles of states (for accessible results) cannot be circumvented without losing some generality because we also expect our results to have a reasonable classical limit, and many natural observables in individual trajectories of classical systems can never equilibrate[22]. To see this explicitly, consider a classical (e.g. Ising) model of $N \to \infty$ two-level particles with a local "spin" observable $s_k \in \{0, 1\}$ (which may also be obtained by coarse-graining a continuous [or other] degree of freedom into two equal-sized regions in the phase space labeled by 0 and 1). The (microcanonical) thermal value of $s_k$ is $\langle s_k \rangle = 1/2$ (the phase space average of its two values), which can never be attained in an individual trajectory as it is not among the allowed values of $s_k$, making thermal equilibration impossible in a single state (but intensive observables such as $\sum_k s_k / N$ can equilibrate by being thermal in almost all classical states by default, irrespective of dynamics [9]). However, an ensemble of initial states that dynamically evolves to have an equal distribution of $s_k$ in the 0 and 1 states at some long time can equilibrate. Indeed, Eq. (160) is analogous to the classical statement that mixing [5–7] (loosely, corresponding to thermal equilibrium at all long times) can only be shown for statistical ensembles of states $A$ with nonzero measure $\mu(A) \neq 0$, corresponding here to $\mu(\rho) = \Theta(1)$ as $d \to \infty$.

In summary, given cloned energy-band thermalization, we can show that (1) almost all pairs of states have uncorrelated expectation value fluctuations around the thermal value over accessible timescales, and (2) *all* sufficiently large ensembles of states attain thermal equilibrium at almost all accessible times. Now we turn to the question of how to establish cloned energy-band thermalization, and consequently the thermal equilibration of almost all physical states at almost all times, from echoes.

## 8.3 Cloned energy-band thermalization from quantum dynamical echoes

Following the strategy of Sec. 7, we would like to access cloned energy-band thermalization using suitable quantum dynamical echoes. To restrict these echoes to energy shells of a system rather than the cloned Hilbert space, we choose two different echo averaging functions $v_{L+}(t)$ and $v_{R+}(t)$ for the cloned version $\hat{\Gamma}_A^{\text{shell}}$ of the quantum dynamical echo $\hat{\gamma}_A^{\text{shell}}$ in Eq. (120) (where we have transformed to new time variables for convenience; in our original setting, their analogues are obtained by $t_1 = t + \delta t$ and $t_2 = t - \delta t$):

$$\hat{\Gamma}_A^{\text{shell}} \equiv \frac{1}{\left(\text{Tr}[\hat{\Pi}_A]\right)^2} \int dt \int 2 \, d\delta t_L \int 2 \, d\delta t_R \, \left\{ w_+(t) v_{L+}(2\delta t_L) v_{R+}(2\delta t_R) e^{-2iE_c(\delta t_L + \delta t_R)} \right.$$
$$\left. \left[ e^{-i\hat{H}(t+\delta t_L)} \left( \hat{\Pi}_A - \langle \hat{\Pi}_A \rangle_{\Sigma_d} \hat{\mathbb{1}} \right) e^{i\hat{H}(t-\delta t_L)} \right] \otimes \left[ e^{-i\hat{H}(t+\delta t_R)} \left( \hat{\Pi}_A - \langle \hat{\Pi}_A \rangle_{\Sigma_d} \hat{\mathbb{1}} \right) e^{i\hat{H}(t-\delta t_R)} \right] \right\}.$$
$$(161)$$

---

[22]While equilibration in almost all pure states may "almost" follow from Eq. (151) for generic quantum systems along the lines of Theorem 5.3, the obstacle for a rigorous approach is that when splitting basis states into those of positive and negative deviations at a given time $t_0$ as in the proof of this Theorem, Eq. (157) may apply to each such subset with a *different* set of exceptional times, possibly including the original time $t_0$ of consideration, therefore ruling out thermal equilibrium at $t_0$ for these smaller subsets. This may not be the case for generic fully quantum systems (in which case it is possible that suitable assumptions that imply equilibration in almost all individual pure states may be identified), but we expect this to be the case if the dynamics is effectively classical (which would imply no equilibration in general in individual states). In particular, it would be interesting to consider if there are any assumptions that assume only an *accessible* range of parameters that can distinguish between the classical case with statistical thermalization and the fully quantum case with thermalization in individual pure states; of course, with (inaccessible) sensitivity to the dimension $d$ of the energy shell, this may easily be done within our formalism as discussed after Eq. (151) and Corollary 8.2.

To see why this should work, we can write the expectation value of the cloned version of $(\hat{\Pi}_A - \langle \hat{\Pi}_A \rangle_{\Sigma_d} \hat{\mathbb{1}})$ in $\hat{\Gamma}_A^{\text{shell}}$ in the energy eigenbasis :

$$
\mathrm{Tr}_{\mathcal{H} \otimes \mathcal{H}} \left[ \hat{\Gamma}_A^{\text{shell}} \left( \hat{\Pi}_A - \langle \hat{\Pi}_A \rangle_{\Sigma_d} \hat{\mathbb{1}} \right) \otimes \left( \hat{\Pi}_A - \langle \hat{\Pi}_A \rangle_{\Sigma_d} \hat{\mathbb{1}} \right) \right]
$$
$$
= \frac{1}{(\mathrm{Tr}[\hat{\Pi}_A])^2} \sum_{n,m} \left\{ \widetilde{w}_+(E_n + E_\ell - E_m - E_k) \widetilde{v}_{L+} \left( \frac{E_n + E_m}{2} - E_c \right) \widetilde{v}_{R+} \left( \frac{E_k + E_\ell}{2} - E_c \right) \right.
$$
$$
\left. \left| \langle E_m | \left( \hat{\Pi}_A - \langle \hat{\Pi}_A \rangle_{\Sigma_d} \hat{\mathbb{1}} \right) | E_n \rangle \right|^2 \left| \langle E_k | \left( \hat{\Pi}_A - \langle \hat{\Pi}_A \rangle_{\Sigma_d} \hat{\mathbb{1}} \right) | E_\ell \rangle \right|^2 \right\}. \tag{162}
$$

The matrix elements of the autocorrelators relevant for global thermal equilibrium can be recovered from this expression by setting $v_{L+}(t) = v_{L-}(t) = \delta(t)$. Otherwise, for energy shell thermal equilibrium, we can focus on energy shells in which both $\widetilde{v}_{L+}(E - E_c)$ and $\widetilde{v}_{R+}(E - E_c)$ are larger than some $V > 0$ (or, in the simplest case, chose $v_{L+} = v_{R+} = v_+$). Given that Eq. (162) is the echo of interest, it is straightforward to generalize Theorem 7.1 to its cloned variant. In particular, given that

$$
\mathrm{Tr}_{\mathcal{H} \otimes \mathcal{H}} \left[ \hat{\Gamma}_A^{\text{shell}} \left( \hat{\Pi}_A - \langle \hat{\Pi}_A \rangle_{\Sigma_d} \hat{\mathbb{1}} \right) \otimes \left( \hat{\Pi}_A - \langle \hat{\Pi}_A \rangle_{\Sigma_d} \hat{\mathbb{1}} \right) \right] < \epsilon_A, \tag{163}
$$

we get in place of Eq. (131):

$$
\frac{1}{d^2} \mathrm{Tr} \left\{ \left[ \left( \hat{\Pi}_A - \langle \hat{\Pi}_A \rangle_{\Sigma_d} \hat{\mathbb{1}} \right) \otimes \left( \hat{\Pi}_A - \langle \hat{\Pi}_A \rangle_{\Sigma_d} \hat{\mathbb{1}} \right) \right]^2_{(\Sigma_d \otimes \Sigma_d, \Delta E)} \right\} < \epsilon^2 = \frac{(\mathrm{Tr}[\hat{\Pi}_A])^2}{W V^2 d^2} \epsilon_A, \tag{164}
$$

corresponding to cloned energy-band thermalization as in Eq. (148).

   For immediate future reference, we also note that the echo in Eq. (162) can be constructed in terms of correlations between the same quantities $L_{AA}(t_1, t_2)$, $L_A(t)$, and $L_{\mathcal{H}}(t)$ that appeared for thermalization o.a. in Sec. 7.1:

$$
\mathrm{Tr}_{\mathcal{H} \otimes \mathcal{H}} \left[ \hat{\Gamma}_A^{\text{shell}} \left( \hat{\Pi}_A - \langle \hat{\Pi}_A \rangle_{\Sigma_d} \hat{\mathbb{1}} \right) \otimes \left( \hat{\Pi}_A - \langle \hat{\Pi}_A \rangle_{\Sigma_d} \hat{\mathbb{1}} \right) \right]
$$
$$
= \int \mathrm{d}t \int 2 \, \mathrm{d}\delta t_L \int 2 \, \mathrm{d}\delta t_R \left\{ w_+(t) v_{L+}(2\delta t_L) v_{R+}(2\delta t_R) e^{-2iE_c(\delta t_L + \delta t_R)} \right.
$$
$$
\left[ L_{AA}(t + \delta t_L, t - \delta t_L) - 2 \langle \hat{\Pi}_A \rangle_{\Sigma_d} L_A(2\delta t_L) + \frac{\langle \hat{\Pi}_A \rangle_{\Sigma_d}^2}{\langle \hat{\Pi}_A \rangle} L_{\mathcal{H}}(2\delta t_L) \right]
$$
$$
\left. \left[ L_{AA}(t + \delta t_R, t - \delta t_R) - 2 \langle \hat{\Pi}_A \rangle_{\Sigma_d} L_A(2\delta t_R) + \frac{\langle \hat{\Pi}_A \rangle_{\Sigma_d}^2}{\langle \hat{\Pi}_A \rangle} L_{\mathcal{H}}(2\delta t_R) \right] \right\} \tag{165}
$$

It follows that performing measurements of these quantities in a single copy of the system is sufficient to constrain thermal equilibrium as well; the only difference is in the more complicated integrals, which may in any case be evaluated classically. The question of how to perform these measurements will occupy our interest in the next section.

# 9   Discussion: A sketch of experimental measurement protocols

In Secs. 6, 7, and 8, we have shown that the thermalization (both on average, and for thermal equilibrium) of an observable $\hat{\Pi}_A$ in any set of physical basis states, over finite time scales in

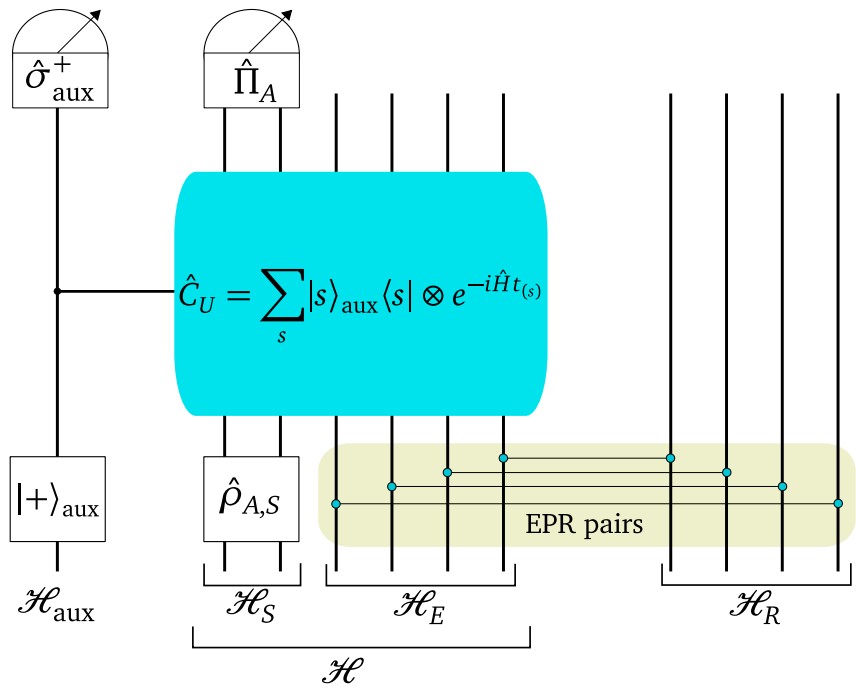

Figure 6: Schematic depiction of the experimental protocol to determine quantum thermalization over accessible timescales described in Sec. 9, via a quantum circuit diagram where each vertical line is the worldline of, e.g., a qubit, and time flows upward. While the depiction here assumes a measurement of interference effects relevant for quantum dynamical echoes in including the auxiliary qubit $\mathcal{H}_{\text{aux}}$, we note that the auxiliary qubit is not necessary for global thermalization which requires only autocorrelators. As all the relevant measurements $\hat{\Pi}_A$ and $\hat{\sigma}_{\text{aux}}^+$ are only carried out in the subsystem of interest $\mathcal{H}_S$ or the auxiliary qubit $\mathcal{H}_{\text{aux}}$, while only $\mathcal{H}_E$ and $\mathcal{H}_R$ are usually thermodynamically large systems, we expect all measurements to be completely accessible provided the dynamics of the Hamiltonian $\hat{H}$ can be implemented (for echoes, via the controlled operation $\hat{C}_U$). Our emphasis here is that a relatively simple, inexpensive protocol (in terms of the complexity of quantum measurements) within accessible timescales can conclusively determine the thermalization of a few-body observable, to within any experimental accuracy, in almost all physically relevant (or irrelevant) initial states.

energy shells can be accessed through measurements of the following quantities (repeated here for convenience):

$$L_{AA}(t_1, t_2) \equiv \text{Tr}[e^{-i\hat{H}t_1}\hat{\rho}_A e^{i\hat{H}t_2}\hat{\Pi}_A], \tag{166}$$

$$L_A(t) \equiv \text{Tr}[e^{-i\hat{H}t}\hat{\rho}_A], \tag{167}$$

$$L_{\mathcal{H}}(t) \equiv \frac{1}{D}\text{Tr}[e^{-i\hat{H}t}]. \tag{168}$$

Other details such as the energy band and shells of interest (however, with widths constrained by the range of times over which the above quantities have been measured) and the corresponding thermal values may be determined by classical post-processing. These measurements correspond (in the first two cases) to the dynamics of $\hat{\Pi}_A$ in the single initial state:

$$\hat{\rho}_A \equiv \frac{\hat{\Pi}_A}{\text{Tr}[\hat{\Pi}_A]}. \tag{169}$$

1750  Our interest in this section is to demonstrate the theoretical feasibility of such measurements
1751  with resources that scale at most linearly in the system size $N$ (for few-body observables). We
1752  do not perform a detailed analysis of additional $O(1)$ overheads and classical computation
1753  costs that may be incurred by the time averaging procedure, and leave a concrete experimental
1754  proposal in specific platforms with a rigorous, quantitative analysis of the measurement budget
1755  for future work. A schematic of the protocol discussed here is illustrated in Fig. 6.

**Setup: Few-body observables in a many-body system**

1757  For definiteness, let us consider a system of $N$ qubits, with Hilbert space $\mathcal{H}$ of dimension $D = 2^N$.
1758  The Hamiltonian $\hat{H}$ is implemented on this set of qubits. To define the observable $\hat{\Pi}_A$, say, to
1759  correspond to a few-body operator, it is convenient to select a subsystem $\mathcal{H}_S$ of $N_S$ qubits with
1760  dimension $D_S = 2^{N_S}$ (with the remaining $N_E = N - N_S$ qubits comprising $\mathcal{H}_E$ of dimension
1761  $D_E = 2^{N_E}$). We will take $\hat{\Pi}_A$ to project onto some state $|a\rangle_S \in \mathcal{H}_S$:

$$\hat{\Pi}_A = |a\rangle_S \langle a| \otimes \hat{\mathbb{1}}_E. \tag{170}$$

1762  This corresponds to a very general setting of a few-body observable in an interacting many-body
1763  system. For convenience, where questions of accessibility are concerned, we will assume that
1764  $N_S = O(1)$, so that $N_E = \Theta(N)$.
1765      For example, given the computational basis states $|\{s_j\}_{j=1}^N\rangle$ of the $N$ qubits (where $s_j \in \{0, 1\}$),
1766  $|a\rangle_S$ could be any computational basis state of the chosen $N_S$-qubit subsystem, with the state
1767  of the remaining qubits being ignored. In this case, $\hat{\Pi}_A$ may be measured via projective mea-
1768  surements in the computational basis of $\mathcal{H}_S$, while not performing any measurements on
1769  $\mathcal{H}_E$.

**State preparation**

1771  The initial state in Eq. (169), due to Eq. (170), is given by:

$$\hat{\rho}_A = |a\rangle_S \langle a| \otimes \frac{\hat{\mathbb{1}}_E}{D_E} \tag{171}$$

1772  To prepare this state, one can initialize $\mathcal{H}_S$ e.g. in the computational basis state $|a\rangle_S$, which we
1773  assume to be a straightforward operation in a many-qubit system that may be achieved with
1774  $O(1)$ costs. However, we must prepare $\mathcal{H}_E$ in the maximally mixed state.
1775      A simple strategy to achieve the maximally mixed state [52] is to maximally entangle
1776  $\mathcal{H}_E$ with a duplicate subsystem $\mathcal{H}_R$, also of (at least) $N_E$ qubits[23]. This can be done e.g. by
1777  maximally entangling each qubit in $\mathcal{H}_E$ (say, indexed by $j$) with a corresponding qubit in $\mathcal{H}_R$,
1778  thereby generating an EPR pair [52] of these qubits; as entangling any pair of qubits requires
1779  only $O(1)$ steps (e.g., a Hadamard gate followed by a CNOT gate on each pair initialized to
1780  the $|00\rangle$ state), we expect the preparation of the maximally mixed state $\hat{\mathbb{1}}_E/D_E$ to require
1781  only $O(N_E)$ gates. In fact, we expect this step to be the most expensive system-independent
1782  step in our protocol in terms of the scaling of gate complexity with system size (except for
1783  implementing the actual dynamics, which depends on the Hamiltonian $\hat{H}$). As all of these
1784  maximally entangled pairs can be generated in parallel, the time complexity of this step remains
1785  $O(1)$.
1786      We also note that one way to prepare the state $|a\rangle_S$ in $\mathcal{H}_S$ is to prepare the maximally mixed
1787  state in $\mathcal{H}_S$ as well, by entanglement with $N_S$ additional qubits in a duplicate subsystem, and
1788  then perform the projective measurement $\hat{\Pi}_A$ and postselect on the outcome (which requires
1789  $O(1)$ measurements as $N_S = O(1)$).

---

[23]In this case, the setting is similar to one of relevance to the fast scrambling problem [55].

## Autocorrelator measurements

If one is only concerned with global thermalization, the measurement of the autocorrelator

$$\text{Tr}[\hat{\rho}_A(t)\hat{\Pi}_A] = \text{Tr}[e^{-i\hat{H}t}\hat{\rho}_A e^{i\hat{H}t}\hat{\Pi}_A] = L_{AA}(t,t), \tag{172}$$

relevant for Sec. 6, is extremely straightforward. We prepare the system in the initial state $\hat{\Pi}_A$, evolve it with the Hamiltonian $\mathcal{H}$, and measure $\hat{\Pi}_A$ after a time $t$ — for example, the probability of the outcome $|a\rangle_S$ in the setting of computational basis states. Typically, we expect that the autocorrelator takes values of the order of $1/D_S$ or larger (e.g., this is comparable to the expectation value of the projector in the maximally mixed state). Thus, the number of measurements required to measure a subsystem return probability of this size scales with $D_S$, which we have assumed to be $O(1)$.

As entanglement with the external qubits in $\mathcal{H}_R$ ensures that our system $\mathcal{H}$ is genuinely in the mixed quantum state $\hat{\rho}_A$ rather than a classical ensemble of states amounting to $\hat{\rho}_A$, we do not have to worry about whether any finite classical ensemble is representative of the full ensemble (which may be a concern in the classical case, as discussed in Secs. 1 and 3). Statistically, the expectation value of $\hat{\Pi}_A$ should automatically converge to that in $\hat{\rho}_A$ over a large number of experimental measurements.

## Echo measurements

Now, let us consider the measurement of echoes. The echo involves the *trace* of a quantum mechanical operator, which is not as straightforward to measure as a return probability. We are aware of two measurement frameworks, which can be implemented in present-day experimental platforms, that can measure the squared magnitude of a trace — specifically, the spectral form factor $K(t) = |L_{\mathcal{H}}(t)|^2$ [see also Eq. (126)]. The first is by implementing controlled dynamics with an auxiliary qubit [22], and the second by using randomized measurements [23]. As the former is conceptually simpler, and also related to phase measurement algorithms [52], we will describe measurements of echoes using an auxiliary qubit, generalizing the discussion in Ref. [22]. It may also be possible to adapt other measurement strategies free of auxiliary qubits [80], provided there are more intrinsic ways to identify a preferred "zero" of energy $E_c = 0$ in the system.

Let $\mathcal{H}_{\text{aux}}$ be the Hilbert space of the auxiliary qubit, which we initialize in the state

$$|+\rangle_{\text{aux}} = \frac{1}{\sqrt{2}} \left( |0\rangle_{\text{aux}} + |1\rangle_{\text{aux}} \right). \tag{173}$$

Now, consider applying a controlled operator to the combined system and auxiliary qubit $\mathcal{H} \otimes \mathcal{H}_{\text{aux}}$, with the general form

$$\hat{C}_U = \hat{U}_0 \otimes |0\rangle_{\text{aux}}\langle 0| + \hat{U}_1 \otimes |1\rangle_{\text{aux}}\langle 1|, \tag{174}$$

where $\hat{U}_0$ and $\hat{U}_1$ (which are not necessarily unitary) act on the system $\mathcal{H}$ alone. Then, if the system $\mathcal{H}$ is in some initial state $\hat{\rho}$, the final state of $\mathcal{H}_{\text{aux}}$ is given by the reduced density operator:

$$\hat{\rho}_{\text{aux}}[\hat{C}_U] = \text{Tr}_{\mathcal{H}}\left[ \hat{C}_U \hat{\rho} \otimes |+\rangle_{\text{aux}}\langle +|\hat{C}_U^\dagger \right] = \frac{1}{2} \sum_{r,s \in \{0,1\}} \text{Tr}_{\mathcal{H}}[\hat{U}_r \hat{\rho} \hat{U}_s^\dagger] \, |r\rangle_{\text{aux}}\langle s|. \tag{175}$$

It follows that the expectation value of $\sigma_{\text{aux}}^+ \equiv |1\rangle_{\text{aux}}\langle 0|$ in this final state measures the echo of $\hat{U}_0$ and $\hat{U}_1$ in the state $\hat{\rho}$:

$$2\langle \sigma_{\text{aux}}^+ \rangle = 2\,\text{Tr}[\hat{\rho}_{\text{aux}}[\hat{C}_U]\sigma_{\text{aux}}^+] = \text{Tr}_{\mathcal{H}}[\hat{U}_0 \hat{\rho} \hat{U}_1^\dagger]. \tag{176}$$

As this is not a Hermitian operator, in practice, its expectation value can be generated using

$$2\langle\sigma^+_{\text{aux}}\rangle = \langle\sigma^x_{\text{aux}}\rangle + i\langle\sigma^y_{\text{aux}}\rangle, \tag{177}$$

where $\sigma^x_{\text{aux}} = |0\rangle\langle1| + |1\rangle\langle0|$ and $\sigma^y_{\text{aux}} = -i|0\rangle\langle1| + i|1\rangle\langle0|$ are the usual Hermitian Pauli spin operators, whose measurements we assume to be straightforward in the relevant experimental platforms.

In the above setting, the echoes of Eqs. (166), (167), and (168) can be obtained as follows:

1. For $L_{AA}(t_1, t_2)$, we set $\hat{\rho} = \hat{\rho}_A$, and

$$\hat{U}_0 = \hat{\Pi}_A e^{-i\hat{H}t_1}, \tag{178}$$

corresponding to evolution for a time $t_1$ followed by the measurement $\hat{\Pi}_A$, and

$$\hat{U}_1 = \hat{\Pi}_A e^{-i\hat{H}t_2}, \tag{179}$$

corresponding to evolution for a time $t_2$ followed by the measurement $\hat{\Pi}_A$. While it is not strictly necessary for the measurement $\hat{\Pi}_A$ to be carried out in both the auxiliary $|0\rangle_{\text{aux}}$ and $|1\rangle_{\text{aux}}$ branches of Eq. (175), doing so eliminates the need to implement a *controlled* measurement. This is because $\hat{C}_U$ can then be written as:

$$\hat{C}_U = \hat{\Pi}_A \otimes \hat{\mathbb{1}}_{\text{aux}} \left( e^{-i\hat{H}t_1} \otimes |0\rangle_{\text{aux}}\langle0| + e^{-i\hat{H}t_2} \otimes |1\rangle_{\text{aux}}\langle1| \right), \tag{180}$$

which only requires the implementation of controlled Hamiltonian dynamics (which implicitly sets a 0 value of the energy $E$ as the one that causes no phase changes in the auxiliary qubit, relative to which we must choose the center $E_c$ of the energy shell of interest), with the measurement of $\hat{\Pi}_A$ being performed just on $\mathcal{H}$ subsequent to the evolution. One way to think of such a controlled operation with different times is, e.g., if $t_2 > t_1 > 0$, then one could implement the unitary $\exp[-i\hat{H}t_1]$ on $\mathcal{H}$ without any control, then apply a controlled unitary $\exp[-i\hat{H}(t_2 - t_1)] \otimes |1\rangle_{\text{aux}}\langle1|$ only for the remaining duration $(t_2 - t_1)$. Additionally, if reverse time evolution is not feasible for a choice of $t_1 < 0$ or $t_2 < 0$, we may move the corresponding dynamics to the other term to ensure that strictly positive evolution; for example, we may also set (if $t_2 < 0$)

$$\hat{U}_0 = e^{-i\hat{H}(-t_2)}\hat{\Pi}_A e^{-i\hat{H}t_1}, \ \hat{U}_1 = \hat{\Pi}_A. \tag{181}$$

If both times have to be reversed, an easy option is to take the complex conjugate of the trace with $t_1, t_2 > 0$ to effectively obtain $t_1, t_2 < 0$ without additional measurements.

2. Similarly, for $L_A(t)$, we may set $\hat{\rho} = \hat{\rho}_A$, and

$$\hat{U}_0 = e^{-i\hat{H}t}, \ \hat{U}_1 = \hat{\mathbb{1}}, \tag{182}$$

for example. Alternatively, we may also prepare the initial state $\hat{\rho} = \hat{\mathbb{1}}/D$, the maximally mixed state in $\mathcal{H}$, and include projectors $\hat{\Pi}_A$ in either or both of $\hat{U}_{0,1}$. In addition to preparing the maximally mixed state in $\mathcal{H}_E$, this would require $O(N_S)$ additional steps with $N_S$ additional qubits to prepare $\mathcal{H}_S$ in a maximally mixed state in this alternative procedure.

3. For $L_{\mathcal{H}}(t)$, we prepare the maximally mixed state $\hat{\rho} = \hat{\mathbb{1}}/D$ in $\mathcal{H}$, and only implement the dynamics without any measurements in $\mathcal{H}$ (consequently, the only measurement performed is on the auxiliary system $\mathcal{H}_{\text{aux}}$), for example with

$$\hat{U}_0 = e^{-i\hat{H}t}, \ \hat{U}_1 = \hat{\mathbb{1}}. \tag{183}$$

1857    This is essentially the proposal for spectral form factors in Ref. [22], where $|L_\mathcal{H}(t)|^2$
1858    requires a measurement of $\langle \sigma^x_{\text{aux}} \rangle^2 + \langle \sigma^y_{\text{aux}} \rangle^2$, but the spectral function $L_\mathcal{H}(t)$ itself is
1859    obtained by measuring $\langle \sigma^+_{\text{aux}} \rangle$ via Eq. (177).

1860    Additionally, to account for conserved charges through echoes such as in Eq. (137), we
1861    should implement an additional controlled unitary (for each conserved charge $Q$) of the form:

$$\hat{C}_Q = e^{-i\hat{Q}s_1} \otimes |0\rangle_{\text{aux}}\langle 0| + e^{-i\hat{Q}s_2} \otimes |1\rangle_{\text{aux}}\langle 1|. \tag{184}$$

1862    The preceding discussion generalizes directly to this case. We recall from the discussion
1863    following Eq. (137) that it is sufficient for the experimentally accessible version of the charge
1864    $\hat{Q}$ to closely follow the dynamics of the actual conserved charge (that precisely commutes with
1865    $\hat{H}$) for accessible timescales.

1866    We emphasize that except for the implementation of controlled dynamics, there is no specific
1867    aspect of these measurements that intrinsically scale with the system size, except for state
1868    preparation. The number of measurements required will be determined by the desired accuracy
1869    $\epsilon_A$ of the echoes, which we expect to scale with $N_S$ in most practical situations (assumed here to
1870    be $O(1)$) rather than $N_E$. For an example with a concrete proposal for implementing controlled
1871    Hamiltonian dynamics, specifically in Rydberg atoms with an auxiliary atomic clock qubit, see
1872    Ref. [22].

1873    Additionally, in implementations where the qubits in $\mathcal{H}$ are each part of a larger local
1874    Hilbert space (e.g., 2-level subspaces of 4-level systems), it is possible to directly "upgrade" any
1875    isolated Hamiltonian evolution $e^{-i\hat{H}t}$ of the qubits to a variant $\hat{C}_U$ controlled by an external
1876    qubit, without any explicit knowledge of $\hat{H}$ or interfering with the isolated dynamics of qubits.
1877    This may be done by rapidly applying simpler controlled operations that swap each local
1878    qubit subspace with an orthogonal local subspace that is not subject to any time evolution,
1879    before and after the Hamiltonian dynamics of qubits occurs for the desired length of time over
1880    which it must be controlled [81]. This may further allow the implementation of such echo
1881    measurements in experimental platforms where controlled variants of a given Hamiltonian $\hat{H}$
1882    cannot be "natively" implemented.

### Sampling discrete times

1884    In an experiment, one expects to sample only a discrete set of times $t_j$, with weight functions
1885    such as $v(t)$ [or $w(t)$] taking the form:

$$v(t) = \sum_j v(t_j)\delta(t - t_j). \tag{185}$$

1886    If the $t_j$ form a regular lattice of spacing $\delta t$, i.e. $t_j - t_k = \delta t(j - k)$, then there is a potential
1887    complication in the energy domain, because $\tilde{v}(E)$ becomes periodic with period $2\pi/\delta t$:

$$\tilde{v}\left(E + \frac{2\pi\ell}{\delta t} - E_c\right) = \tilde{v}(E - E_c), \quad \text{for } \ell \in \mathbb{Z}. \tag{186}$$

1888    This periodicity may be an issue to the extent that it may include contributions from the matrix
1889    elements of observables outside an energy shell or energy band of interest as $\tilde{v}(2\pi\ell/\delta t) = 1$,
1890    and therefore, one ends up selecting several energy shells centered around the different energies
1891    $E_c + 2\pi\ell/\delta t$. We particularly expect this to be an issue for energy shells via $v(t)$, due the
1892    thermal value $\langle \hat{\Pi}_A \rangle_{\Sigma_d}$ potentially being significantly different between energy shells, but similar
1893    issues may often occur with $w(t)$ as well due to contributions from different energy bands,
1894    which can be accounted for in the below discussion by replacing $\tilde{v}(E - E_c)$ with $w(\delta E)$.

For discrete-time systems whose time step is $\Delta t_{\text{step}} = \delta t$, no effective additional contributions exist because the (quasi-)energy spectrum itself is periodic with the same period. However, even in these systems, one may wish to sample the dynamics sparsely over widely spaced instants of time $\delta t \gg \Delta t_{\text{step}}$ to reduce the amount of data collected, where this periodicity would become an issue. In either case (continuous time systems or sparse sampling in discrete time systems), the effect of additional contributions from outside the energy band or shell of interest may be reduced by taking $\delta t \to 0$ (i.e. sampling more points), but we point out that there is a significantly more efficient way to reduce these contributions.

If the $t_j$ are sampled *irregularly* with average spacing $\delta t$ according to some sufficiently random statistical (e.g. Poisson) process, then $\widetilde{v}(E)$ instead becomes quasiperiodic, with a significantly larger "recurrence" energy $E_{\text{rec}}$ (this is connected to quantum recurrence times [82, 83] for an irregular energy spectrum, except in our case time and energy have switched places). In particular, if one has $\tau$ such irregular samples of time and $\widetilde{v}(E_0) = 1$, one expects $\widetilde{v}(E) \ll 1$ between $E_c$ and $E_c \pm E_{\text{rec}}$, where

$$E_{\text{rec}} \sim \frac{2\pi}{\delta t} \exp(\tau). \tag{187}$$

This means that the number of "extraneous" energy bands or shells is reduced *exponentially* with the number of samples $\tau$ if one adopts an irregular sampling of times. Often, the "full width" of the energy spectrum in a many-body system (e.g., the separation between the lowest and highest energy levels) scales [84, 85] as a small power of $N$:

$$E_{\text{max}} - E_{\text{min}} \sim N^{\mu}, \;\; \text{for } \mu > 0. \tag{188}$$

This means that we can completely exclude contributions from other regions of the spectrum than the energy band or energy shell of interest, i.e., ensure that

$$E_{\text{rec}} \gg E_{\text{max}} - E_{\text{min}}, \tag{189}$$

if we take a number of samples at least logarithmic in $N$ (if $\delta t \sim 1$):

$$\tau \gg \mu \log N. \tag{190}$$

We expect this sampling (and therefore the dynamics) to have the largest time complexity in our protocol, as all other steps can be executed over $O(1)$ times.

A remaining question is whether we can still generate *completely positive* functions $v_+(t)$ of this type. This is indeed possible[24], and one example is the spectral form factor $K(t) = |L_{\mathcal{H}}(t)|^2$ of some energy spectrum, which remains non-negative $K(t) > 0$, and whose Fourier transform is the probability distribution (and therefore, non-negative) of two-level spacings in the spectrum. In our case, where the samples are of time rather than energy levels, we should identify an analogue of the spectral form factor with the energy domain and the 2-level distribution with the time domain. This suggests that, given a set of $\eta$ points $\zeta_1, \ldots, \zeta_{\eta}$, we can set $\tau = \eta^2$ and

$$v_+(E - E_c) = \frac{1}{\tau} \sum_{j,k=1}^{\eta} e^{i(\zeta_j - \zeta_k)(E - E_c)} = \left| \frac{1}{\eta} \sum_{k=1}^{\eta} e^{-i\zeta_k(E - E_c)} \right|^2 \geq 0, \tag{191}$$

in which case

$$v_+(t) = \frac{1}{\tau} \sum_{j,k=1}^{\eta} \delta(t - \zeta_j + \zeta_k) \geq 0. \tag{192}$$

---

[24]For a related argument in a separate context that involves convolutions and applies to real-valued functions, due to Laura Shou, see Ref. [28, Appendix D 1]. We use a slightly different argument here to avoid the requirement of real-valuedness, i.e., we do not require $\sum_{k=1}^{\eta} e^{-i\zeta_k(E-E_c)} \in \mathbb{R}$, which allows a more unconstrained choice of the $\zeta_k$.

Here, by sampling the $\zeta_k$ sufficiently irregularly, such as a Poisson sequence (or a uniformly random distribution of points in a fixed interval), we can ensure that $E_{\text{rec}}$ satisfies Eq. (187), and therefore [with a choice of number of points $\tau$ such as Eq. (190) that scales very mildly with $N$] focus on a single energy shell around $E_c$ (or, for $w(t)$, a single energy band around $\delta E = 0$) without contributions from other regions of the spectrum. We also emphasize that the choice of samples $\zeta_j$ and the resulting properties of $\widetilde{v}_+(E - E_c)$ (or $\widetilde{w}_+(\delta E)$) may be verified entirely via classical computation and do not require any specific quantum measurements of the system, as these functions are chosen externally.

## 10   Conclusion

**Summary and outlook**

We have described an approach to quantum statistical mechanics that is based on the accessible dynamical properties of a single initial state corresponding to an observable instead of the detailed properties of the energy levels, mirroring classical approaches [8, 9] that bypass the ergodic hypothesis (see also [86] for a historical survey of related issues). Crucially, as described in Sec. 9, this allows us to make a conclusive experimental determination of thermalization even in large systems of several qubits with tractable resources. While our results center around energy-band thermalization as a finite-resolution variant of eigenstate thermalization, which is aesthetically desirable given the viewpoint that the energy eigenvalues and eigenstates completely determine quantum dynamics, we found that it is possible, and perhaps even more convenient, to directly connect time-domain quantities to thermalization and bypass the energy domain[25].

From an analytical standpoint, our results generalize the observed connection between classical ergodicity and eigenstate thermalization in quantized classical systems [40, 44–50] to observable-dependent quantum thermalization in fully quantum systems. In doing so, we have introduced a framework to account for thermalization over finite energy and time scales, in particular showing that these finite-time features are sufficient to determine thermalization over arbitrarily long timescales due to interference effects (which, to our knowledge, has no obvious classical counterpart). In addition to thermalization, we used this approach to obtain a finite-time Mazur-Suzuki inequality for autocorrelators in terms of approximately conserved charges in App. A, which appears to be the first rigorous result of this nature and may be useful for quantum transport problems in finite but large systems. We have also developed intrinsically quantum methods to access thermalization in energy shells and conserved charges that again rely on interference effects with no classical counterpart.

For the case of global thermalization, where the thermal value is independent of energy or other conserved charges, it follows that an analytical computation of autocorrelators of an observable — for which several methods are known — over some finite timescale, even in a thermodynamic limit, is sufficient to establish its thermalization with a strong notion of "almost all" states. This is illustrated for dual-unitary quantum circuits in App. B. For the more general phenomenon of thermalization in an energy shell or with conserved charges, we believe that it will be valuable to develop a theoretical understanding of the "quantum dynamical echoes" of Sec. 7 in different systems in which mere projection to an energy shell may not be a straightforward operation. We also expect that such echoes may be useful in

---

[25]We view this as being analogous to the "energy-time uncertainty principle" for many-body systems in Refs. [27, 28], where instead of using the spectral form factor to determine the structure of the energy spectrum and then formulate a speed limit in terms of energy parameters, we directly formulate a time-domain speed limit in terms of the time-domain spectral form factor, which implicitly contains information about the energy domain but is best measured in the time domain.

numerical simulations for sufficiently large systems that diagonalizing the Hamiltonian (to obtain the energy levels and project onto an energy shell) becomes computationally expensive. With sufficiently well-developed computational techniques for these echoes, we may hope to analytically or numerically establish the thermalization of suitable observables over (almost) all physical states in specific systems in a fully rigorous manner.

One physical scenario where we expect such fully time-domain results to be particularly relevant is in the case of weak decoherence or dissipation in an otherwise Hamiltonian system. For example, if the system loses coherence after a long time $T_{\rm dec}$ due to weak interactions with the environment, then its energy eigenstates are usually not even definable in a formal sense, but we expect that our energy-independent thermalization results (such as Eq. (26) or Summaries 6.4, 7.3) will generalize immediately to account for thermalization at times $|t| < T_{\rm dec}$. Whether a similar simplification of statistical mechanics is possible once the effects of the environment (if not trivially Markovian) dominate is an interesting open question (for example, we expect the time-translation invariance associated with Hamiltonians to be crucial for finite time measurements to be able to guarantee thermalization over all time scales).

**Thermalization in "almost all" vs. "all" physical states**

Finally, we address an important question that has been of some concern in the literature, which is the problem of the physical relevance of "weak" thermalization. In particular, our results show that thermalization can be established in an accessible manner for "almost all" (rather than precisely all) physical states (i.e., "weak" thermalization), such as computational basis states. While our methods can also show thermalization in every conceivable initial state (i.e., "strong" thermalization), as discussed in Sec. 5, this requires a level of accuracy that we do not expect to be experimentally accessible (nor analytically tractable except perhaps in special cases [59]) in the thermodynamic limit of many particles. Given that our focus has been on "accessible" aspects of quantum statistical mechanics, it is worth highlighting some common objections to weak thermalization and their relevance to our approach.

Let us then consider "weak ETH", which refers to the diagonal part of ETH [Eq. (3)] applying to almost all energy eigenstates rather than all eigenstates. Such a property is trivially true for *local* observables in translation invariant systems in the limit of infinite volume [34, 35, 87], for a translation invariant energy eigenbasis. Barring some trivial cases (such as noninteracting particles) where a highly degenerate spectrum ensures that (weak) eigenstate thermalization is not sufficient for thermalization in (almost) all physical states, this translation invariance property alone is sufficient for time-averaged thermalization in the absence of degeneracies, at least over infinitely long timescales.

This is true even in "integrable" systems with an extensive number of local conserved quantities [34] (again for local observables in the infinite volume limit), and weak ETH has therefore been criticized (e.g., [19]) as being of unclear relevance to thermalization. Here, we point out that this may be an effect of the infinite volume limit: the *non-interacting* ideal gas, despite being a prototypical "trivially" integrable system, is one of the few classical systems that can be rigorously proven to be ergodic and mixing (loosely, thermalizing) in the infinite volume limit with fixed particle density [7]. This can be understood as being due to the "diffusion" of information about the initial state to infinitely far away [88, 89]. In such systems, it may be the case that more nonlocal observables that still remain sensitive to the volume of the system (such as a current density of excitations e.g. average "hopping" velocity of qubit "1" states in a finite fraction of the total volume, coarse grained over a range of values to have only macroscopic accuracy) may be more nontrivial to consider, e.g., if one takes the thermodynamic limit by increasing the density of particles but keeping the volume fixed.

A related objection to concluding thermalization from weak ETH is relevant in "quantum quench" protocols, where the initial state is prepared as a (low energy) stationary state of

2017 one Hamiltonian $\hat{H}_0$, while the dynamics implemented is that of a different Hamiltonian $\hat{H}$.
2018 Here, it is typically the case that the low energy states of $\hat{H}_0$ form (part of) the small fraction of
2019 states in which an observable satisfying weak ETH may fail to thermalize under the dynamics
2020 of $\hat{H}$ if the latter is "integrable" [90]. For these quench protocols, we must develop a way to
2021 restrict the dynamics to an energy shell of $\hat{H}$ which has a large overlap with the low energy
2022 states of $\hat{H}_0$, though it is not presently clear how this can be achieved while maintaining
2023 rigorous accessibility[26]. We also note that there are systems where the quench protocol leads
2024 to thermalization even with just weak ETH at higher energies [91].

2025      Outside the context of quench protocols, however, such as in many-qubit systems where
2026 one expects to have the ability to prepare all computational basis states (for example), we
2027 expect that weak thermalization may be the strongest rigorous and accessible statement one
2028 can make in generic cases (e.g., an analysis in terms of Turing machines suggests that for
2029 translation-invariant Hamiltonians, thermalization in specific initial states is a computationally
2030 undecidable problem [92, 93], translating in our case to $\epsilon$ in Theorem 5.3 being insufficiently
2031 small for thermalization in all states). This is indeed the case in classical statistical mechanics
2032 as well: one can almost never rule out a "measure" zero set of points such as periodic orbits
2033 (which are quite generic in "chaotic" systems [68, 94]) from failing to thermalize [6–8]. Our
2034 view is that in systems where this property trivially follows for a certain class of observables
2035 due to some symmetry but does not describe the behavior of initial states of interest, one
2036 should either focus on a different class of observables, or restrict the Hilbert space to a smaller
2037 subspace with the initial states of interest, corresponding to the two cases described above.

# Acknowledgements

2039 We thank Andrew Lucas for comments on the manuscript, Marcos Rigol for useful discussions
2040 on the physical relevance of "weak" eigenstate thermalization, and R. Shankar for providing a
2041 historical perspective on Ref. [11]. We also thank Peter Reimann for a suggestion to clarify the
2042 term "physical states".

2043 **Funding information**     This work was supported by the Heising-Simons Foundation under
2044 Grant 2024-4848.

# A    Quantum Mazur-Suzuki inequality over finite times

2046 The Mazur-Suzuki inequality [57, 58] states that for an observable $A$ in a classical or quantum
2047 system with exact "orthogonal" conserved quantities $Q_k$ and expectation values $\langle \cdot \rangle$ (so that
2048 $\langle Q_k Q_j \rangle = \langle Q_k^2 \rangle \delta_{kj}$), the infinite time average of the autocorrelator is bounded by:

$$\lim_{T \to \infty} \frac{1}{2T} \int_{-T}^{T} \mathrm{d}t \ \langle A(t)A(0) \rangle \geq \sum_k \frac{\langle AQ_k \rangle^2}{\langle Q_k^2 \rangle}. \tag{A.1}$$

2049 In finite-dimensional quantum systems, this has a significant issue with accessibility [60, 61]: as
2050 with Eq. (21), this requires the infinite time average [58] to be over longer timescales than the
2051 energy level spacings, where each $Q_k = \sum_n q_{kn}|E_n\rangle\langle E_n|$ is a linear combination of a different
2052 subset of energy projectors (not necessarily spanning the full spectrum).

---

[26]As $\hat{H}_0$ does not in general commute with $\hat{H}$ by some significant amount, we do not expect that the echo strategy for conserved charges in Eq. (137) can be applied in a rigorous manner for similar reasons as in Eq. (28).

Here, we will show how completely positive averages of autocorrelators can be used to obtain a general analogue of Eq. (A.1) over finite timescales and even with approximate conserved quantities $\hat{Q}_k$ for any finite-dimensional quantum system. To our knowledge, this type of inequality has not previously been obtained from first principles in this general setting, but only in the classical limit [57] or in a strict thermodynamic limit for local systems, in the (weaker) limit of infinite times [60]. Starting with a time average[27] of the autocorrelator of $\hat{A}$ weighted by $w_+(t)$, and some energy band $\Delta E$ of interest where $\widetilde{w}_+(\delta E < \Delta E) > W$ (with $W < 1$), we have from the derivation of Proposition 6.1 (see App. D.4.1):

$$\int dt \, w_+(t) \operatorname{Tr}[\hat{A}(t)\hat{A}(0)] > W \operatorname{Tr}\left[[\hat{A}]_{\Delta E}^2\right]. \tag{A.2}$$

We recall that $[\hat{A}]_{\Delta E}$ has been defined in Eq. (7); by definition[28], $\hat{A}(t) = e^{i\hat{H}t}\hat{A}(0)e^{-i\hat{H}t}$ and $\hat{A} \equiv \hat{A}(0)$.

Now, consider a set of $M_Q$ (nonzero) orthogonal Hermitian operators $\{\hat{Q}_k\}_{k=1}^{M_Q}$ under the trace inner product, i.e., $\operatorname{Tr}[\hat{Q}_k^\dagger \hat{Q}_j] = \operatorname{Tr}[\hat{Q}_k^2]\delta_{kj}$, $\hat{Q}_k^\dagger = \hat{Q}_k$ and $\operatorname{Tr}[\hat{Q}_k^2] > 0$. Orthogonality restricts $M_Q \le D^2$, where $D^2$ is the dimension of the space of linear operators on $\mathcal{H}$; however, in practice, we expect $M_Q$ to be some much smaller accessible value such as $O(1)$. We can form a complete orthonormal basis for the linear space of operators by introducing $(D^2 - M_Q)$ additional operators $\{\hat{J}_k\}_{k=1}^{D^2-M_Q}$ orthogonal to the $Q_k$ and to each other: $\operatorname{Tr}[\hat{J}_k^\dagger \hat{Q}_k] = 0$, and $\operatorname{Tr}[\hat{J}_k^\dagger \hat{J}_j] = \operatorname{Tr}[\hat{J}_k^\dagger \hat{J}_k]\delta_{kj}$ with $\operatorname{Tr}[\hat{J}_k^\dagger \hat{J}_k] > 0$. Then for any operator $\hat{O}$ acting on $\mathcal{H}$, we have by the completeness relation for this orthonormal basis:

$$\begin{aligned}
\operatorname{Tr}[\hat{O}^\dagger \hat{O}] &= \sum_{k=1}^{M_Q} \frac{\operatorname{Tr}[\hat{O}^\dagger \hat{Q}_k]\operatorname{Tr}[\hat{Q}_k^\dagger \hat{O}]}{\operatorname{Tr}[\hat{Q}_k^\dagger \hat{Q}_k]} + \sum_{k=1}^{D^2-M_Q} \frac{\operatorname{Tr}[\hat{O}^\dagger \hat{J}_k]\operatorname{Tr}[\hat{J}_k^\dagger \hat{O}]}{\operatorname{Tr}[\hat{J}_k^\dagger \hat{J}_k]} \\
&\ge \sum_{k=1}^{M_Q} \frac{\operatorname{Tr}[\hat{O}^\dagger \hat{Q}_k]\operatorname{Tr}[\hat{Q}_k^\dagger \hat{O}]}{\operatorname{Tr}[\hat{Q}_k^\dagger \hat{Q}_k]},
\end{aligned} \tag{A.3}$$

where the second line follows from the fact that each term in the first line is non-negative. Applying this inequality to $\hat{O} = [\hat{A}]_{\Delta E}$ and using the Hermiticity of $\hat{A}$ and the $\hat{Q}_k$ gives:

$$\operatorname{Tr}\left([\hat{A}]_{\Delta E}^2\right) \ge \sum_{k=1}^{M_Q} \frac{\left(\operatorname{Tr}\left[[\hat{A}]_{\Delta E}\hat{Q}_k\right]\right)^2}{\operatorname{Tr}[\hat{Q}_k^2]}. \tag{A.4}$$

Further noting that $\operatorname{Tr}\left([\hat{A}]_{\Delta E}\hat{Q}_k\right) = \operatorname{Tr}\left[\hat{A}[\hat{Q}_k]_{\Delta E}\right]$, this gives for the autocorrelator [with Eq. (A.2)]:

$$\int dt \, w_+(t)\operatorname{Tr}[\hat{A}(t)\hat{A}(0)] > W \sum_{k=1}^{M_Q} \frac{\left(\operatorname{Tr}\left[\hat{A}[\hat{Q}_k]_{\Delta E}\right]\right)^2}{\operatorname{Tr}[\hat{Q}_k^2]}. \tag{A.5}$$

So far, the $\hat{Q}_k$ could have been any set of operators subject to Hermiticity and orthogonality. Now, we will require that each $\hat{Q}_k$ is an approximately conserved quantity. There is no physical

---

[27]For finite temperature averages with some density operator $\hat{\rho}(\hat{H})$, we note that all our considerations generalize if the inner product $\langle \hat{A}, \hat{B} \rangle = \operatorname{Tr}[\hat{A}^\dagger \hat{B}]$ is replaced by $\langle \hat{A}, \hat{B} \rangle_\rho = \operatorname{Tr}[\hat{\rho}(\hat{H})\hat{A}^\dagger \hat{B}]$ throughout, including in Eq. (A.2), provided that $\hat{\rho}(\hat{H})$ is exactly diagonal in the energy eigenbasis as per Eq. (28). In the context of the Mazur-Suzuki inequality, as our interest is not in rigorously determining the dynamics of *other* states in specific energy shells of interest as in Sec. 7, but only in the single state $\hat{\rho}(\hat{H})$ supported on the full system, it may be appropriate to regard the experimental inability to precisely prepare $\hat{\rho}(\hat{H})$ as a mere experimental error that changes the values of the measured correlators in the final inequality by some small $\epsilon$ compared to their ideal values.

[28]Here, we are making a subtle but unimportant switch to Hamiltonian evolution in the Heisenberg picture, in place of the Schrödinger picture in the rest of the text.

2077 loss of generality due to orthogonality: given a set of approximate conserved quantities, we
2078 can always form their linear combinations to generate an orthogonal set, which we identify
2079 with the $\hat{Q}_k$. However, let us pause for a moment to consider a potential issue of accessibility:
2080 ensuring the exact orthogonality of a set of operators is usually inaccessible because of the
2081 large dimension of the Hilbert space; the best one can usually ensure is:

$$\left| \mathrm{Tr}[\hat{Q}_k^\dagger \hat{Q}_j] \right| < \epsilon \sqrt{\mathrm{Tr}[\hat{Q}_k^\dagger \hat{Q}_k] \, \mathrm{Tr}[\hat{Q}_j^\dagger \hat{Q}_j]}, \tag{A.6}$$

2082 for some small $\epsilon > 0$. However, it is easy enough to account for this in practice without any
2083 nontrivial physics: we can derive a result for exactly orthogonal operators, and then consider
2084 the $O(\epsilon)$ errors made if the exactly orthogonal $\hat{Q}_k$ are replaced by experimentally accessible
2085 operators that are sufficiently close to them and satisfy Eq. (A.6) instead; we will therefore
2086 continue to assume exact orthogonality.

2087    Returning to our approximately conserved quantities, we identify such quantities by the
2088 defining requirement that $\hat{Q}_k(t) \approx \hat{Q}_k$ at least over short timescales associated with the energy
2089 band $t \lesssim 2\pi/\Delta E$. More quantitatively, let some completely positive weight function $w_{2+}(t)$
2090 satisfy[29] $\widetilde{w}_{2+}(\delta E > \Delta E) < w_{20}$, where we expect that $0 < w_{20} \ll 1$. We define a measure of
2091 the dynamics of $\hat{Q}_k(t)$ by the difference between its $t = 0$ autocorrelator and a time averaged
2092 one weighted by $w_{2+}(t)$:

$$\delta Q_k^2[w_{2+}] \equiv \left| \mathrm{Tr}[\hat{Q}_k^2] - \int \mathrm{d}t \; w_{2+}(t) \, \mathrm{Tr}[\hat{Q}_k(t)\hat{Q}_k(0)] \right|. \tag{A.7}$$

2093 For approximately conserved quantities, we will require that $\delta Q_k^2[w_{2+}] < \epsilon_k$ for some small $\epsilon_k$
2094 and a suitable choice of $w_{2+}(t)$. However, we do not need to *formally* impose this requirement
2095 for our results, allowing us to analyze what happens even for large $\delta Q_k^2[w_{2+}]$, e.g., when the
2096 conservation laws of a system break down completely, say under some perturbation.

2097    Again, similar to the derivation of Proposition 6.1 (i.e., App. D.4.1), we get the implication:

$$\sum_{n,m} \left\{ 1 - \widetilde{w}_{2+}(E_n - E_m) \right\} \left| \langle E_n | \hat{Q}_k | E_m \rangle \right|^2 = \delta Q_k^2[w_{2+}]$$

$$\implies \mathrm{Tr}[\hat{Q}_k^2] - \mathrm{Tr}\left([\hat{Q}_k]_{\Delta E}^2\right) = \mathrm{Tr}\left[ \left(\hat{Q}_k - [\hat{Q}_k]_{\Delta E}\right)^2 \right] < \frac{\delta Q_k^2[w_{2+}]}{1 - w_{20}}. \tag{A.8}$$

2098 It follows that we can effectively replace $[\hat{Q}]_{\Delta E}$ in Eq. (A.5) with just $\hat{Q}_k$, making some small
2099 error. Specifically, we have by the Cauchy-Schwarz inequality,

$$\left| \mathrm{Tr}\left[\hat{A}\hat{Q}_k\right] - \mathrm{Tr}\left[\hat{A}\,[\hat{Q}_k]_{\Delta E}\right] \right| \le \sqrt{\mathrm{Tr}[\hat{A}^2] \, \mathrm{Tr}\left[\left(\hat{Q}_k - [\hat{Q}_k]_{\Delta E}\right)^2\right]} < \sqrt{\frac{\delta Q_k^2[w_{2+}]}{1 - w_{20}} \, \mathrm{Tr}[\hat{A}^2]}. \tag{A.9}$$

2100 Using this in Eq. (A.5) with the triangle inequality [in the present context, the negative form
2101 $|x - y| \ge |(|x| - |y|)|$], which implies that

$$\left| \mathrm{Tr}\left[\hat{A}\,[\hat{Q}_k]_{\Delta E}\right] \right| \ge \left| \left| \mathrm{Tr}\left[\hat{A}\hat{Q}_k\right] \right| - \sqrt{\frac{\delta Q_k^2[w_{2+}]}{1 - w_{20}} \, \mathrm{Tr}[\hat{A}^2]} \right|, \tag{A.10}$$

---

[29]This assumes continuous time for simplicity. For discrete times, if one wants to verify such a property, one must sample erratically so that the "recurrence" energy lies outside the width of the spectrum, as in Eq. (187), and only require the condition on $\widetilde{w}_{2+}(\delta E)$ for $\delta E$ within the width of the spectrum.

we obtain a rigorous finite-time Mazur-Suzuki inequality that applies to any finite-dimensional quantum system and accounts for approximate conserved quantities $\hat{Q}_k$ via Eq. (A.7):

$$\int dt \, w_+(t) \, \text{Tr}[\hat{A}(t)\hat{A}(0)] > W \sum_k \frac{1}{\text{Tr}[\hat{Q}_k^2]} \left| \left| \text{Tr}[\hat{A}\hat{Q}_k] \right| - \sqrt{\frac{\delta Q_k^2[w_{2+}]}{1 - w_{20}} \, \text{Tr}[\hat{A}^2]} \right|^2 . \quad \text{(A.11)}$$

This recovers Eq. (A.1) for infinite times and exact conserved quantities as $W \to 1$, $\delta Q_k^2[w_{2+}] \to 0$ and $w_{20} \to 0$. But we emphasize that in this formulation, despite being for a general finite-dimensional quantum system, it is entirely possible to take the thermodynamic limit $N \to \infty$ first before the infinite time limit [60].

Eq. (A.11) is nontrivial when the right hand side is larger than $A_{\text{th}}^2$, which is the minimum value of the autocorrelator and corresponds to (global) thermalization. However, it only constrains thermalization over the same timescale (determined by $\Delta E$) in which the charge remains approximately conserved. For a noticeably broken conservation law (say, after a long timescale), $\delta Q_k^2[w_{2+}]$ is large, and consequently the second term within the absolute value that depends on $\delta Q_k^2[w_{2+}]$ weakens the bound, allowing the autocorrelator to decay further than with a perfect conservation law and potentially thermalize (as expected for systems with fewer conservation laws). For the question [61] of when this inequality can be saturated over finite timescales with a finite number of accessible charges $Q_k$, which are not necessarily energy projectors as in eigenstate thermalization [Eq. (21)], we note that this is possible when $[\hat{A}]_{\Delta E}$ has most of its overlap with the $\hat{Q}_k$ rather than the $\hat{J}_k$ by Eq. (A.4) and the completeness relation Eq. (A.3) with $\hat{O} = [\hat{A}]_{\Delta E}$. It remains to be seen if this criterion can be used more systematically to determine when the inequality may be saturated in different classes of systems. It would also be interesting to consider the implications of the exact inequality in Eq. (A.11) for quantum transport problems, e.g., in a finite but large number of qubits.

## B  Thermalization in dual-unitary quantum circuits

As a case study, mainly for illustrative purposes, let us consider the example of dual-unitary quantum circuits [62–64]. These are brickwork quantum circuits of, say, $N$ qubits (with Hilbert space dimension $D = 2^N$), where the qubits are arranged in one dimension in a periodic chain and 2-qubit unitary gates, each with the special property of "dual-unitarity", simultaneously act on alternate pairs of qubits[30]. We will impose time translation invariance so that the resulting (Floquet) discrete-time system, with each step consisting of two alternating parallel applications of local gates, has well-defined (quasi-)energy levels $E_n$. But we do not require spatial translation invariance, i.e., each 2-qubit dual-unitary gate at a given time slice may be entirely different. The latter excludes spatial translation invariance as one of the special symmetries under which weak eigenstate thermalization may be directly shown for local observables in the infinite volume limit (assuming nondegenerate levels) [34, 35]; therefore, it is not yet clear if weak eigenstate thermalization is satisfied in general in our case. Even for translation-invariant circuits, it is not clear if the spectrum is necessarily nondegenerate for every single circuit of interest, which is required for thermalization to follow from eigenstate thermalization.

Under these circumstances, it can be shown that all autocorrelators of traceless single-site observables vanish for all time steps $|t| < N$, provided each unitary 2-qubit gate satisfies the property of "dual unitarity", for the details of which we refer to Ref. [62]. While the dynamics

---

[30]We consider qubits here for definiteness, but our conclusions should generalize to dual-unitary circuits for qu*d*its [63, 64].

of specific (matrix product) initial states have been computed exactly for such circuits [95], and $f(E_1, E_2)$ in the ETH ansatz for certain observables has been estimated from autocorrelators [96] assuming the form in Eq. (3), our intention here is to show how more general conclusions on thermalization may be drawn in a much simpler way. Quantitatively, all traceless single-qubit observables $\widehat{\delta a}_i$ (with $i = 1$ to $N$ indexing the qubit) satisfy:

$$\text{Tr}[\widehat{\delta a}_i(t)\widehat{\delta a}_i(0)] = 0, \text{ for all } t : 0 < |t| < N. \tag{B.1}$$

For intuition, we note that in the $N \to \infty$ thermodynamic limit, all autocorrelators thermalize at all (nonzero) finite times. Then, our results [we will directly base our statements on Sec. 2.1 and 2.2, which are informal versions of Summary 6.4 and the Theorems therein combined with Sec. 8] imply the following: Given any basis of initial states $|\psi_k\rangle$ (which may be the computational basis states, or more nontrivially any family of states obtained by acting with an arbitrary unitary circuit on the computational basis states), and every single-qubit observable $\hat{a}_i$, for every $\epsilon > 0$, there exists some $N_0 \in \mathbb{N}$ and $T_0 \in \mathbb{N}$ such that the following hold for any choice of $t_0 \in \mathbb{Z}$ for any $N$-qubit dual unitary circuit if $N > N_0$:

1. Time-averaged thermalization in almost all basis states [as in Eq. (9)]:

$$\left| \frac{1}{2T} \int_{t_0-T}^{t_0+T} dt \, \langle \psi_k(t) | \, \hat{a}_i \, | \psi_k(t) \rangle - \frac{1}{2} \text{Tr}_i[\hat{a}_i] \right| < \epsilon, \text{ for almost all } k \text{ for any } T > T_0,$$

$$\tag{B.2}$$

2. Eigenstate thermalization in almost all eigenstates [by Eq. (10)]:

$$\left| \langle E_n | \hat{a}_i | E_n \rangle - \frac{1}{2} \text{Tr}_i[\hat{a}_i] \right| < \epsilon, \text{ for almost all } n. \tag{B.3}$$

   We also have, as in Eq. (12), that $\hat{a}_i$ is trivially thermal in almost all states in any basis of initial states with support on a narrow band of quasi-energies, and that off-diagonal fluctuations have low variance as in Eq. (15).

3. Basis-state-averaged thermalization at almost all times [as in Eq. (18)]:

$$\left| \sum_k p_k \langle \psi_k(t) | \, \hat{a}_i \, | \psi_k(t) \rangle - \frac{1}{2} \text{Tr}_i[\hat{a}_i] \right| < \epsilon, \text{ for almost all } t \in [t_0-T, t_0+T], \text{ for any } T > T_0,$$

$$\tag{B.4}$$

   for any set of weights $p_k$ provided that $2^N \sum_k p_k^2 \leq 1/\mu$ for some constant $0 < \mu \leq 1$ (see Eq. (18) for the interpretation of "almost all t").

4. Thermalization of subsystems with a statistical bath [as in Eq. (20)]: if one partitions the $N$ qubits into $N_C$ non-thermal qubits in *any* pure state $|\psi\rangle_C$ and $N_B = N - N_C$ bath qubits in a mixed (e.g. high-temperature) state $\hat{\rho}_B$, the observable $a_i$ (which may e.g. be chosen to be within the $N_C$ non-thermal qubits) is guaranteed to attain thermal equilibrium at almost all times:

$$\left| \text{Tr}\left[ \left( |\psi\rangle_C \langle\psi| \otimes \hat{\rho}_B \right)(t) \, \hat{a}_i \right] - \frac{1}{2} \text{Tr}_i[\hat{a}_i] \right| < \epsilon, \text{ for almost all } t \in [t_0-T, t_0+T], \text{ for any } T > T_0,$$

$$\tag{B.5}$$

   as long as $N_C \lesssim (\log N)^\alpha$ for a suitable constant $\alpha \geq 0$ (so that $2^{N_C}$ is at most accessibly large as a function of $N$), and $2^{N_B} \text{Tr}_B[\hat{\rho}_B^2] \leq c$ for any accessibly large constant $c$.

Therefore, given a method to compute autocorrelators, our results enable a rigorous proof of different kinds of accessible thermalization processes in individual many-body systems in the thermodynamic limit, without any direct input required from the energy levels, which become inaccessible in this limit. At the same time, we can also make rigorous statements about eigenstate thermalization from these autocorrelators.

While we have expressed Eq. (B.4) for single-qubit observables, larger local observables on, say, $N_S$ consecutive qubits must also have vanishing autocorrelators except at extremely short $t = O(N_S)$ and extremely long $t = \Omega(N)$ times, by our understanding of Ref. [62]. We therefore expect an analogue of Eq. (B.4) to hold for any traceless local observable spanning $N_S = o(N)$ consecutive sites. Despite the strong thermalization behavior (at a level comparable to classical weak-mixing [5–7]) indicated by the above statements, it is known that some of these circuits show spectral correlations corresponding to dynamical non-ergodicity, e.g., Poisson spectral statistics [63]. These examples therefore also directly illustrate the logical separation between ergodic dynamics and the statistical mechanics of observables.

Another interesting class of observables are "comoving" single-qubit observables $\hat{a}_i^{\text{co}}$, which are chosen to implicitly shift by two qubits after every Floquet time step (which corresponds to a "wavefront" of information propagation in any local brickwork circuit — information cannot advance by more than 2 qubits in a Floquet time step, which consists of two successive applications of alternating 2-qubit gates). To every single-particle observable $\hat{a}_i$, we can associate a comoving analogue via (still keeping to the Schrödinger picture where states have dynamics while operators do not; the $t$ below should be interpreted as a property of the relation between the observables at different times rather than explicit dynamics):

$$\hat{a}_i^{\text{co}}[t] = \hat{a}_{i+2t}. \tag{B.6}$$

The dynamics of $\hat{a}_i^{\text{co}}$ in a dual unitary circuit is equivalent to the dynamics of the original $\hat{a}_i$ in a circuit formed by staggering each Floquet step of the dual unitary circuit with a (periodic) 2-qubit shift/translation of the qubits in one dimension, in the opposite direction [97, 98]. In this case, however, the analogue of Eq. (B.1) is not satisfied for all dual unitary circuits, but only as an asymptotic equality (i.e. the autocorrelator is less than any given $\epsilon$) for special classes called mixing circuits at long but finite times (and for higher qudits, Bernoulli circuits at small times as well) [62–64]. Comoving single-particle observables can therefore thermalize in the sense of Eq. (B.4) or (B.5) for almost all initial states if they are mixing or Bernoulli.

As an aside, to emphasize the nontrivial role of comoving observables, it is worth considering the dynamics of a shift unitary itself, i.e. merely translating the qubits by (say) one unit in one direction in each time step e.g. $\hat{a}_i(t) = \hat{a}_{i-t}(0)$. While this is an entirely classical system, local observables also satisfy Eq. (B.1) (as it takes a full period $t = N$ for the shift to return the original qubit to itself to have any autocorrelation), and the corresponding implications for thermalization (comparable to weak-mixing) follow directly in the $N \to \infty$ limit. From a classical standpoint, this is not surprising: when acting on classical bitstrings (the computational basis states), the shift unitary approaches a maximally chaotic Bernoulli shift with a binary alphabet $\{0, 1\}$ in the $N \to \infty$ limit, which has very strong ergodic and mixing properties (including K-mixing and Bernoulli) [6], and therefore must thermalize almost all ensembles of states (as well as higher order correlators, though never thermalizing in any individual bitstring). Nevertheless, a comoving observable $\hat{a}_i^{\text{co}}[t] = \hat{a}_{i+t}$ strongly violates Eq. (B.1) and trivially does not thermalize for shift unitaries. This shows that mixing and Bernoulli dual-unitary circuits (in which comoving observables also thermalize) even have stronger accessible thermalization properties than classical Bernoulli shifts implemented as quantum circuits.

To summarize our case study, our results allow the rigorous characterization of how local observables may thermalize or fail to thermalize (see Summary 6.4) over accessible timescales entirely based on the behavior of autocorrelators for different classes of dual-unitary gates. This

illustrates how quantum thermalization in many-body systems, at least in cases corresponding to "global thermalization" without explicit energy dependence, may be rigorously treated in a simple manner similar to the classical statistical mechanics of observables, as in Eq. (1).

## C  Energy-band thermalization $\implies$ eigenspaces *usually* thermalize

Here, we will constrain eigenspace thermalization given energy-band thermalization, deriving and expanding on Eq. (74). In particular, given energy-band thermalization, we constrain the dimension of the subspace of the energy shell $\Sigma_d$ in which eigenspace thermalization may be violated. To formalize this, let us write

$$\Sigma_d = \Sigma_{d_s} \oplus \Sigma_{\delta d} \tag{C.1}$$

such that $\hat{\Pi}_A$ satisfies eigenspace thermalization to some accuracy $\lambda > 0$ for all eigenspaces $\mathcal{H}_s(E) \subseteq \Sigma_{d_s}$, and does not satisfy eigenspace thermalization to this accuracy for all eigenspaces $\mathcal{H}_\delta(E) \subseteq \Sigma_{\delta d}$. It is also convenient to use $n(\Sigma_d)$ to denote the number of eigenspaces contained in the energy shell $\Sigma_d$ (and likewise for $\Sigma_{d_s}$, $\Sigma_{\delta d}$).

Then, Eq. (73) implies that:

**Proposition C.1** (Energy-band thermalization constrains the number of non-thermal eigenspaces)**.** *Let $\hat{\Pi}_A$ satisfy energy-band thermalization to $\langle\hat{\Pi}_A\rangle_{\Sigma_d}$ with some bandwidth $\Delta E > 0$ and accuracy $\epsilon > 0$ in $\Sigma_d$. Then, any smaller energy shell $\Sigma_{\delta d} \subseteq \Sigma_d$ in which $\hat{\Pi}_A$ does not satisfy eigenspace thermalization to $\langle\hat{\Pi}_A\rangle_{\Sigma_d}$ with a given accuracy $\lambda > 0$, can contain only $n(\Sigma_{\delta d})$ eigenspaces, constrained by*

$$n(\Sigma_{\delta d}) \equiv \left( \sum_{\mathcal{H}_\delta(E) \in \Sigma_{\delta d}} 1 \right) < \frac{\epsilon^2}{\lambda^2} d. \tag{C.2}$$

*Proof.* This follows from interpreting Eq. (73) as an equal-weight average over different eigenspaces $\mathcal{H}(E)$; see App. D.3.1. $\qquad\square$

Let us consider the implications of Eq. (C.2). First, we note that for almost all eigenspaces to thermalize, the ratio $n(\Sigma_{\delta d})/n(\Sigma_d)$ should become negligible:

$$\frac{n(\Sigma_{\delta d})}{n(\Sigma_d)} \ll 1. \tag{C.3}$$

In analogy with "weak eigenstate thermalization" [18,34,35], we will diagnose "weak eigenspace thermalization" if Eq. (C.3) is satisfied. By Proposition 4.3, this implies that thermalization o.a. occurs in arbitrary initial states supported on an energy shell $\Sigma_{d_s}$ that contains most of the eigenspaces in $\Sigma_d$. Determining weak eigenspace thermalization requires our accuracy for energy-band thermalization to be:

$$\epsilon_{\text{weak}} \ll \lambda \sqrt{\frac{n(\Sigma_d)}{d}}. \tag{C.4}$$

This depends on the number of eigenspaces $n(\Sigma_d)$ in the energy shell, and is therefore no longer independent of the energy spectrum. In particular, for $\epsilon_{\text{weak}}$ to be an "accessible" quantity (i.e. is significantly larger than $1/d$), $n(\Sigma_d)$ must be a correspondingly large fraction of $d$ (which can still be $\ll 1$). For example, in a system of $N$ particles, if we want (for some $\alpha > 0$)

$$\epsilon_{\text{weak}} > \frac{1}{N^\alpha}, \tag{C.5}$$

then we must have the following constraint on the total number of eigenspaces in the energy spectrum:

$$n(\Sigma_d) > \frac{d}{N^{2\alpha}}. \tag{C.6}$$

That we cannot show weak eigenspace thermalization with an accessible accuracy of energy-band thermalization without some constraint on the spectrum is why the title of this Appendix emphasizes the qualifier "usually". Concretely, "usually" refers to systems satisfying Eq. (C.6); it implies that the spectrum must generally have a low degree of degeneracy. Nevertheless, we expect this condition to apply to a large class of systems, especially seeing that it allows $n(\Sigma_d) \ll d$ (such as systems with a finite or at most weakly growing degree of degeneracy in each level, in the thermodynamic limit $d \to \infty$).

In contrast, to ensure that eigenspace thermalization is satisfied in $\Sigma_d$ rather than just the smaller $\Sigma_{d_s}$, i.e. every eigenspace in $\Sigma_d$ satisfies eigenspace thermalization, we require

$$n(\Sigma_{\delta d}) < 1, \tag{C.7}$$

to diagnose which we need an accuracy of

$$\epsilon < \frac{\lambda}{\sqrt{d}} \tag{C.8}$$

in energy-band thermalization. This is consistent with Eq. (72). This constraint is independent of the energy spectrum, but as $\lambda < 1$ for a nontrivial expression of eigenspace thermalization, Eq. (C.8) is not expected to be an accessible degree of accuracy (scaling as a power of $1/d$).

# D  Proofs

## D.1  Classical "eigenstate" thermalization without ergodicity

### D.1.1  Proposition 3.1: Classical eigenstate thermalization implies thermalization o.a.

We have

$$\int \overline{dt}\; \pi_A[\rho(x,t)] = \sum_k \int \overline{dt}\; \pi_{A \cap \mathcal{P}_k}[\rho(x,t)]$$
$$= \sum_k \frac{\mu(A \cap \mathcal{P}_k)}{\mu(\mathcal{P}_k)} \pi_{\mathcal{P}_k}[\rho]. \tag{D.1}$$

In the second line, we have used the ergodicity of each subset $\mathcal{P}_k$ by applying Eq. (42). It now follows that if Eq. (43) holds, then we get the appearance of ergodicity on the full phase space as per Eq. (38), as

$$\sum_k \pi_{\mathcal{P}_k}[\rho] = \pi_{\mathcal{P}}[\rho]. \tag{D.2}$$

### D.1.2  Proposition 3.2: A single initial distribution determines classical eigenstate thermalization

From Eq. (44), rewriting the left hand side in terms of the ergodic subsets $\mathcal{P}_k$ as in Eq. (D.1), we get

$$\sum_k \frac{\mu(A \cap \mathcal{P}_k)}{\mu(\mathcal{P}_k)} \pi_{\mathcal{P}_k}[\rho_A] = \frac{\mu(A)}{\mu(\mathcal{P})}. \tag{D.3}$$

For the specific initial state in Eq. (45), we have

$$\pi_{\mathcal{P}_k}[\rho_A] = \frac{\mu(A \cap \mathcal{P}_k)}{\mu(A)}, \tag{D.4}$$

using which Eq. (D.3) becomes (together with some manipulations on the right hand side using $\sum_k \mu(\mathcal{P}_k) = \mu(\mathcal{P})$):

$$\frac{1}{\mu(A)} \sum_k \mu(\mathcal{P}_k) \frac{\mu(A \cap \mathcal{P}_k)^2}{\mu(\mathcal{P}_k)^2} = \frac{1}{\mu(A)} \left[ 2\mu(A) \frac{\mu(A)}{\mu(\mathcal{P})} - \sum_k \mu(\mathcal{P}_k) \frac{\mu(A)^2}{\mu(\mathcal{P})^2} \right]$$

$$\implies \frac{1}{\mu(A)} \sum_k \mu(\mathcal{P}_k) \frac{\mu(A \cap \mathcal{P}_k)^2}{\mu(\mathcal{P}_k)^2} = \frac{1}{\mu(A)} \left[ 2 \sum_k \mu(\mathcal{P}_k) \frac{\mu(A \cap \mathcal{P}_k)}{\mu(\mathcal{P}_k)} \frac{\mu(A)}{\mu(\mathcal{P})} - \sum_k \mu(\mathcal{P}_k) \frac{\mu(A)^2}{\mu(\mathcal{P})^2} \right]. \tag{D.5}$$

Collecting all the terms in Eq. (D.5), we get a weighted variance:

$$\frac{1}{\mu(A)} \sum_k \mu(\mathcal{P}_k) \left[ \frac{\mu(A \cap \mathcal{P}_k)}{\mu(\mathcal{P}_k)} - \frac{\mu(A)}{\mu(\mathcal{P})} \right]^2 = 0. \tag{D.6}$$

Each term is non-negative and $\mu(\mathcal{P}_k) > 0$ by assumption, implying eigenstate thermalization as in Eq. (43):

$$\frac{\mu(A \cap \mathcal{P}_k)}{\mu(\mathcal{P}_k)} - \frac{\mu(A)}{\mu(\mathcal{P})} = 0. \tag{D.7}$$

## D.2    Quantum eigenspace thermalization with degeneracies

### D.2.1    Proposition 4.1: Quantum eigenstate thermalization implies thermalization o.a. given a nondegenerate spectrum [10, 15]

From Eq. (51), we have by the triangle inequality:

$$\left| \int \overline{dt} \ \mathrm{Tr}[\hat{\rho}(t) \hat{\Pi}_A] - \langle \hat{\Pi}_A \rangle_{\Sigma_d} \right| \leq \left( \sum_{n: |E_n\rangle \in \Sigma_d} \langle E_n | \hat{\rho} | E_n \rangle \right) \max_{n: |E_n\rangle \in \Sigma_d} \left| \langle E_n | \hat{\Pi}_A | E_n \rangle - \langle \hat{\Pi}_A \rangle_{\Sigma_d} \right| \tag{D.8}$$

As $\sum_n \langle E_n | \hat{\rho} | E_n \rangle = 1$, Eq. (50) follows from here, given Eq. (49).

### D.2.2    Proposition 4.3: Quantum eigenspace thermalization implies thermalization o.a.

Rewriting Eq. (59) to include the $-\langle \hat{\Pi}_A \rangle_{\Sigma_d}$ term, we get:

$$\int \overline{dt} \ \mathrm{Tr}[\hat{\rho}(t) \hat{\Pi}_A] - \langle \hat{\Pi}_A \rangle_{\Sigma_d} = \sum_{E \in \mathcal{E}} \left[ \sum_{|E_n\rangle, |E_m\rangle \in \mathcal{H}(E)} \langle E_n | \hat{\rho} | E_m \rangle \langle E_m | \left( \hat{\Pi}_A - \langle \hat{\Pi}_A \rangle_{\Sigma_d} \hat{\mathbb{1}} \right) | E_n \rangle \right]$$

$$= \sum_{E \in \mathcal{E}} \mathrm{Tr} \left[ \hat{\Pi}(E) \hat{\rho} \hat{\Pi}(E) \hat{\Pi}(E) \left( \hat{\Pi}_A - \langle \hat{\Pi}_A \rangle_{\Sigma_d} \hat{\mathbb{1}} \right) \hat{\Pi}(E) \right], \tag{D.9}$$

where the second line follows from

$$\hat{\Pi}(E) = \hat{\Pi}(E)^2 = \sum_{n: |E_n\rangle \in \mathcal{H}_E} |E_n\rangle \langle E_n|. \tag{D.10}$$

Now applying the Cauchy-Schwarz inequality to each term in Eq. (D.9) with a given $E$, and the triangle inequality to consider each such term separately, we get

$$\left| \int \overline{dt} \ \text{Tr}[\hat{\rho}(t)\hat{\Pi}_A] - \langle \hat{\Pi}_A \rangle_{\Sigma_d} \right| \leq \sum_{E \in \mathcal{E}} \sqrt{\text{Tr}\left[ \{ \hat{\rho}\hat{\Pi}(E) \}^2 \right] \text{Tr}\left[ \left\{ \left( \hat{\Pi}_A - \langle \hat{\Pi}_A \rangle_{\Sigma_d} \hat{\mathbb{1}} \right) \hat{\Pi}(E) \right\}^2 \right]} \quad \text{(D.11)}$$

$$\leq \epsilon \sum_{E \in \mathcal{E}} \sqrt{\text{Tr}\left[ \{ \hat{\rho}\hat{\Pi}(E) \}^2 \right]}. \quad \text{(D.12)}$$

For any density operator $\hat{\rho}$, we have the inequality

$$|\langle E_n | \hat{\rho} | E_m \rangle|^2 \leq \langle E_n | \hat{\rho} | E_n \rangle \langle E_m | \hat{\rho} | E_m \rangle, \quad \text{(D.13)}$$

which is the Cauchy-Schwarz inequality for the inner product $\langle \phi, \psi \rangle_\rho \equiv \langle \phi | \hat{\rho} | \psi \rangle$, noting that this is an admissible inner product because $\hat{\rho}$ is a positive linear operator ($\langle \psi | \hat{\rho} | \psi \rangle \geq 0$, with $\hat{\rho} \neq 0$ as $\text{Tr}[\hat{\rho}] = 1$). Consequently,

$$\text{Tr}\left[ \{ \hat{\rho}\hat{\Pi}(E) \}^2 \right] = \sum_{|E_n\rangle, |E_m\rangle \in \mathcal{H}(E)} |\langle E_n | \hat{\rho} | E_m \rangle|^2$$

$$\leq \sum_{|E_n\rangle, |E_m\rangle \in \mathcal{H}(E)} \langle E_n | \hat{\rho} | E_n \rangle \langle E_m | \hat{\rho} | E_m \rangle$$

$$= \text{Tr}\left[ \hat{\rho}\hat{\Pi}(E) \right]^2. \quad \text{(D.14)}$$

Inserting this inequality into Eq. (D.12), we get

$$\left| \int \overline{dt} \ \text{Tr}[\hat{\rho}(t)\hat{\Pi}_A] - \langle \hat{\Pi}_A \rangle_{\Sigma_d} \right| \leq \epsilon \sum_{E \in \mathcal{E}} \text{Tr}\left[ \hat{\rho}\hat{\Pi}(E) \right] = \epsilon, \quad \text{(D.15)}$$

which proves Eq. (58), where we have used $\text{Tr}\left[ \hat{\rho}\hat{\Pi}(E) \right] \geq 0$ and

$$\sum_{E \in \mathcal{E}} \text{Tr}\left[ \hat{\rho}\hat{\Pi}(E) \right] = \text{Tr}[\hat{\rho}] = 1. \quad \text{(D.16)}$$

## D.3 Quantum energy-band thermalization for accessible time scales

### D.3.1 Proposition C.1: Energy-band thermalization constrains the number of non-thermal eigenspaces

From Eq. (73), using the decomposition of eigenspaces into the two energy shells $\Sigma_{d_s}$ and $\Sigma_{\delta d}$ as in Eq. (C.1), with respective projectors $\hat{\Pi}_s(E)$ and $\hat{\Pi}_\delta(E)$, we get

$$\frac{1}{d} \sum_{\mathcal{H}_s(E) \subseteq \Sigma_{d_s}} \text{Tr}\left[ \left\{ \left( \hat{\Pi}_A - \langle \hat{\Pi}_A \rangle_{\Sigma_d} \hat{\mathbb{1}} \right) \hat{\Pi}_s(E) \right\}^2 \right] + \frac{1}{d} \sum_{\mathcal{H}_\delta(E) \subseteq \Sigma_{\delta d}} \text{Tr}\left[ \left\{ \left( \hat{\Pi}_A - \langle \hat{\Pi}_A \rangle_{\Sigma_d} \hat{\mathbb{1}} \right) \hat{\Pi}_\delta(E) \right\}^2 \right] < \epsilon^2. \quad \text{(D.17)}$$

Noting that the first term is non-negative, we get an inequality only for $\Sigma_{\delta d}$:

$$\frac{1}{d} \sum_{\mathcal{H}_\delta(E) \subseteq \Sigma_{\delta d}} \text{Tr}\left[ \left\{ \left( \hat{\Pi}_A - \langle \hat{\Pi}_A \rangle_{\Sigma_d} \hat{\mathbb{1}} \right) \hat{\Pi}_\delta(E) \right\}^2 \right] < \epsilon^2. \quad \text{(D.18)}$$

By assumption, $\hat{\Pi}_A$ satisfies eigenspace thermalization to accuracy $\lambda$ in $\Sigma_{d_s}$:

$$\text{Tr}\left[ \left\{ \left( \hat{\Pi}_A - \langle \hat{\Pi}_A \rangle_{\Sigma_d} \hat{\mathbb{1}} \right) \hat{\Pi}_s(E) \right\}^2 \right] < \lambda^2, \quad \text{(D.19)}$$

2304 and violates it in $\Sigma_{\delta d}$:

$$\text{Tr}\left[\left\{\left(\hat{\Pi}_A - \langle\hat{\Pi}_A\rangle_{\Sigma_d}\hat{\mathbb{1}}\right)\hat{\Pi}_\delta(E)\right\}^2\right] \geq \lambda^2. \tag{D.20}$$

2305 Eq. (D.20) further implies

$$\frac{1}{d}\sum_{\mathcal{H}_\delta(E)\subseteq\Sigma_d}\text{Tr}\left[\left\{\left(\hat{\Pi}_A - \langle\hat{\Pi}_A\rangle_{\Sigma_d}\hat{\mathbb{1}}\right)\hat{\Pi}_\delta(E)\right\}^2\right] \geq \frac{\lambda^2}{d}\left(\sum_{\mathcal{H}_\delta(E)\subseteq\Sigma_{\delta d}}1\right). \tag{D.21}$$

2306 Combining this with Eq. (D.18), we get

$$\left(\sum_{\mathcal{H}_\delta(E)\subseteq\Sigma_{\delta d}}1\right) \leq \frac{\epsilon^2}{\lambda^2}d, \tag{D.22}$$

2307 which is Eq. (C.2).

### D.3.2 Theorem 5.2: Energy-band thermalization implies thermalization o.a. over accessible timescales

2310 From Eq. (63), we have

$$\left|\int dt\, w(t)\,\text{Tr}[\hat{\rho}(t)\hat{\Pi}_A] - \langle\hat{\Pi}_A\rangle_{\Sigma_d}\right| = \left|\sum_{\substack{n,m:\\|E_n\rangle,|E_m\rangle\in\Sigma_d}}\widetilde{w}(E_n - E_m)\langle E_n|\hat{\rho}|E_m\rangle\langle E_m|\left(\hat{\Pi}_A - \langle\hat{\Pi}_A\rangle_{\Sigma_d}\hat{\mathbb{1}}\right)|E_n\rangle\right|. \tag{D.23}$$

2311 To simplify our notation, we will take $|E_n\rangle, |E_m\rangle \in \Sigma_d$ as given in what follows, without explicitly
2312 writing out this condition in the sums. Separating Eq. (D.23) into contributions from within
2313 and outside the bandwidth $\Delta E$ (noting that $\hat{\mathbb{1}}$ has no off-diagonal matrix elements) and using
2314 the triangle inequality, we get

$$\left|\int dt\, w(t)\,\text{Tr}[\hat{\rho}(t)\hat{\Pi}_A] - \langle\hat{\Pi}_A\rangle_{\Sigma_d}\right| \leq \left|\sum_{\substack{n,m:\\|E_n-E_m|<\Delta E}}\widetilde{w}(E_n - E_m)\langle E_n|\hat{\rho}|E_m\rangle\langle E_m|\left(\hat{\Pi}_A - \langle\hat{\Pi}_A\rangle_{\Sigma_d}\hat{\mathbb{1}}\right)|E_n\rangle\right|$$

$$+ \left|\sum_{\substack{n,m:\\|E_n-E_m|\geq\Delta E}}\widetilde{w}(E_n - E_m)\langle E_n|\hat{\rho}|E_m\rangle\langle E_m|\hat{\Pi}_A|E_n\rangle\right|. \tag{D.24}$$

2315 It is convenient to consider each term separately.

2316 For the first term, we can use the Cauchy-Schwarz inequality between (schematically) $\widetilde{w}\hat{\rho}$
2317 and $\hat{\Pi}_A$, and use $|\widetilde{w}(\delta E)| \leq 1$ due to $w(t) \geq 0$ [Eq. (66)] to get:

$$\left|\sum_{\substack{n,m:\\|E_n-E_m|<\Delta E}}\widetilde{w}(E_n - E_m)\langle E_n|\hat{\rho}|E_m\rangle\langle E_m|\left(\hat{\Pi}_A - \langle\hat{\Pi}_A\rangle_{\Sigma_d}\hat{\mathbb{1}}\right)|E_n\rangle\right|$$

$$\leq \sqrt{\left\{\sum_{\substack{n,m:\\|E_n-E_m|<\Delta E}}\left|\widetilde{w}(E_n - E_m)\langle E_n|\hat{\rho}|E_m\rangle\right|^2\right\}\left\{\sum_{\substack{n,m:\\|E_n-E_m|<\Delta E}}\left|\langle E_m|\left(\hat{\Pi}_A - \langle\hat{\Pi}_A\rangle_{\Sigma_d}\hat{\mathbb{1}}\right)|E_n\rangle\right|^2\right\}}$$

$$\leq \sqrt{\left(\sum_{\substack{n,m:\\|E_n-E_m|<\Delta E}}\left|\langle E_n|\hat{\rho}|E_m\rangle\right|^2\right)\text{Tr}\left\{\left[\hat{\Pi}_A - \langle\hat{\Pi}_A\rangle_{\Sigma_d}\hat{\mathbb{1}}\right]^2_{(\Sigma_d,\Delta E)}\right\}} \tag{D.25}$$

Here, it is convenient to use $\sum_{n,m \in S} |\langle n|\hat{A}|m\rangle|^2 \le \mathrm{Tr}[\hat{A}^2]$ for any Hermitian $\hat{A}$ and some set of values $S$ indexing an orthonormal basis $|n\rangle$ for $\hat{\rho}$, which implies (as $\hat{\rho}$ acts only on $\Sigma_d$):

$$\sum_{\substack{n,m: \\ |E_n - E_m| < \Delta E}} \left| \langle E_n|\hat{\rho}|E_m\rangle \right|^2 \le \mathrm{Tr}[\hat{\rho}^2]. \tag{D.26}$$

If $\hat{\Pi}_A$ satisfies energy-band thermalization with bandwidth $\Delta E$ and accuracy $\epsilon$, then by Eq. (71), we get:

$$\left| \sum_{\substack{n,m: \\ |E_n - E_m| < \Delta E}} \widetilde{w}(E_n - E_m)\langle E_n|\hat{\rho}|E_m\rangle\langle E_m| \left( \hat{\Pi}_A - \langle \hat{\Pi}_A\rangle_{\Sigma_d} \hat{\mathbb{1}} \right) |E_n\rangle \right| < \epsilon \sqrt{d\,\mathrm{Tr}[\hat{\rho}^2]}. \tag{D.27}$$

For the second term, we again initially use the Cauchy-Schwarz inequality between $\widetilde{w}\hat{\rho}$ and $\hat{\Pi}_A$, and then $|\widetilde{w}(\delta E)| \le w_0$ for $\delta E \ge \Delta E$ [Eq. (67)]:

$$\left| \sum_{\substack{n,m: \\ |E_n - E_m| \ge \Delta E}} \widetilde{w}(E_n - E_m)\langle E_n|\hat{\rho}|E_m\rangle\langle E_m|\hat{\Pi}_A|E_n\rangle \right|$$

$$\le \sqrt{\left\{ \sum_{\substack{n,m: \\ |E_n - E_m| \ge \Delta E}} \left| \widetilde{w}(E_n - E_m)\langle E_n|\hat{\rho}|E_m\rangle \right|^2 \right\} \left\{ \sum_{\substack{n,m: \\ |E_n - E_m| \ge \Delta E}} \left| \langle E_m|\hat{\Pi}_A|E_n\rangle \right|^2 \right\}}$$

$$\le w_0 \sqrt{\left\{ \sum_{\substack{n,m: \\ |E_n - E_m| \ge \Delta E}} \left| \langle E_n|\hat{\rho}|E_m\rangle \right|^2 \right\} \left\{ \sum_{\substack{n,m: \\ |E_n - E_m| \ge \Delta E}} \left| \langle E_m|\hat{\Pi}_A|E_n\rangle \right|^2 \right\}} \tag{D.28}$$

Here, it is again convenient to use $\sum_{n,m \in S} |\langle n|\hat{A}|m\rangle|^2 \le \mathrm{Tr}[\hat{A}^2]$ (for Hermitian $\hat{A}$ and some set of values $S$), for both $\hat{\rho}$ and $\hat{\Pi}_{\Sigma_d}\hat{\Pi}_A\hat{\Pi}_{\Sigma_d}$ (where we have chosen to replace $\hat{\Pi}_A$ with its restriction to $\Sigma_d$ for convenience, while $\hat{\rho}$ is already restricted to $\Sigma_d$ by assumption):

$$\sum_{\substack{n,m: \\ |E_n - E_m| \ge \Delta E}} \left| \langle E_n|\hat{\rho}|E_m\rangle \right|^2 \le \mathrm{Tr}[\hat{\rho}^2] \tag{D.29}$$

$$\sum_{\substack{n,m: \\ |E_n - E_m| \ge \Delta E}} \left| \langle E_m|\hat{\Pi}_A|E_n\rangle \right|^2 \le \mathrm{Tr}\left[ \left\{ \hat{\Pi}_A\hat{\Pi}_{\Sigma_d} \right\}^2 \right]. \tag{D.30}$$

For $\hat{\Pi}_A$ in particular, as the operator $\hat{\Pi}_{\Sigma_d}\hat{\Pi}_A\hat{\Pi}_{\Sigma_d}$ has eigenvalues in $[0,1]$, the squared trace of this operator is bounded from above by its trace; this gives (using $\hat{\Pi}_{\Sigma_d}^2 = \hat{\Pi}_{\Sigma_d}$ and the cyclic property of the trace):

$$\mathrm{Tr}\left[ \left\{ \hat{\Pi}_A\hat{\Pi}_{\Sigma_d} \right\}^2 \right] \le \mathrm{Tr}[\hat{\Pi}_A\hat{\Pi}_{\Sigma_d}] = d\,\langle \hat{\Pi}_A\rangle_{\Sigma_d}, \tag{D.31}$$

where we have recognized the expression for the thermal value $\langle \hat{\Pi}_A\rangle_{\Sigma_d}$ of $\hat{\Pi}_A$ in the energy shell $\Sigma_d$ [see Eq. (48)]. Combining these observations, we obtain the overall inequality:

$$\left| \sum_{\substack{n,m: \\ |E_n - E_m| \ge \Delta E}} \widetilde{w}(E_n - E_m)\langle E_n|\hat{\rho}|E_m\rangle\langle E_m|\hat{\Pi}_A|E_n\rangle \right| \le w_0 \sqrt{d\,\mathrm{Tr}[\hat{\rho}^2]\langle \hat{\Pi}_A\rangle_{\Sigma_d}}. \tag{D.32}$$

Inserting Eqs. (D.27) and (D.32) into Eq. (D.24), we get Eq. (75).

 **D.3.3 Theorem 5.3: Energy-band thermalization implies almost all physical basis states**
 **thermalize o.a.**

2335 Noting that $\lambda_k[w] \in \mathbb{R}$ [defined in Eq. (83)], it is convenient to separate the basis states in $\mathcal{B}$
2336 into a set of states $\mathcal{B}_+$ in which $\lambda_k[w] \geq 0$, numbering $n_+$, and a set $\mathcal{B}_-$ in which $\lambda_k[w] < 0$,
2337 with $n_-$ states:

$$\mathcal{B}_+ = \{|k\rangle \in \mathcal{B} : \lambda_k[w] \geq 0\}, \quad |\mathcal{B}_+| = n_+, \tag{D.33}$$

$$\mathcal{B}_- = \{|k\rangle \in \mathcal{B} : \lambda_k[w] < 0\}, \quad |\mathcal{B}_+| = n_-. \tag{D.34}$$

2338 We can now form the mixed states:

$$\hat{\rho}_+ \equiv \frac{1}{n_+} \sum_{|k\rangle \in \mathcal{B}_+} |k\rangle\langle k|, \tag{D.35}$$

$$\hat{\rho}_- \equiv \frac{1}{n_-} \sum_{|k\rangle \in \mathcal{B}_-} |k\rangle\langle k|, \tag{D.36}$$

2339 such that $\mu(\rho_\pm) = n_\pm/d$, where $\mu(\rho) = 1/(d\,\mathrm{Tr}[\hat{\rho}^2])$ as in Eq. (78).
2340    Given that $\hat{\Pi}_A$ shows energy-band thermalization with accuracy $\epsilon$, Theorem 5.2 applied to
2341 $\hat{\rho}_\pm$ implies that

$$\frac{1}{n_\pm} \sum_{k \in \mathcal{B}_\pm} \left| \lambda_k[w] \right| < \left( \epsilon + w_0 \sqrt{\langle \hat{\Pi}_A \rangle_{\Sigma_d}} \right) \sqrt{\frac{d}{n_\pm}}, \tag{D.37}$$

2342 from which we obtain (by adding both the $\rho_+$ and $\rho_-$ versions with the appropriate weights
2343 $n_+$ and $n_-$ multiplying them):

$$\frac{1}{d} \sum_{k \in \mathcal{B}_\pm} \left| \lambda_k[w] \right| < \left( \epsilon + w_0 \sqrt{\langle \hat{\Pi}_A \rangle_{\Sigma_d}} \right) \frac{\sqrt{n_+} + \sqrt{n_-}}{\sqrt{d}} \leq \left( \epsilon + w_0 \sqrt{\langle \hat{\Pi}_A \rangle_{\Sigma_d}} \right) \sqrt{2}, \tag{D.38}$$

2344 where the second inequality follows from $\sqrt{n_+} + \sqrt{n_-} < \sqrt{2d}$ as $n_+ + n_- = d$.
2345    Finally, if we assume that a fraction $f$ of the $|\lambda_k[w]|$ are at least $\lambda$ and the rest are smaller
2346 than $\lambda$, we have

$$f\lambda \leq \frac{1}{d} \sum_{k \in \mathcal{B}_\pm} \left| \lambda_k[w] \right|, \tag{D.39}$$

2347 which, on substitution into Eq. (D.38), gives Eq. (85).

2348 **D.4 Global thermalization from a single initial state**

2349 **D.4.1 Proposition 6.1: Autocorrelators that thermalize o.a. imply global energy-band**
2350 **thermalization**

2351 From Eq. (91), we have

$$\int \mathrm{d}t \, w_+(t) \, \mathrm{Tr}[\hat{\rho}_A(t)\hat{\Pi}_A] - \langle \hat{\Pi}_A \rangle = \frac{1}{\mathrm{Tr}[\hat{\Pi}_A]} \sum_{\substack{n,m: \\ |E_n - E_m| < \Delta E}} \widetilde{w}_+(E_n - E_m) \left| \langle E_n | \left( \hat{\Pi}_A - \langle \hat{\Pi}_A \rangle \hat{\mathbb{1}} \right) | E_m \rangle \right|^2$$
$$+ \frac{1}{\mathrm{Tr}[\hat{\Pi}_A]} \sum_{\substack{n,m: \\ |E_n - E_m| \geq \Delta E}} \widetilde{w}_+(E_n - E_m) \left| \langle E_n | \left( \hat{\Pi}_A - \langle \hat{\Pi}_A \rangle \hat{\mathbb{1}} \right) | E_m \rangle \right|^2 \tag{D.40}$$

As $\widetilde{w}_+(E_n - E_m) \geq 0$, the second line is non-negative, and we can write:

$$\int \mathrm{d}t \, w_+(t) \operatorname{Tr}[\hat{\rho}_A(t)\hat{\Pi}_A] - \langle\hat{\Pi}_A\rangle \geq \frac{1}{\operatorname{Tr}[\hat{\Pi}_A]} \sum_{\substack{n,m: \\ |E_n - E_m| < \Delta E}} \widetilde{w}_+(E_n - E_m) \left|\langle E_n|\left(\hat{\Pi}_A - \langle\hat{\Pi}_A\rangle\hat{\mathbb{1}}\right)|E_m\rangle\right|^2 \tag{D.41}$$

By assumption, $\widetilde{w}_+(E_n - E_m) > W$ for $|E_n - E_m| < \Delta E_W$, and $\Delta E < \Delta E_W$. We therefore obtain the inequality (swapping the two sides of the above equation):

$$\frac{1}{D} \sum_{\substack{n,m: \\ |E_n - E_m| < \Delta E}} \left|\langle E_n|\left(\hat{\Pi}_A - \langle\hat{\Pi}_A\rangle\hat{\mathbb{1}}\right)|E_m\rangle\right|^2 < \frac{\operatorname{Tr}[\hat{\Pi}_A]}{WD}\left[\int \mathrm{d}t \, w_+(t)\operatorname{Tr}[\hat{\rho}_A(t)\hat{\Pi}_A] - \langle\hat{\Pi}_A\rangle\right]. \tag{D.42}$$

The left hand side contains the relevant quantity for energy-band thermalization in the full Hilbert space $\mathcal{H}$, with $D = \dim\mathcal{H}$; Proposition 6.1 follows from here.

## D.5 Energy shell thermalization from quantum dynamical echoes

### D.5.1 Theorem 7.1: Quantum dynamical echoes constrain energy-band thermalization in energy shells

Starting from Eq. (129) and restricting the matrix elements to pairs of energy levels $(E_n, E_m)$ where both are within the energy shell $\Sigma_d$ and satisfy $|E_n - E_m| < \Delta E$, we obtain the inequality (again due to the non-negativity of the right hand side):

$$\operatorname{Tr}\left[\hat{\gamma}_A^{\mathrm{shell}}\left(\hat{\Pi}_A - \langle\hat{\Pi}_A\rangle_{\Sigma_d}\hat{\mathbb{1}}\right)\right] \geq \frac{1}{\operatorname{Tr}[\hat{\Pi}_A]} \sum_{\substack{n,m: \\ |E_n\rangle, |E_m\rangle \in \Sigma_d, \\ |E_n - E_m| < \Delta E}} \widetilde{w}_+(E_n - E_m)\widetilde{v}_+\left(\frac{E_n + E_m}{2} - E_c\right)\left|\langle E_m|\left(\hat{\Pi}_A - \langle\hat{\Pi}_A\rangle_{\Sigma_d}\hat{\mathbb{1}}\right)|E_n\rangle\right|^2. \tag{D.43}$$

Using the constraints on $\widetilde{v}_+$ and $\widetilde{w}_+$ within this band, we have $\widetilde{w}_+(E_n - E_m) > W$ by Eq. (98), and also $\widetilde{v}_+((E_n + E_m)/2 - E_c) \geq V$ by Eq. (127), noting that $(E_n + E_m)/2 \in \mathcal{E}_V$ if $|E_n\rangle, |E_m\rangle \in \Sigma_d \subseteq \Sigma(\mathcal{E}_V)$. This leads to the inequality (after transposing the left and right hand sides):

$$\frac{1}{d} \sum_{\substack{n,m: \\ |E_n\rangle, |E_m\rangle \in \Sigma_d, \\ |E_n - E_m| < \Delta E}} \left|\langle E_m|\left(\hat{\Pi}_A - \langle\hat{\Pi}_A\rangle_{\Sigma_d}\hat{\mathbb{1}}\right)|E_n\rangle\right|^2 < \frac{\operatorname{Tr}[\hat{\Pi}_A]}{WVd}\operatorname{Tr}\left[\hat{\gamma}_A^{\mathrm{shell}}\left(\hat{\Pi}_A - \langle\hat{\Pi}_A\rangle_{\Sigma_d}\hat{\mathbb{1}}\right)\right], \tag{D.44}$$

which implies Eq. (131).

### D.5.2 Theorem 7.2: Energy-band thermalization in energy shells constrains quantum dynamical echoes

From Eqs. (111) and (120), we have

$$\operatorname{Tr}\left[\hat{\gamma}_A^{\mathrm{shell}}\left(\hat{\Pi}_A - \langle\hat{\Pi}_A\rangle_{\Sigma_d}\hat{\mathbb{1}}\right)\right] = \frac{1}{\operatorname{Tr}[\hat{\Pi}_A]} \sum_{n,m} \widetilde{w}(E_n - E_m)\widetilde{v}\left(\frac{E_n + E_m}{2} - E_c\right)\left|\langle E_m|\left(\hat{\Pi}_A - \langle\hat{\Pi}_A\rangle_{\Sigma_d}\hat{\mathbb{1}}\right)|E_n\rangle\right|^2 \tag{D.45}$$

Schematically, recalling the definition of $\mathcal{E}_{\Delta E}$ as the range of energies of the energy shell further away than $\Delta E$ from its boundaries, we can split this as follows:

(Full expression) =

$$\text{(Terms with } |E_n - E_m| < \Delta E \text{ and } E_n, E_m \in \Sigma_d) \tag{D.46}$$

$$+\text{(Terms with } |E_n - E_m| < \Delta E \text{ and } [(E_n \notin \Sigma_d) \text{ or } (E_m \notin \Sigma_d)]) \tag{D.47}$$

$$+\text{(Terms with } |E_n - E_m| \geq \Delta E \text{ and } (E_n + E_m)/2 \notin \mathcal{E}_{\Delta E}) \tag{D.48}$$

$$+\text{(Terms with } |E_n - E_m| \geq \Delta E \text{ and } (E_n + E_m)/2 \in \mathcal{E}_{\Delta E}) \tag{D.49}$$

For the first kind of terms in Line (D.46), using the triangle inequality and $|\widetilde{v}(E)| \leq 1$ and $|\widetilde{w}(\delta E)| \leq 1$ (from positivity, but not necessarily complete positivity), we have:

$$|(\text{Line (D.46)})| \leq \frac{1}{\text{Tr}[\hat{\Pi}_A]} \text{Tr}\left\{\left[\hat{\Pi}_A - \langle \hat{\Pi}_A \rangle_{\Sigma_d} \hat{\mathbb{1}}\right]^2_{\Sigma_d, \Delta E}\right\}. \tag{D.50}$$

For the second kind of terms in Line (D.47), we use the triangle inequality with $|\widetilde{v}(E)| < v_0$ and $|\widetilde{w}(\delta E)| \leq 1$; the former condition on $v$ follows by assumption, due to the fact that if either of $E_n, E_m$ are not in $\Sigma_d$ but are subject to $|E_n - E_m| < \Delta E$, then the interval $[(E_n + E_m)/2 - \Delta E, (E_n + E_m)/2 + \Delta E]$ is not contained in $\mathcal{E}$, and therefore $(E_n + E_m)/2 \notin \mathcal{E}_{\Delta E}$. In addition, we can add nonnegative terms with $v_0/\text{Tr}[\hat{\Pi}_A]$ multiplying the squared magnitude of the matrix elements of $(\hat{\Pi}_A - \langle \hat{\Pi}_A \rangle_{\Sigma_d} \hat{\mathbb{1}})$ within the energy shell to obtain the trace of its square (similar to "completing the square"), and the result must be greater than or equal to the original expression due to adding nonnegative terms. This gives the inequality:

$$|(\text{Line (D.47)})| < \frac{v_0}{\text{Tr}[\hat{\Pi}_A]} \text{Tr}\left[\left(\hat{\Pi}_A - \langle \hat{\Pi}_A \rangle_{\Sigma_d} \hat{\mathbb{1}}\right)^2\right] \leq v_0, \tag{D.51}$$

where we have noted that

$$\text{Tr}\left[\left(\hat{\Pi}_A - \langle \hat{\Pi}_A \rangle_{\Sigma_d} \hat{\mathbb{1}}\right)^2\right] \leq \text{Tr}[\hat{\Pi}_A^2] = \text{Tr}[\hat{\Pi}_A]. \tag{D.52}$$

Applying a similar strategy to Line (D.48), where in addition to $|\widetilde{v}(E)| < v_0$ as $(E_n + E_m)/2 \notin \mathcal{E}_{\Delta E}$, we use $|\widetilde{w}(\delta E)| \leq w_0$ as $|E_n - E_m| < \Delta E$ and add the remaining absolute squared matrix elements of $(\hat{\Pi}_A - \langle \hat{\Pi}_A \rangle_{\Sigma_d} \hat{\mathbb{1}})$ multiplied by $w_0 v_0/\text{Tr}[\hat{\Pi}_A]$ to get:

$$|(\text{Line (D.48)})| < v_0 w_0. \tag{D.53}$$

Finally, for Line (D.49), we use $|\widetilde{w}(\delta E)| \leq w_0$ and $|\widetilde{v}(E)| \leq 1$ and apply a similar strategy as above to obtain the trace of the squared operator to get

$$|(\text{Line (D.49)})| \leq w_0. \tag{D.54}$$

On the whole, we are left with the inequality:

$$\text{Tr}\left[\hat{\gamma}_A^{\text{shell}}\left(\hat{\Pi}_A - \langle \hat{\Pi}_A \rangle_{\Sigma_d} \hat{\mathbb{1}}\right)\right] < \frac{1}{\text{Tr}[\hat{\Pi}_A]} \text{Tr}\left\{\left[\hat{\Pi}_A - \langle \hat{\Pi}_A \rangle_{\Sigma_d} \hat{\mathbb{1}}\right]^2_{\Sigma_d, \Delta E}\right\} + v_0 + v_0 w_0 + w_0, \tag{D.55}$$

which gives Eq. (135).

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
