# Peer review of "Bypassing eigenstate thermalization with experimentally accessible quantum dynamics"

_SciPost Physics_

## Round 1 · Referee Report · Anonymous (Referee 1) · 2025-12-25

The referee discloses that the following generative AI tools have been used in the preparation of this report:
I use chat GPT for correcting typo for the reports. I stress that I do not use AI for giving technical contents.
Strengths
1)
In particular, the notion of energy-band thermalization discussed in Section~5 provides a nice resolution to subtleties that many papers typically dismiss by assumption for simplicity.
2)
The discussion in Section~6 offers a robust clarification of finite-time thermalization. To my knowledge, this point has not been discussed clearly in the existing literature.
3)
I have checked the derivations of the inequalities, and they appear to be mathematically correct.
Weaknesses
1)
Although the authors aim to address realistic situations, they do not discuss cases involving multiple observables that may not commute with each other.
2)
In several places, the authors refer to a ``physical basis.'' However, the meaning of this term is unclear to me. In realistic situations, and even from a theoretical perspective, one generally considers superpositions of such bases.
Report
Requested changes
I add the pdf file which including comments and request for revision.
Recommendation
Publish (meets expectations and criteria for this Journal)

---

## Editorial Decision

unknown